# *First is Not Really Better Than Last*: EVALUATING LAYER CHOICE AND AGGREGATION STRATEGIES IN LANGUAGE MODEL DATA INFLUENCE ESTIMATION

**Dmytro Vitel & Anshuman Chhabra**
Bellini College of Artificial Intelligence, Cybersecurity, and Computing,
University of South Florida
`{dvitel,anshumanc}@usf.edu`

## ABSTRACT

Identifying how training samples influence/impact Large Language Model (LLM) decision-making is essential for effectively interpreting model decisions and auditing large-scale datasets. Current training sample influence estimation methods (also known as *influence functions*) undertake this goal by utilizing information flow through the model via its first-order and higher-order gradient terms. However, owing to the large model sizes of today consisting of billions of parameters, these influence computations are often restricted to some subset of model layers to ensure computational feasibility. Prior seminal work by Yeh et al. (2022) in assessing which layers are best suited for computing language data influence concluded that the first (*embedding*) layers are the most informative for this purpose, using a hypothesis based on influence scores canceling out (i.e., the *cancellation effect*). In this work, we propose theoretical and empirical evidence demonstrating how the cancellation effect is *unreliable*, and that *middle* attention layers are better estimators for influence. Furthermore, we address the broader challenge of *aggregating* influence scores across layers, and showcase how alternatives to standard averaging (such as ranking and vote-based methods) can lead to significantly improved performance. Finally, we propose better methods for evaluating influence score efficacy in LLMs without undertaking model retraining, and propose a new metric known as the Noise Detection Rate (NDR) that exhibits strong predictive capability compared to the cancellation effect. Through extensive experiments across LLMs of varying types and scales, we concretely determine that the *first* (layers) are not *necessarily better* than the *last* (layers) for LLM influence estimation, contrasting with prior knowledge in the field.

## 1 INTRODUCTION

Large Language Models (LLMs) have demonstrated stellar performance on tasks across a number of applications and domains (Street et al., 2024; Mittelstädt et al., 2024; Marco et al., 2025). Despite these advancements, current state-of-the-art models still exhibit suboptimal behavior on complex reasoning tasks (Jiang et al., 2025), hallucinate responses and facts (Cleti and Jano, 2024), and can make biased and unfair decisions (Gallegos et al., 2024; Peters and Chin-Yee, 2025). To improve the trust and safety of LLMs, it is imperative to better interpret and understand model decision-making (Singh et al., 2024).

Recently, data attribution and influence methods (Hammoudeh and Lowd, 2024) have shown great promise in interpreting LLM behavior from the perspective of the training data (Grosse et al., 2023; Chhabra et al., 2025). These approaches seek to detect noisy, anomalous, out-of-distribution, or problematic training samples and conceptually link them to model output performance (Pleiss et al., 2020; Yang et al., 2024; Jiang et al., 2021). Even highly curated datasets can contain detrimental samples, often introduced unknowingly through human error or surreptitiously by malicious adversaries (Ekambaram et al., 2017). More specifically, *influence functions* (Pruthi et al., 2020; Sui et al., 2021; Kwon et al., 2024) employ gradient-based analysis to help developers understand how models' decisions are influenced by training data. Despite their clear benefits, a fundamental issue in influence computation is the computational overhead of utilizing the entire gradient space from all the layers of the LLM, which, for modern models, amount to billions of parameters.

A common workaround (Figure 1) when employing influence functions in LLMs is to restrict the gradient input to only certain layers (Yeh et al., 2022; Chhabra et al., 2025). However, there is little consensus on what the best layers are for influence estimation, and it is not known whether the output head, attention layers, or word embedding layers are better suited for this purpose. This is partly because very little work has been undertaken to analyze which layers are more use-

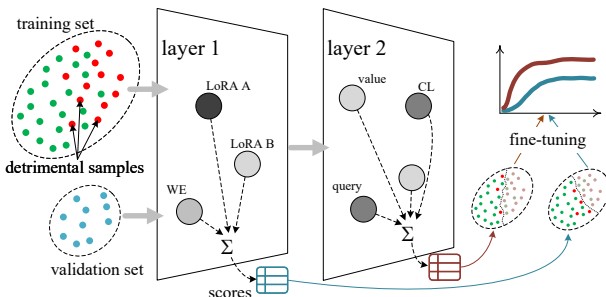

Figure 1: The influence estimation pipeline for LLMs.

ful for influence analysis. More specifically, one prior seminal work by Yeh et al. (2022) analyzes layer choice in computing language model data influence and finds that the first few (embedding) layers are most beneficial for influence estimation. However, as we discuss subsequently in our paper, these conclusions are derived from a strong assumption about the observed effectiveness of the layers being used, which might not hold in practice, requiring a deeper investigation into the effect of layers. Additionally, the work only studies one influence method and smaller-sized language models such as RoBERTa (Liu et al., 2019), whereas we undertake our study across various LLM parameter sizes and recent models, thereby capturing general patterns of influence estimation as models scale.

Moreover, influence functions that utilize gradients from multiple layers generally aggregate their contributions through averaging across parameter subsets. However, this averaging can obscure important distinctions by allowing compensatory effects between layers, potentially masking the true contribution of individual training samples. In this work, we also explore and propose alternative strategies for combining influence estimates computed independently across different layers, aiming to preserve layer-specific insights and improve the interpretability of influence attribution.

Finally, assessing new influence-based attribution methods involves measuring how accurately they identify detrimental training samples. In practice, this evaluation is typically performed by removing the least influential samples, fine-tuning the model on the filtered dataset, and then observing the impact on downstream task performance. However, this process is computationally expensive, particularly when influence is measured across different layers or parameter subsets of a model. As a result, it becomes valuable to identify a reliable extrinsic indicator – one that can be computed efficiently and correlates well with the final task accuracy. Such a metric can guide the development and selection of influence-based attribution methods without requiring exhaustive fine-tuning experiments. To this end, we also propose a novel metric (the Noise Detection Rate) and show how it can be used as a reliable proxy indicator of downstream influence estimation performance.

**Contributions.** In sum, our contributions are as follows:
- We evaluate the cancellation effect metric proposed by Yeh et al. (2022) for identifying layers most suitable for influence estimation. Our theoretical analysis and experiments demonstrate that, when filtering large numbers of samples, the cancellation effect is an unreliable indicator.
- We quantify the impact of detrimental training samples on language model behavior and identify the model layers most informative for influence estimation. We conduct extensive experiments using noisy versions of the GLUE benchmark as in prior influence work (Kwon et al., 2024; Chhabra et al., 2025) and employ several LLMs such as Llama-3.2, Qwen-2.5, and Mistral 7B.
- We also introduce novel strategies for aggregating influence scores across layers, and demonstrate that they frequently outperform the standard averaging method used in current functions.
- We propose and evaluate the validity of noise distribution-based proxy measures: Noise Detection Rate (NDR) metric and Area Under the Curve (AUC), as reliable indicators of downstream influence estimation performance, with the goal of minimizing reliance on exhaustive fine-tuning for assessing influence estimation accuracy.

## 2 RELATED WORKS

Our work falls within the broader area of interpretable machine learning, particularly focusing on gradient-method estimations of training data importance (Koh and Liang, 2017; Yeh et al., 2018; 2019; Jia et al., 2019; Pruthi et al., 2020; Khanna et al., 2018; Sui et al., 2021). In gradient-based

models trained via empirical risk minimization, influence functions offer a practical means to estimate the impact of individual training samples without performing costly leave-one-out retraining. For deep learning, the foundational work (Koh and Liang, 2017) introduced a Taylor-series approximation combined with the LiSSA optimization algorithm (Agarwal et al., 2017) to make influence estimation tractable. Subsequent efforts, such as Representer Point (Yeh et al., 2018) and Hydra (Chen et al., 2021), aimed to improve these estimates, though their focus remained primarily on vision tasks. More recent advancements, including DataInf (Kwon et al., 2024), Arnoldi iteration (Schioppa et al., 2022), and Kronecker-factored curvature approximations (Grosse et al., 2023), have scaled influence estimation to large generative language models like LLMs. Simpler methods have also emerged, using raw gradients as a proxy for influence (Pruthi et al., 2020; Charpiat et al., 2019), often enhanced with ensemble-based strategies (Bae et al., 2024; Kim et al., 2023). Additionally, self-influence – computed solely within the training set – has been shown to be effective for identifying influential samples (Bejan et al., 2023; Thakkar et al., 2023). Influence functions have proven valuable across diverse applications, including classification (Koh and Liang, 2017; Chhabra et al., 2025; 2024), generative modeling (Kwon et al., 2024; Schioppa et al., 2022; Grosse et al., 2023), active learning (Liu et al., 2021), in-context learning (Nguyen and Wong, 2023; S. et al., 2024; Askari et al., 2025a), and layer quality estimation (Askari et al., 2025b).

Gradient compensation in feature-attributed gradients has been studied previously (Liu et al., 2020; Kapishnikov et al., 2021). In data attribution, influence functions have been used to detect spurious training artifacts and improve performance (Han et al., 2020; Pezeshkpour et al., 2022; Askari et al., 2025a). Prior work (Hammoudeh and Lowd, 2024; Yeh et al., 2022) highlights that dot-product-based methods and high gradient cancellation may reduce the discriminative power of influence scores. We extend this line of work by empirically and theoretically investigating whether such cancellation impairs the detection of mislabeled data.

It is also interesting to note that prior work has studied the *applicability* of influence functions in LLMs, with varying results. Clearly, while influence functions have been used successfully across several LLM tasks and applications Wang et al. (2025); Zhang et al. (2025); Askari et al. (2025b); Xia et al. (2024), recent work Li et al. (2025) finds that Hessian-Free, DataInf, and LiSSA perform poorly on tasks such as harmful data detection, class attribution, and backdoor identification, due to iHVP approximation errors, uncertain convergence, and weak correlation between parameter changes and model behavior. Their experiments use default influence-function implementations with mean aggregation across all layers. In contrast, we extend these insights by identifying specific layers and aggregation strategies that achieve performance capable of exceeding the default settings in Li et al. (2025). As part of future work, we believe indetifying optimal layers for influence analysis can significantly improve the efficacy and fidelity of influence functions in LLMs, as observable in our main paper results.

Moreover, given our focus on *where* influence scores are strongest and *how* individual valuations can be effectively combined, we relate our analysis to knowledge-editing methods that similarly perform localization and aggregation. ROME (Meng et al., 2022) and R-ROME (Gupta et al., 2024a) use *causal tracing* to identify the MLP layer encoding a fact and apply a rank-one edit. MEMIT (Meng et al., 2023) generalizes this via per-layer causal-effect scores and top-$k$ distributed edits, while EMMET (Gupta et al., 2024b) refines aggregation to reduce redundancy. In contrast, we use gradient-based influence to locate layers most affected by beneficial or harmful training samples and, consistent with KE studies, find the strongest discriminative signals in middle layers.

## 3 PRELIMINARIES AND NOTATION

### 3.1 INFLUENCE FUNCTIONS

Given a training sample $\bar{x} \in X$, model weights $\Theta$, and a validation sample $\bar{x}' \in X'$, the influence is defined as follows for the utility function $f$.

$$I(\bar{x}, X', \Theta) = \left( \frac{1}{|X'|} \sum_{\bar{x}' \in X'} \nabla_\Theta f(\bar{x}', \Theta) \right)^T H_\Theta^{-1} \nabla l(\bar{x}, \Theta) \tag{1}$$

The optimization step $\Delta\Theta_1$ in the direction of the vector given by the term under the parentheses decreases $f$ on $X'$. The second term, $H_\Theta^{-1}\nabla l(\bar{x}, \Theta)$ ($H$ - Hessian matrix) defines the optimization step $\Delta\Theta_2$ that would minimize the loss on the sample $\bar{x}$.

The loss optimization on some training samples could be misaligned with the optimization of $f$. Therefore, fitting these samples would increase $f$. We refer to such samples as *detrimental* for the utility under consideration. This definition corresponds to the previously used definition in (Koh and Liang, 2017; Kwon et al., 2024; Chhabra et al., 2025). In practice, it is not feasible to compute $I$ on large models due to the time complexity of the Hessian inversion (cubic time in the exact case). Therefore, influence is estimated approximately. The TracIn method (Pruthi et al., 2020) replaces the *Hessian vector product* $\mathcal{H} = H_\Theta^{-1} \nabla l(\bar{x}, \Theta)$ with the gradient $\nabla l(\bar{x}, \Theta)$. Cosine similarity between the utility term and $\mathcal{H} \approx \nabla l(\bar{x}, \Theta)$ serves as another first-order influence estimation approximation. DataInf (Kwon et al., 2024) is a recently proposed second-order method based on swapping the order of matrix inversion and averaging across training samples in $\mathcal{H}$. Its error decreases when $|\Theta|$ is small, which makes this method suitable for fine-tuning with LoRA (Hu et al., 2022; Dettmers et al., 2023).

For cross-entropy loss, the Hessian is the block diagonal matrix of gradient second moments $|X|^{-1} \sum_{\bar{x} \in X} \nabla l(\bar{x}, \Theta) \nabla l^T(\bar{x}, \Theta)$, one block for one layer $l \in L$ in the neural network. Thus, $I$ is an aggregation of influence values across validation samples and model layers, defined as follows:

$$I(\bar{x}, X', \Theta) = \frac{1}{|X'|} \sum_{\bar{x}' \in X', l \in L} I'(\bar{x}, \bar{x}', \Theta_l) \text{ s.t. } I'(\bar{x}, \bar{x}', \Theta_l) = \nabla_{\Theta_l} f^T(\bar{x}', \Theta_l) \mathcal{H}(\bar{x}, \Theta_l) \tag{2}$$

$I'(\bar{x}, \bar{x}', \Theta_l)$ are tensors collected on different layers and for different influence functions. We employ TracIn, Cosine, and DataInf in our experiments ($\nabla l_{\bar{x}} = \nabla_{\Theta_l} l(\bar{x}, \Theta_l)$), analytically defined as follows:

$$\text{TracIn}(\bar{x}, \bar{x}', \Theta_l) = \nabla l_{\bar{x}'}^T \nabla l_{\bar{x}}. \tag{3}$$

$$\text{Cosine}(\bar{x}, \bar{x}', \Theta_l) = \frac{\nabla l_{\bar{x}'}^T \nabla l_{\bar{x}}}{|\nabla l_{\bar{x}'}^T| |\nabla l_{\bar{x}}^T|}. \tag{4}$$

$$\text{DataInf}(\bar{x}, \bar{x}', \Theta_l) = \frac{\nabla l_{\bar{x}'}^T \nabla l_{\bar{x}} - |X|^{-1} \sum_{\bar{z} \in X} \nabla l_{\bar{x}'}^T \nabla l_{\bar{z}} \nabla l_{\bar{z}}^T \nabla l_{\bar{x}}}{(|X||l|\lambda)^{-1} \sum_{\bar{x} \in X} |\nabla l_{\bar{x}}|^2}. \tag{5}$$

To address the question of which layers are more effective at detecting detrimental samples, we also consider two influence functions proposed by Yeh et al. (2022). Both of them, $\text{TracIn}_{we}$ and $\text{TracIn}_{we}^{10}$, operate on word embeddings. The first, $\text{TracIn}_{we}$, considers only the weights of the tokens (including special tokens) in both the training and the validation samples:

$$\text{TracIn}_{we}(\bar{x}, \bar{x}', \Theta_{we}) = \sum_{t \in \bar{x} \cap \bar{x}', \theta_t \in \Theta_{we}} \frac{\partial l(\bar{x}', \theta_t)}{\partial \theta_t} \frac{\partial l(\bar{x}, \theta_t)}{\partial \theta_t}. \tag{6}$$

The second, $\text{TracIn}_{we}^{10}$ additionally filters out the weights $t \in \bar{x} \cap \bar{x}'$, $\theta_t \in \Theta_{we}$ keeping only the top-10 that have highest-by-magnitude gradients for training sample $\bar{x}$.

## 3.2 Cancellation Effect

In their study of layer choice effects in influence estimation, Yeh et al. (2022) consider the compensation of gradients as contributed by training samples. They argue that if the sum of gradients w.r.t a particular weight $\theta$ becomes very small, then the optimizer would not change $\theta$ significantly because of them. However, the magnitudes of the summed training gradients could be very large and hence, the $\theta$-contributed influence score could also be large. The authors further state that $\theta$, the weight with a high cancellation effect, should be ignored in influence estimation because it reduces the model's discrimination power. To this end, they propose a metric $C$ to estimate the cancellation effect on a layer $W$ and present the results where smaller $C$ values lead to better model confidence (Yeh et al., 2022):

$$C(W) = \frac{\sum_{\bar{x}} |\nabla_W l(\bar{x}, W)|}{|\Delta W|}. \tag{7}$$

$C$ can be computed throughout the fine-tuning procedure, considering every checkpoint individually. Accumulating the magnitudes of atomic gradient updates per sample requires *size-one batching* during training, which is computationally expensive and infeasible for current models. We thus adopt the original formulation to estimate the cancellation on a single checkpoint in evaluation mode.

### 3.3 Proposed Influence Aggregation Approaches

As is evident, Eq. 2 averages influence scores across all layers and validation samples. This can cause influence *compensation*, where opposing contributions cancel out, or *dominance*, where large scores overshadow others. In our work, we aim to investigate alternative aggregation strategies that can be employed instead to *mitigate* these effects. Generally, influence attribution can be abstracted as:

$$I(\bar{x}, X', \Theta_L) = \mathcal{A}_{X',L}(I'(\cdot, \bar{x}', \Theta_l))(\bar{x}), \tag{8}$$

where $\mathcal{A}$ is an *aggregation operator* (e.g., taking the average across samples/layers). We thus propose our novel aggregation approaches as follows:

**Ranking**. Every validation sample and layer ranks training samples according to their influence. These ranks are summed, eliminating the domination of individual influences with high magnitude. Additionally, our proposed `Rank` method ignores incorrectly predicted validation samples by the selected checkpoint:

$$\text{Rank}(I') = \sum_{\bar{x}' \in X'', l \in L} \sum_{\bar{y} \in X} \mathbb{I}(I'(\bar{y}, \bar{x}', \Theta_l) < I'(\cdot, \bar{x}', \Theta_l)), \tag{9}$$

where $\mathbb{I}$ is the indicator function, and $X''$ denotes the correctly predicted validation samples.

**Positional voting**. To avoid the domination of very low or very high ranks in averaging, we propose a method based on voting. Every validation sample and layer assigns $k$ votes to the least influential training datum, $k-1$ to the next one, etc., until the number of votes reaches zero. We then pick $k$ equal to the number of training samples for filtering as potentially detrimental. As with `Rank`, the `Vote` method considers only correctly predicted validation samples by the selected checkpoint:

$$\text{Vote}_k(I') = -\sum_{\bar{x}' \in X'', l \in L} \max(k - \sum_{\bar{y} \in X} \mathbb{I}(I'(\bar{y}, \bar{x}', \Theta_l) < I'(\cdot, \bar{x}', \Theta_l)), 0). \tag{10}$$

## 4 Research Questions

We now formalize the research questions (RQs) we seek to study in this paper. To bridge the gaps identified above regarding the role of layers in LLM influence analysis, the impact of layer-wise aggregation strategies in influence functions, and whether influence functions can be evaluation through novel external metrics, without requiring costly fine-tuning, we propose the following RQs:

**[RQ1]**. *Can the gradient cancellation effect serve as a reliable predictor of layer contribution in influence estimation?* In Section 5.1, we propose both theoretical and empirical evidence to answer RQ1, showing that high cancellation within a layer does not necessarily imply poor attribution performance.

**[RQ2]**. *Which model layers yield the most effective influence scores for detecting detrimental samples?* Addressing this question (Section 5.2) serves two practical goals: (1) improving training data filtering methods, and (2) revealing which parts of the model are most sensitive to such samples. These efforts are motivated by the lack of general consensus on layer choice in influence estimation.

**[RQ3]**. *Can alternative aggregation strategies improve influence estimation performance compared to traditional averaging?* Moving beyond influence score averaging across layers, we introduce two novel aggregation methods (Section 3.3): `Rank` and `Vote`, for combining individual influence scores. In Section 5.3, we evaluate their impact on model accuracy following sample filtering.

**[RQ4]**. *How reliably can the detrimental sample distribution measures predict the downstream performance of influence-based data filtering methods?* In Section 5.4, we study whether our novel proposed metrics: Noise Detection Rate (NDR) and Area Under the Curve (AUC), can serve as useful metrics for assessing the performance of influence functions, without requiring expensive fine-tuning.

## 5 Analysis and Experimental Results

**Datasets**. We evaluate detrimental sample filtering across eight GLUE datasets (Wang et al., 2018): QNLI, MRPC, QQP, SST-2, CoLA, MNLI, RTE, and STS-B. These benchmarks cover a range of

natural language understanding tasks, including sentence similarity, paraphrase detection, sentiment analysis, linguistic acceptability, and natural language inference. Together, they provide a diverse and widely used testbed for assessing model performance and the impact of data quality.

**Models**. The experiments are conducted on several LLM types: RoBERTa-Large (Liu et al., 2019), LLaMA-3.2 1B (Touvron et al., 2023), Qwen-2.5 1.5B (Qwen et al., 2025), and Mistral 7B (Jiang et al., 2023). These models represent various widely used transformer-based language models ranging from millions to several billions of parameters, offering a comprehensive view of influence behavior across model sizes and architectures.

**Methodology and Protocol**. Similar to prior work (Kwon et al., 2024; Chhabra et al., 2025), we employ the five-stage pipeline detailed further in Appendix A): (1) Inject synthetic noise into training data and initialize from compressed embeddings. (2) Fine-tune on noisy data, selecting checkpoints with the lowest validation loss (Pruthi et al., 2020), which we verify outperforms final-epoch or accuracy-based selection. (3) Compute influence values across all tunable layers. (4) Partition the model into embeddings (`WE`), attention groups (four splits), and classification head (`CL`), aggregating influence per training sample. (5) Remove 30% least influential samples and retrain, evaluating effectiveness by the best test accuracy. Each configuration is repeated over 10 seeds for robustness.

Furthermore, to robustly compare configurations, we compare them pairwise across multiple runs. Configuration `A` outperforms `B` if it performs better for the same dataset, checkpoint, and random seed. We also report the counts of configuration `A` being better more times than some predefined ranking threshold (e.g. 75%) compared to other configurations. This results in a score matrix showing which configurations consistently outperform others. From this matrix, we identify Pareto fronts and assess which influence methods perform better.

## 5.1 [RQ1] VERIFICATION OF ASSUMPTIONS ABOUT CANCELLATION EFFECT

The *cancellation effect* arises when opposing gradient contributions cause small net parameter updates ($\Delta W$) (Eq. 7), while the cancellation metric $C(W)$ remains high. Intuitively, the key issue arising in Eq. 7 is that $C$ is not additive across parameter subsets. For example, although many gradients pass through the classification head (`CL`), frequent-token embeddings in `WE` also accumulate large but opposing updates. Since $C$ aggregates changes via norms, such cancellations are obscured, limiting its reliability as an indicator.

Table 1 presents the cancellation effect measured across LLMs, datasets, and seeds. The embeddings `WE` indeed have the smallest values. The median value of $C$ reflects the *cancellation of an individual parameter*, predominantly observed within a corresponding layer group. Value $C = 1$ denotes the smallest cancellation when all the gradients are codirectional (e.g. majority of `WE` parameters). However,

Table 1: Training gradient cancellations for various LLMs ($1 \rightarrow$ no cancellation, $\infty \rightarrow$ extreme cancellation).

| | Layer | Mean $\pm$ Std | Median | Min | Max | $\rho$ |
|---|---|---|---|---|---|---|
| RoBERTa-Large | WE | $2.2 \pm 0.3$ | 1 | 1 | $\infty$ | -0.3 |
| | 00-05 | $9.4 \pm 3.9$ | 11.8 | 1 | $10^6$ | 0.1 |
| | 06-11 | $10.5 \pm 4.6$ | 14.3 | 1.7 | $10^6$ | 0.1 |
| | 12-17 | $9.4 \pm 5.1$ | 12.5 | 1.6 | $\infty$ | 0.1 |
| | 18-23 | $8.5 \pm 4.4$ | 11.1 | 1.5 | $10^6$ | 0.2 |
| | CL | $8.5 \pm 6.1$ | 11.1 | 1.6 | $\infty$ | 0.1 |
| Llama-3.2 1B | WE | $2.9 \pm 0.3$ | 1 | 1 | $\infty$ | 0.3 |
| | 00-03 | $8.4 \pm 2.9$ | 11.8 | 1.7 | $\infty$ | -0.1 |
| | 04-07 | $5.8 \pm 2.3$ | 7.7 | 1.2 | $10^6$ | -0.2 |
| | 08-11 | $4.4 \pm 1.6$ | 5.8 | 1 | $10^5$ | -0.1 |
| | 12-15 | $4.0 \pm 1.7$ | 5.3 | 1 | $10^6$ | -0.1 |
| | CL | $3.1 \pm 2.3$ | 2.5 | 1 | $10^4$ | -0.1 |
| Mistral 7B | WE | $3.5 \pm 0.3$ | 1.1 | 1 | $\infty$ | 0.1 |
| | 00-07 | $17.7 \pm 3.5$ | 17 | 1.6 | $\infty$ | 0.0 |
| | 08-15 | $16.4 \pm 6.4$ | 18.6 | 2.2 | $\infty$ | 0.1 |
| | 16-23 | $15.6 \pm 8.6$ | 16 | 1.9 | $\infty$ | 0.0 |
| | 24-31 | $15.7 \pm 10.2$ | 15.5 | 1.8 | $\infty$ | 0.0 |
| | CL | $20.5 \pm 19.5$ | 11.7 | 3.8 | $10^5$ | 0.1 |

there are `WE` weights with infinite cancellation scores that have two or more collected gradients from training samples. Clearly, the results confirm that layer group cancellation $C$ "smoothes" the gradient compensation over a large number of parameters and hides those weights where the individual parameter gradient cancellation is high. Yeh et al. (2022) showed that influence estimates from layers with lower $C$ better capture expected prediction changes for ground-truth classes when top-$k$ opponents are filtered out for a given sample. In contrast, we extend this analysis to the validation set as a whole, filtering opponents across many samples and measuring downstream performance via task accuracy. Results demonstrate only weak or no correlation between $C$ and performance (Table 1, column $\rho$: Spearman correlation).

Next, we provide theoretical evidence that reinforces the aforementioned issues with the cancellation effect and demonstrate how it is an unreliable estimator of influence performance (*see Appendix G for the proof*).

**Theorem 5.1 (Cancellation Can Improve Influence Estimation.).** Let $X$ be a training set. Consider:

     1. two samples $\bar{x}_1, \bar{x}_2 \in X$, where $\bar{x}_1$ is noisy and $\bar{x}_2$ is clean;

2. two parameter vectors $\theta$ and $\omega$ such that their cancellation scores satisfy $C(\theta) \ll C(\omega)$ with $C(\omega) \to \infty$ for $\{\bar{x}_1, \bar{x}_2\}$;
3. the TracIn influence scores $I_\theta$ (based on $\theta$ alone) and $I_{\theta,\omega}$ (based jointly on $\theta, \omega$);
4. influence score distance between $\bar{x}_1$ and $\bar{x}_2$ w.r.t. weights $\Theta$ and validation samples $X'$:
$\Delta_\Theta I(\bar{x}_1, \bar{x}_2 \mid X') = |I(\bar{x}_1, X', \Theta) - I(\bar{x}_2, X', \Theta)|$.

Then there exists a validation point $\bar{x}_3$ such that:

$$\Delta I_{\theta,\omega}(\bar{x}_1, \bar{x}_2 \mid \bar{x}_3) > \Delta I_\theta(\bar{x}_1, \bar{x}_2 \mid \bar{x}_3),$$

i.e. the separation between noisy and clean samples is strictly larger under $I_{\theta,\omega}$, showing that contrary to the claim of Yeh et al. (2022), the inclusion of high cancellation weights can *improve* influence.

**Remarks on RQ1 Findings.** Our empirical results demonstrate an absent-to-weak correlation between the cancellation effect and the accuracy of models retrained on datasets filtered by corresponding layers. Further, our theoretical analysis provides a counterexample, demonstrating that weights with high cancellation *can*, in fact, improve influence estimation.

## 5.2 [RQ2] LAYER EFFECTIVENESS IN INFLUENCE FUNCTION ANALYSIS

Next, owing to the limitations of the cancellation effect as previously discussed, we now seek to analyze which layers are better suited for influence estimation in language models. We compare layers based on how well they can detect detrimental samples in accordance with the framework outlined at the beginning of the section. In general, our results are only aligned with (Yeh et al., 2022) regarding the classification head `CL` performance. We observe that `CL` is highly susceptible to influential noise across models and methods. Dataset filtering with scores from `CL` improves downstream accuracy less than other layers (Figure 2).

At the same time, `WE` is not the best choice of layer for every case scenario. While `TracIn` applied on the smaller model RoBERTa-Large demonstrates best accuracies when `WE` is used, in general, the first or second attention layer groups are better options– both in terms of performance and computational complexity, when larger models are used. Methods $\text{TracIn}_{we}^{10}$ and $\text{TracIn}_{we}$ improves upon `TracIn` with `WE`, but requires more clock time in practice (Section 3.1). For other influence functions, we also observe the best performance on the first and second attention layer groups. Appendix C contains details on the layer ranking per model.

Figure 2 demonstrates the best test set accuracy variation (mean and Q1-Q3 quartile range) after filtering with influence scores for Mistral 7B. Configurations have statistically significant differences. The best layers outperform the worst ones (generally, `CL` for most tasks and methods) by 10-15%. Similar observations for other models are in provided in Appendix B due to space constraints.

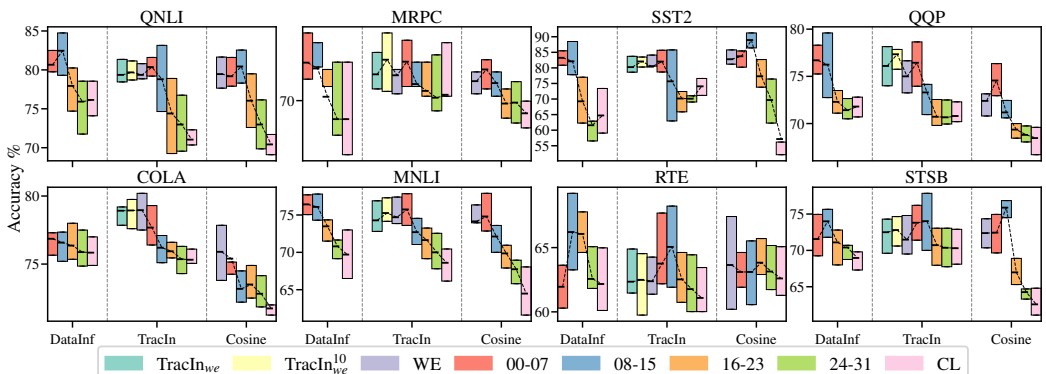

Figure 2: Best test accuracy of Mistral 7B after 30% filtering (averaged over 10 runs). Early and middle attention layers yield the strongest influence method performance on most tasks. Differences are statistically significant (*Friedman p-value: 1e-142*).

**RoBERTa-Large**. For this comparatively smaller model, filtering performance peaks in later attention layers. DataInf achieves the strongest result on group 4 (layers 18–23) with a 0.70 win rate, outperforming other configurations. Cosine on group 3 (layers 12–17, 0.56 win rate) and TracIn on

the embedding layer (0.51) are the closest competitors. The two variants $\text{TracIn}\,we$ and $\text{TracIn}^{10}\,we$ yield nearly identical scores (0.47) but are consistently dominated by TracIn on `WE`. Interestingly, COLA accuracy trends (see Figure 9, Appendix B) reveal that the oracle setting `Full` is occasionally surpassed by influence-based filtering that retains a fraction of noisy samples.

**Llama-3.2 1B**. For Llama, influence functions perform worst among all settings: none of the configurations surpass the uniform removal baseline (`Random`), even when a large fraction of mislabeled samples is discarded. The strongest outcomes are obtained with DataInf on group 2 (layers 04–07, 0.64 win rate) and group 1 (layers 00–03, 0.58), comparable to `Random` (0.64). Accuracy trajectories (Figure 10, Appendix B) further show that `Random` outperforms influence methods on QQP, COLA, and STSB in later epochs. Despite this, noise distribution analysis (Figure 21, Appendix I) reveals non-uniform patterns, with mislabeled samples concentrating at both extremes of the influence range, describing a substantial fraction of detrimental samples as influential for the task goal. We provide additional discussion on Llama results in Appendix M.

**Qwen-2.5 1.5B**. The strongest results are achieved by Cosine on group 2 (layers 07–13) and by DataInf on both groups 1 (layers 00–06) and 2, all ranked top-1. TracIn follows as the runner-up, with its best performance on group 2, while $\text{TracIn}_{we}$ produces comparable outcomes on the same group. Accuracy trends (Figure 8, Appendix B) further show that middle layers (07–13) consistently outperform other layers across GLUE tasks and methods.

**Mistral 7B**. Top-1 configurations constitute DataInf and TracIn. DataInf achieves the strongest results on layers 08–15 (0.71 win rate), 00–07 (0.69), and 16–23 (0.51). TracIn performs best on layers 00–07, closely matched by $\text{TracIn}_{we}^{10}$, which improves over TracIn on `WE`. Cosine on layers 00–07 and 08–15 ranks as runner-up. Accuracy trends (Figure 12, Appendix B) reveal task-specific preferences: Cosine (08–15) excels on SST2 & STSB, DataInf (08–15) on QNLI & RTE, and TracIn on COLA.

**Remarks on RQ2 Findings.** Across models and methods, we observe a consistent pattern in layer-level performance. With mean-score aggregation, the classification head (`CL`) shows limited ability to discriminate noise across all influence methods. DataInf and Cosine frequently outperform TracIn, particularly on early–middle attention layers of deeper models, suggesting that these layers encode representations that are most informative for noise filtering. Comprehensive rankings across all configurations are further provided in Appendix F (Tables 4, 5, 6). Additionally, we verify and reconfirm our findings on autoregressive datasets in Appendix J.

## 5.3 [RQ3] EXPLORING BETTER STRATEGIES TO COMBINE INFLUENCE SCORES

Next, we aim to study whether our proposed aggregation strategies (Section 3.3) can result in better influence function performance. We first analyze the correlation of training-sample influence scores across layers and methods (shown in Appendix E Figure 13), revealing three layer groups: *early, middle*, and *late*. Within a group, layers assign similar scores: for example, TracIn variations on `WE` consistently identify the same noisy samples, and late layers show strong agreement across all methods. Correlations between groups are weak, indicating that layers often *compensate* for one another. Across methods, DataInf closely aligns with TracIn on later layers, while Cosine shows low agreement, providing a complementary view of importance attribution. These patterns motivate the underlying ideas behind our proposed aggregation strategies, `Rank` and `Vote`, that reconcile layer- and method-level discrepancies. Figure 3 illustrates the performance differences relative to mean aggregation for Mistral 7B (results for other models show similar trends and are provided in Appendix H).

Voting substantially improves filtering for TracIn and DataInf on Mistral, with no statistically significant losses across datasets, though CoLA remains the most challenging benchmark. Comparable gains are observed for TracIn on Llama and RoBERTa (Appendix H), with occasional drops on CoLA, QQP. Moreover, performance improvement is observed for NLI tasks, SST-2, and STS-B. For DataInf on Llama, `Rank` slightly outperforms `Vote`, whereas in most other settings `Vote` remains superior.

Utilizing influence functions with `Vote` on Mistral leads to significant improvements and promotes outlier configurations (layer–influence method combinations) to higher positions in the global ranking (Appendix F, Table 6). For instance, TracIn `CL` and $\text{TracIn}_{we}^{10}$ now share the top rank and thus become non-dominant. On RoBERTa, the rank of TracIn `CL` improves from 10 (`Mean`) to 2 (`Vote`). On Mistral, the TracIn `CL` layer rank rises from 12 (`Mean`) to 1 (`Vote`).

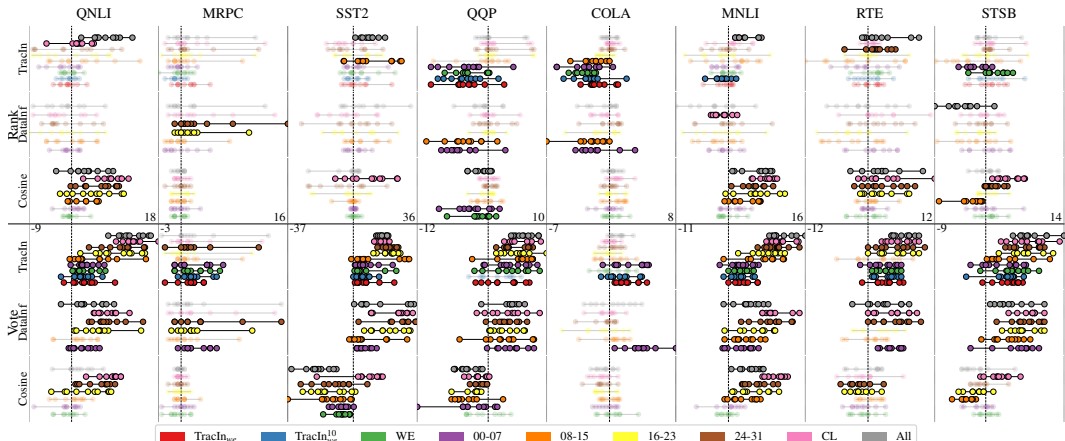

Figure 3: Mistral 7B: Accuracy improvements (% change) of `Rank` and `Vote` relative to mean aggregation, 10 runs per layer group. Voting consistently improves accuracy for DataInf and TracIn across most datasets and layers, while degrading performance for Cosine. Statistically significant differences are shown in opaque colors (Wilcoxon test, $p < 0.1$ and mean change $> 1\%$).

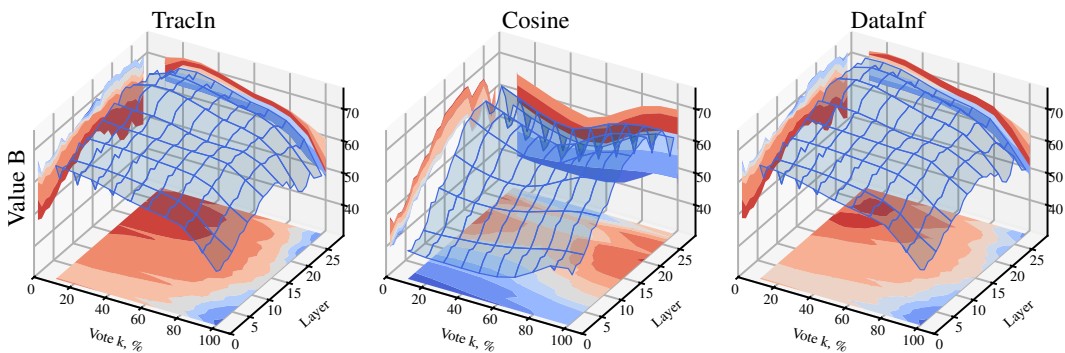

Figure 4: Filtered detrimental samples % with LoRA Value B module for Qwen-2.5 1.5B as a function of the number of votes $k$ (positional voting, Equation 10) and the selected model attention layer. The heatmaps on the axes show where performance is maximized.

The positional voting scheme introduces the hyperparameter $k$ in Equation 10, representing the number of votes each validation sample assigns to training samples. We also study how aggregation performance varies as a function of $k$. Figure 4 shows the percentage of filtered detrimental samples across different values of $k$ and model layers for Qwen-2.5 1.5B with LoRA Value B module, which provides the strongest overall performance. For both DataInf and TracIn, performance is maximized for $k \in [10, 50]$. In contrast, the Cosine-based method attains a local minimum around $k \approx 50$ and reaches its maximum at larger $k$. Similar trends are observed for other LoRA modules (see Appendix L for more details).

**Remarks on RQ3 Findings.** Our results demonstrate that alternative aggregation strategies can significantly improve the effectiveness of influence estimation. In particular, `Vote` consistently outperforms mean aggregation across multiple datasets and models, with notable gains such as an increase in the win rate of DataInf (layers 00–07) to 0.84. These findings establish that aggregation is a critical factor in influence analysis. While `Vote` offers a simple and effective alternative, future work could explore efficient unsupervised or multi-objective aggregation methods $\mathcal{A}$ to exploit the influence signals and further enhance filtering performance.

### 5.4 [RQ4] IDENTIFYING MEASURES TO ESTIMATE DOWNSTREAM PERFORMANCE

We now investigate whether we can utilize an external metric as proxy to evaluate influence function methods, thereby foregoing expensive retraining/fine-tuning based evaluation. To this end, we propose two metrics. The *Noise Detection Rate* (NDR), quantifies the proportion of mislabeled samples

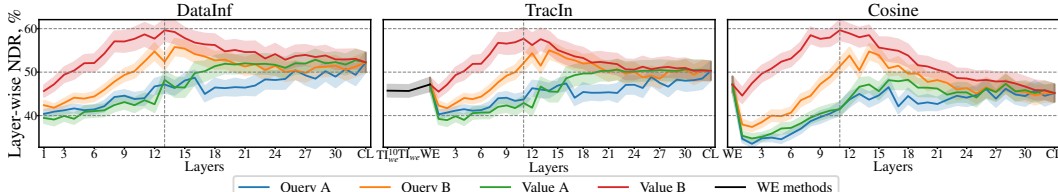

Figure 5: Layer-wise NDR (%) across layers of Mistral 7B.

ranked among the top-$k$% least influential, i.e., *the fraction of noise filtered prior to final fine-tuning.* In contrast, the Area Under the Curve (AUC) metric captures the overall skew of noise across the influence ranking, indicating whether mislabeled samples concentrate at the top of the list or are more uniformly spread. We validate the hypothesis that NDR and AUC can serve as reliable proxy metrics for influence estimation by measuring the metric correlations to the actual ground-truth performance observed after fine-tuning on filtered datasets. Table 2 reports the correlation of NDR and AUC with the actual ground-truth downstream performance. Overall, we observe moderate-to-strong statistically significant positive correlations. NDR at 30% generally exhibits higher values than AUC. However, values in the 0.5–0.7 range indicate that these metrics are not consistently reliable across all configurations. `Vote` aggregation has stronger correlations of 0.8–0.9, suggesting that combining influence scores appropriately can enhance *proxy predictiveness*.

Figure 5 further illustrates the proportion of noise filtered by each layer of Mistral 7B. Across models and influence functions, the Value B LoRA modules consistently capture the largest fraction of noisy samples, with peak filtering observed in the middle attention layers for larger models and later layers for smaller models (see Appendix E for additional results). Additional *sample-influence ranking* variation analysis in Appendix K indicates that average noise influence attains its minimum at layers with the highest noise-detection efficiency.

Table 2: Spearman correlation ($\rho$) of training gradient cancellation ($C$), noise detection rate (NDR), and NDR-AUC with final task accuracy after filtering, across influence aggregation methods. High correlations are shown in bold and * denotes non-statistically significant results.

| | Infl.func | Roberta-Large | | | Llama-3.2 1B | | | Mistral 7B | | |
|---|---|---|---|---|---|---|---|---|---|---|
| | | $C$ | NDR | AUC | $C$ | NDR | AUC | $C$ | NDR | AUC |
| Mean | DataInf | 0.2 | 0.7 | 0.5 | 0.0* | 0.6 | 0.5 | 0.1 | 0.5 | 0.5 |
| | TracIn$_{we}$ | -0.3 | 0.6 | 0.4 | 0.2* | 0.6 | 0.5 | 0.1* | 0.4 | 0.3 |
| | TracIn$_{we}^{10}$ | -0.3 | 0.6 | 0.4 | 0.0* | 0.5 | 0.5 | 0.0* | 0.5 | 0.3 |
| | TracIn | 0.0* | 0.4 | 0.3 | 0.1* | 0.6 | 0.5 | 0.0* | 0.6 | 0.5 |
| | Cosine | 0.2 | 0.7 | 0.6 | -0.0* | 0.5 | 0.5 | -0.1 | 0.6 | 0.5 |
| Rank | DataInf | 0.3 | 0.7 | 0.7 | -0.1* | 0.6 | 0.6 | 0.0* | 0.6 | 0.7 |
| | TracIn$_{we}$ | -0.3 | 0.6 | 0.5 | -0.1* | 0.2 | 0.2* | 0.0* | 0.3 | 0.2 |
| | TracIn$_{we}^{10}$ | -0.3 | 0.5 | 0.5 | -0.2* | 0.1* | 0.0* | -0.1* | 0.2* | 0.1* |
| | TracIn | 0.2 | 0.5 | 0.5 | -0.1* | 0.5 | 0.5 | -0.1* | 0.4 | 0.5 |
| | Cosine | 0.2 | 0.7 | 0.7 | -0.1* | 0.6 | 0.6 | 0.1* | 0.6 | 0.6 |
| Vote | DataInf | 0.3 | **0.8** | **0.8** | -0.1 | 0.6 | 0.6 | 0.1 | **0.9** | **0.8** |
| | TracIn$_{we}$ | -0.4 | **0.8** | **0.8** | -0.1* | 0.5 | 0.5 | -0.1* | **0.8** | **0.8** |
| | TracIn$_{we}^{10}$ | -0.4 | 0.7 | **0.8** | -0.1* | 0.5 | 0.5 | 0.0* | **0.8** | **0.8** |
| | TracIn | 0.2 | **0.8** | 0.7 | -0.1 | 0.6 | 0.6 | 0.1 | **0.8** | **0.8** |
| | Cosine | 0.2 | 0.7 | 0.6 | -0.1 | 0.2 | 0.2 | 0.0* | 0.4 | 0.4 |

**Remarks on RQ4 Findings.** The cancellation effect shows little correlation with downstream performance and fails to reliably indicate influential layers or weights. In contrast, NDR serves to be a useful proxy, particularly under `Vote` aggregation, though correlations are generally moderate-to-high across layers, tasks, and models. The proposed AUC metric exhibits similar trends. Together, these results suggest that proxy metrics incorporating both the quantity and quality of filtered anomalies could offer more reliable indicators for influence function evaluation, highlighting a promising direction for future research.

# 6 CONCLUSION

Our paper conducts multi-faceted analysis to better understand how influence functions can be effectively leveraged for data-centric learning in large language models. Firstly, our work challenges prior assumptions in past work about the most informative layers for influence estimation in LLMs and demonstrates that middle attention layers often provide more reliable signals than the *first* embedding layers. Secondly, we introduce improved layer-wise influence score aggregation strategies that move beyond the standard averaging approach, and attain significant performance improvements in influence analysis. Finally, we propose external evaluation metrics, such as the Noise Detection Rate (NDR), to offer a more robust framework for evaluating and interpreting influence scores without undertaking costly retraining for influence function evaluation. These findings advance the understanding of training data influence in large language models and provide practical tools for more accurate and efficient model auditing.

## 7 REPRODUCIBILITY STATEMENT

We have open-sourced our code and implementation for our experiments: https://github.com/dvitel/nn-infl. To support reproducibility, multiple runs were executed with fixed random seeds specified in the accompanying shell scripts, and determinism was enabled in the underlying *pytorch* framework. All experiments were conducted on an Ubuntu server with 8x NVIDIA A100s (40GB VRAM/GPU).

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

APPENDIX

## A  EXPERIMENTAL SETUP DETAILS

This section describes the five-stage experimental pipeline from Section 5 in detail.

**Stage 1.** We define the anomaly detection task on GLUE datasets with injected label noise. For influence computation, we select up to 4500 training samples, 500 validation samples, and 500 test samples for reporting downstream performance, with 10 random seeds controlling different selections. For binary classification tasks, 20% of training labels are flipped uniformly at random. MNLI is binarized by keeping only entailment (0) and neutral (1) classes, and STSB scores 0–5 are converted to 0/1 using threshold 3; noise is added similarly.

Checkpoints preparation adds LoRA modules (Dettmers et al., 2023) to query and value projections in every attention layer, compresses embeddings to the tokens participating in the GLUE dataset splits. The embedding compression greatly speeds up influence computations based on gradients.

RoBERTa-Large has 355M parameters (1.5M trainable), 24 attention layers, and a classification head with dense projection. Llama-3.2 1B is multilingual, with 16 attention layers, attached LoRA modules, and a tunable classification head (0.5M trainable parameters); embedding compression reduces parameters for influence estimation. Qwen-2.5 1.5B has 28 attention layers, and Mistral 7B has 32. Word embeddings (WE) remain frozen during training, but gradients are computed for influence estimation. Learning rates are: Roberta 3e-4, Llama 1e-4, Qwen 3e-4 (1e-3 for RTE), and Mistral 5e-5; all models use 16-bit float weights.

**Stage 2**. Models are fine-tuned on noisy datasets for 10 epochs from the initial checkpoint. Validation loss and accuracy are tracked to select the checkpoint for influence estimation. Following (Pruthi et al., 2020) and preliminary experiments, we use the checkpoint with the lowest validation loss in the following stages.

**Stage 3**. We split the neural network layers into groups: embeddings WE, classification head CL, and 4 groups with an equal number of internal attention layers. The groups in the results are numbered from 1 to 4. For instance, group 1 of RoBERTa-Large includes the first to sixth attention layers, also denoted 00-05. Group 2 has 06-11, group 3 – layers 12-17, and group 4 – 18-23. Similar grouping is done for other models.

**Stage 4**. The scoring routine computes attributed values from the collected influence tensors of layer groups for a particular aggregation method under consideration (default is averaging). This stage also estimates other indicator metrics such as noise detection rate (based on knowledge of the flipped label), noise histograms, and cancellation effect.

**Stage 5**. Training samples are ordered by their influence scores, and the lowest 30% are removed. The filtered datasets are fine-tuned using the same hyperparameters as Stage 2, starting from the Stage 1 checkpoint. Method performance is evaluated via test-set accuracy on samples unseen during training or influence computation.

We apply the following techniques to optimize influence computation.

**Compression**. To reduce computation, we replace the full embedding layer with weights corresponding only to tokens present in the task dataset. While sub-views may be preferable in production, this approach is effective in benchmarking, especially for multilingual models with many unused tokens.

**Batching**. Equations 3,4,5 depend on dot-product $\nabla l_{\bar{x}'}^T \nabla l_{\bar{x}}$. To maintain the gradients for $n$ training samples, $k$ validation samples, and $m$ parameters, TracIn and Cosine requires $O(nkm)$, DataInf $- O(nkm + n^2 m)$ of memory. In one iteration, we pick $n_1$ training and $k_1$ validation samples s.t. $O(n_1 k_1 m)$ gradients fit the available GPU memory, and compute $O(n_1 k_1)$ influence scores between sample pairs. Staying in $O(n_1 k_1 m)$ limit requires the iteration through all pair of $\lceil n/n_1 \rceil$ training and $\lceil k/k_1 \rceil$ validation batches.

Inner loop has to recompute the gradients because of the memory limit. For A100 80GB GPU, 4500 training and 500 validation samples, influence values on Mistral embeddings are computed in 20 batches in our experiments. It is still very expensive for DataInf because it also depends on $\nabla l_{\bar{z}}^T \nabla l_{\bar{x}}$.

# B BEST TEST SET ACCURACY DISTRIBUTIONS AND TRENDS

This section presents the best test set accuracy distribution after filtering for RoBERTa (Figure 6), Llama (Figure 7), and Qwen (Figure 8). TracIn indeed has decreasing performance trend from early WE to later CL layers across models. However, DataInf and Cosine frequently outperform TracIn on early or middle attention layers for bigger models, and later attention layers for the smaller Roberta.

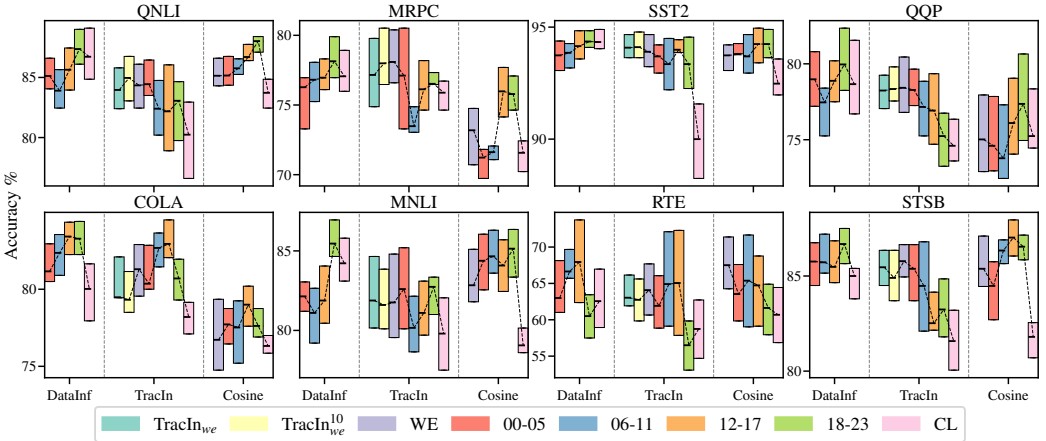

Figure 6: Roberta-Large best test set accuracy after 30% filtering over 10 runs.

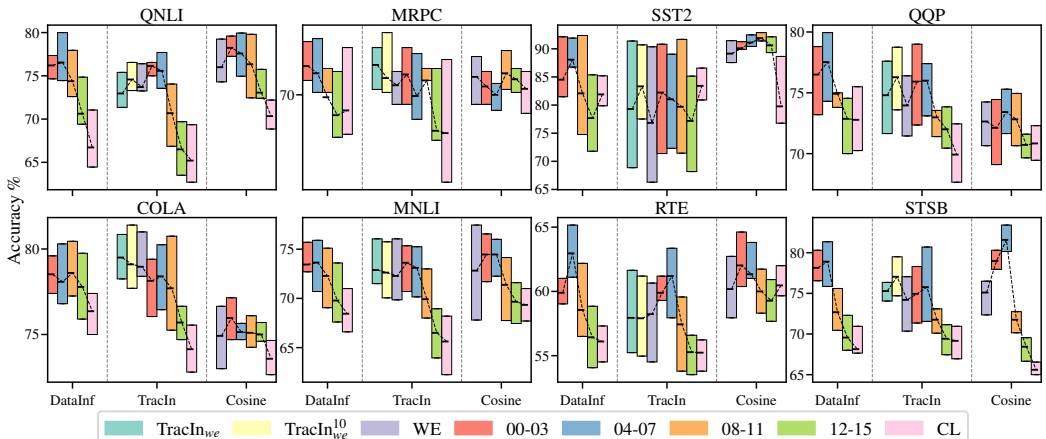

Figure 7: Llama-3.2 1B best test set accuracy after 30% filtering over 10 runs.

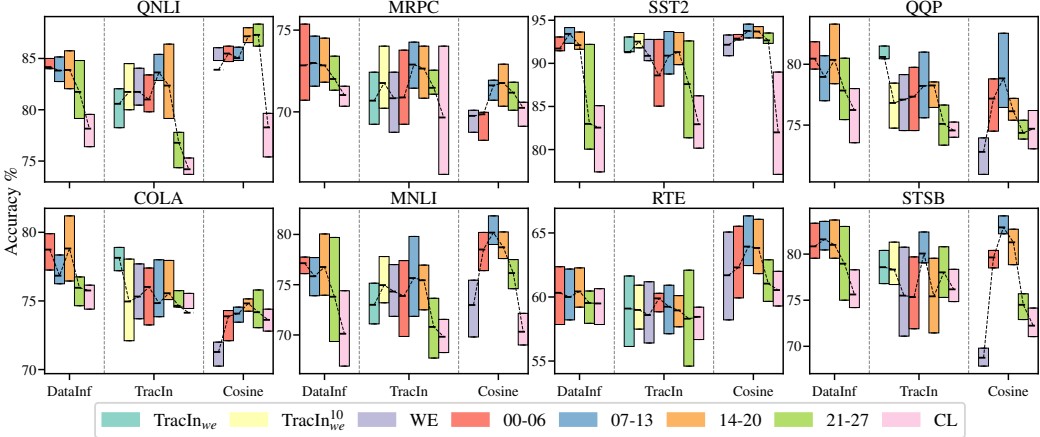

Figure 8: Qwen-2.5 1.5B best test set accuracy after 30% filtering over 10 runs.

The following trends are presented for the best configurations, influence method, and layer group for Roberta (Figure 9), Llama (Figure 10), Qwen (Figure 11), and Mistral (Figure 10). Accuracy trends are measured on the test set, not used for training or influence computation.

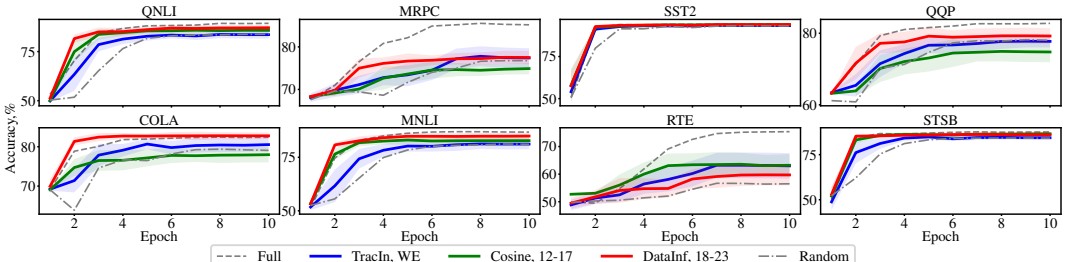

Figure 9: Roberta-Large accuracy trend on test set after 30% filtering.

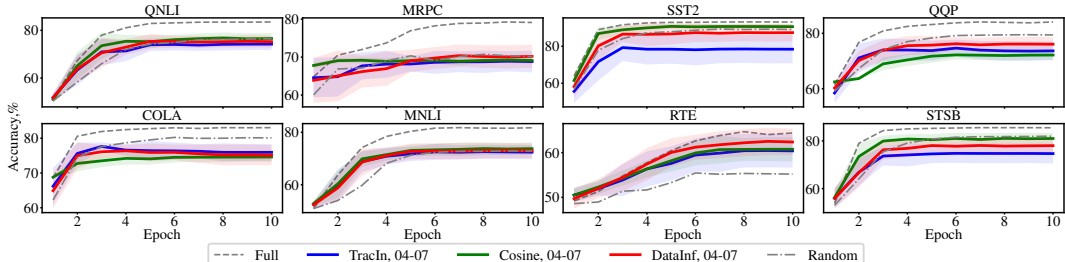

Figure 10: Llama-3.2 1B accuracy trend on test set after 30% filtering.

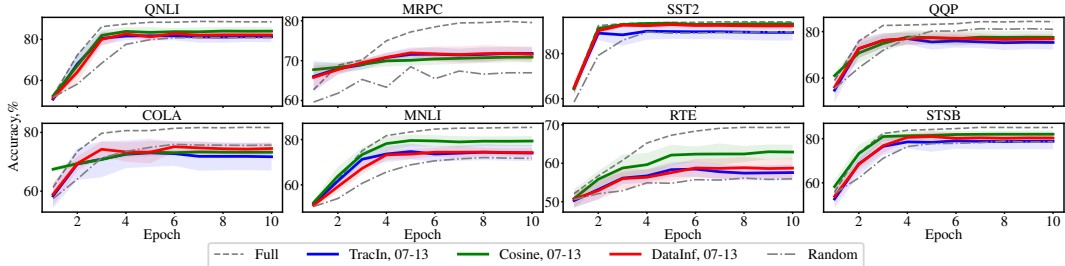

Figure 11: Qwen-2.5 1.5B accuracy trend on test set after 30% filtering.

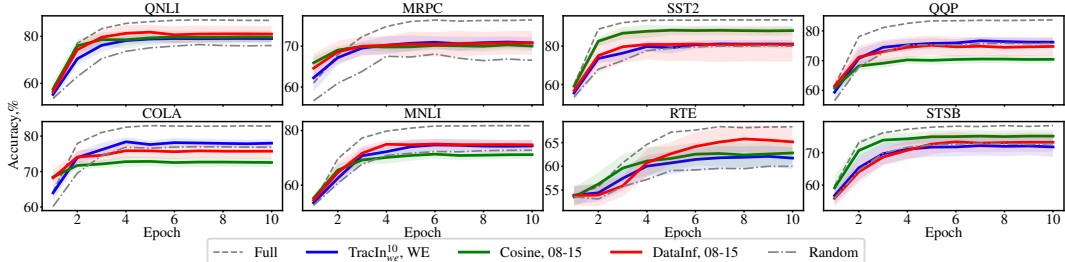

Figure 12: Mistral 7B accuracy trend on test set after 30% filtering.

# C LAYER RANKING PER MODEL

Table 3 presents the results of layer performance ranking based on the measured best test set accuracy after sample filtering, where 30% of the least influential samples according to each layer are discarded and subsequent retraining is performed. In the table, the model weight groups are presented as rows ordered starting from $\text{TracIn}_{we}^{10}$, $\text{TracIn}_{we}$ (i.e., selection from WE), followed by the full WE layer, attention layer groups, and the CL layer. Attention groups are shown in the same order as they appear in the model, but their size varies with the model.

Table 3: Layer ranks (embed WE, attn 1–4, head CL) and win rates (in parentheses) across models and influence functions (threshold=0.5). '–' denotes N/A. More specifically, DataInf is infeasible to run on WE and TracIn specific WE variations TI. Most performant layers are shown in bold.

| Layers | Roberta-Large | | | Llama-3.2 1B | | | Qwen-2.5 1.5B | | | Mistral 7B | | |
| --- | --- | --- | --- | --- | --- | --- | --- | --- | --- | --- | --- | --- |
| | DataInf | TracIn | Cosine | DataInf | TracIn | Cosine | DataInf | TracIn | Cosine | DataInf | TracIn | Cosine |
| $\text{TI}_{we}^{10}$ | - | 1 (.49) | - | - | **1 (.56)** | - | - | 2 (.49) | - | - | 1 (.58) | - |
| $\text{TI}_{we}$ | - | 2 (.48) | - | - | 2 (.52) | - | - | 1 (.51) | - | - | 2 (.50) | - |
| WE | - | **1 (.54)** | 2 (.41) | - | 3 (.44) | 3 (.42) | - | 3 (.40) | 3 (.19) | - | 2 (.50) | 1 (.58) |
| 1 | 4 (.29) | 3 (.50) | 3 (.35) | 1 (.53) | 2 (.54) | **1 (.53)** | **1 (.49)** | 2 (.42) | 2 (.45) | 2 (.53) | **1 (.61)** | **1 (.60)** |
| 2 | 4 (.30) | 5 (.40) | 2 (.40) | **1 (.58)** | 1 (.54) | **1 (.53)** | 1 (.48) | **1 (.57)** | 1 (.60) | **1 (.60)** | 2 (.48) | 2 (.55) |
| 3 | 2 (.40) | 4 (.44) | **1 (.56)** | 2 (.42) | 4 (.38) | 2 (.45) | **1 (.49)** | 2 (.48) | **1 (.61)** | 2 (.39) | 3 (.29) | 3 (.37) |
| 4 | **1 (.55)** | 6 (.36) | 1 (.54) | 3 (.23) | 5 (.22) | 4 (.28) | 2 (.31) | 4 (.30) | 2 (.40) | 3 (.20) | 4 (.20) | 4 (.23) |
| CL | 3 (.36) | 7 (.17) | 4 (.15) | 4 (.18) | 6 (.17) | 5 (.15) | 3 (.16) | 5 (.22) | 3 (.18) | 3 (.20) | 4 (.20) | 5 (.09) |

Here, the ranking is given across layers. In many cases, first or second attention groups happen to be most performant, which makes them a good selection for detrimental sample detection.

# D INFLUENCE SCORE CORRELATIONS BETWEEN LAYERS AND METHODS

Figure 13 presents computed correlations between training-sample attributed scores of different layers and influence methods. TracIn splits the model into three groups of *early, middle, and later* layers based on their score agreement, i.e., the strength of correlation. DataInf has high correlation with TracInf on later layers such as CL. This layer split suggests that different parameters could capture different detrimental samples, and a good aggregation of scores is viable.

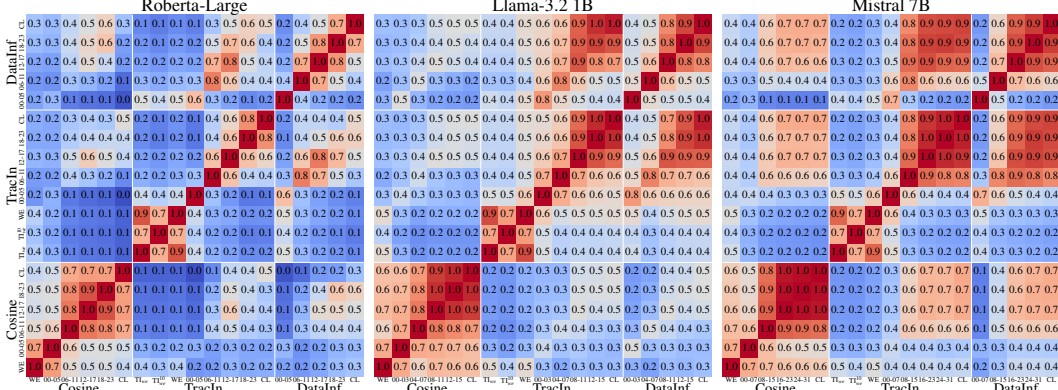

Figure 13: Influence scores correlation for 3 models, methods, and layers.

# E DETECTED NOISE IN 30% LEAST INFLUENTIAL SAMPLES

The following results present the noise detection across layers for Roberta-Large (Figure 14), Llama-3.2 1B (Figure 15), Qwen-2.5 1.5B (Figure 16), Mistral 7B (Figure 17). The consequent conclusions are (1) LoRA Value B is the best in capturing noise across models; (2) layers with high NDR also have high downstream performance; (3) spike of NDR moves from later attention layers for shallow models to the early-middle layers for deeper models.

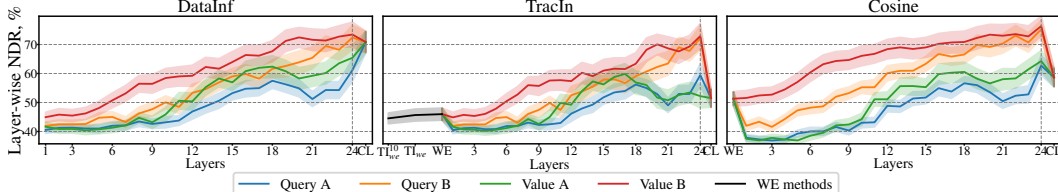

Figure 14: Roberta-Large NDR for 30% threshold across modules and layers.

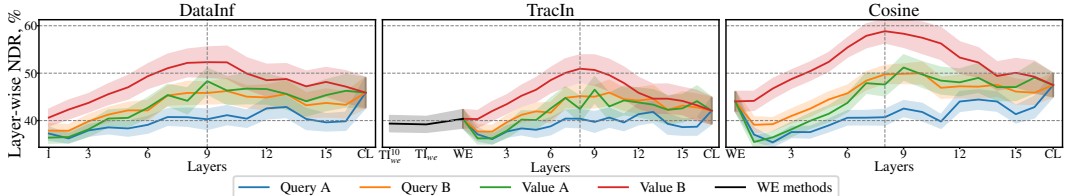

Figure 15: Llama-3.2 1B NDR for 30% threshold across modules and layers.

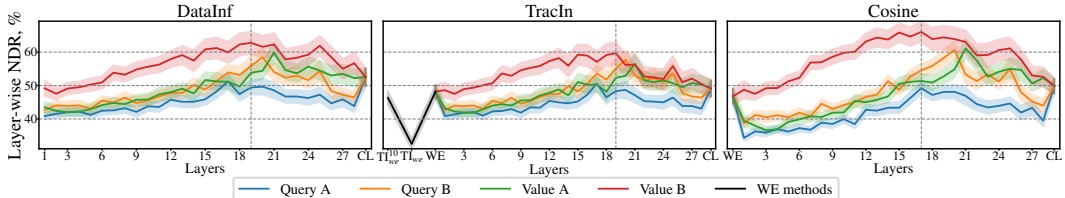

Figure 16: Qwen-2.5 1.5B NDR for 30% threshold across modules and layers.

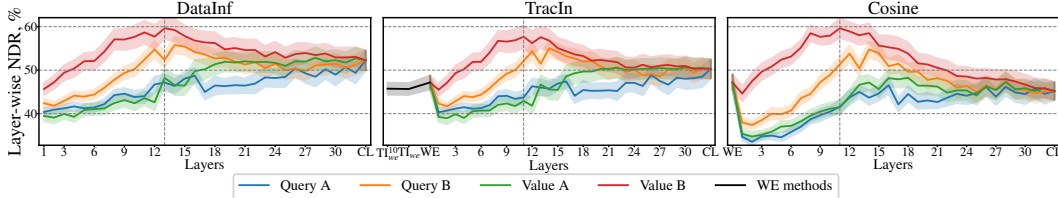

Figure 17: Mistral 7B NDR for 30% threshold across modules and layers.

Next, from the following result we can conclude that Value B on network middle layers captures the highest noise levels and is a good choice for influence estimation.

## F  BEST TEST SET ACCURACY AFTER FILTERING

The following tables 4,5,6 present the ranking of filtering methods based on different influence scoring and aggregation for 3 models under consideration.

Table 4: Roberta-Large best **test set accuracy** on GLUE after 30% filtering with influence scores. Methods are ordered from best to worst with average ranking across 10 runs per setup. The highest accuracy means are highlighted.

| Method | Agg | Layer | Rank | Win Rate | QNLI | MRPC | SST2 | QQP | COLA | MNLI | RTE | STSB |
|---|---|---|---|---|---|---|---|---|---|---|---|---|
| Full | | | 0 | 0.92 ± 0.12 | 89.5 ± 1.2 | 86.0 ± 2.5 | 94.7 ± 0.8 | 83.2 ± 1.5 | 83.0 ± 1.2 | 87.3 ± 0.8 | 76.1 ± 3.0 | 88.0 ± 1.6 |
| Cosine | Rank | 18-23 | 1 | 0.66 ± 0.15 | 87.9 ± 1.9 | 77.0 ± 2.6 | 94.2 ± 0.9 | 79.9 ± 4.9 | 79.6 ± 1.3 | 86.3 ± 2.6 | 62.5 ± 7.0 | 86.9 ± 1.3 |
| DataInf | Mean | 18-23 | 1 | 0.66 ± 0.16 | 87.4 ± 2.1 | 78.1 ± 2.1 | 94.4 ± 0.6 | 80.0 ± 3.1 | 83.3 ± 1.2 | 85.5 ± 2.2 | 60.5 ± 5.6 | 86.7 ± 1.5 |
| DataInf | Mean | all | 1 | 0.66 ± 0.16 | 86.6 ± 2.5 | 77.7 ± 1.3 | 94.3 ± 0.9 | 79.5 ± 2.8 | **83.5 ± 1.2** | 84.1 ± 2.0 | 67.3 ± 6.5 | 86.6 ± 1.7 |
| DataInf | Vote | 18-23 | 1 | 0.65 ± 0.16 | 87.7 ± 1.9 | 77.5 ± 1.8 | 94.1 ± 0.9 | **80.1 ± 4.6** | 79.1 ± 2.2 | 86.1 ± 2.1 | 62.7 ± 5.6 | 86.7 ± 1.4 |
| DataInf | Vote | all | 1 | 0.65 ± 0.16 | 87.6 ± 1.9 | 77.5 ± 1.6 | 94.2 ± 0.9 | 79.8 ± 3.3 | 77.7 ± 1.7 | 86.1 ± 2.5 | 63.6 ± 5.5 | 86.9 ± 1.4 |
| TracIn | Vote | 00-05 | 1 | 0.61 ± 0.16 | 86.4 ± 1.9 | **79.9 ± 2.1** | 94.3 ± 0.8 | 78.6 ± 4.1 | 76.6 ± 2.7 | 86.0 ± 2.1 | 63.5 ± 6.0 | 86.9 ± 1.3 |
| DataInf | Vote | 00-05 | 1 | 0.59 ± 0.16 | 87.0 ± 2.1 | 79.6 ± 2.6 | 94.2 ± 0.8 | 78.6 ± 4.1 | 77.1 ± 1.6 | 85.1 ± 2.8 | 63.6 ± 6.0 | 86.7 ± 1.3 |
| TracIn$_{we}$ | Vote | WE | 1 | 0.58 ± 0.17 | 86.9 ± 1.9 | 78.9 ± 2.2 | 94.1 ± 0.8 | 78.4 ± 3.8 | 77.2 ± 2.1 | 85.4 ± 2.1 | 64.4 ± 5.7 | 86.4 ± 1.2 |
| Cosine | Rank | 12-17 | 1 | 0.55 ± 0.18 | 87.2 ± 2.4 | 75.7 ± 1.9 | 94.0 ± 0.8 | 77.1 ± 4.3 | 78.6 ± 2.1 | 85.9 ± 2.2 | 63.8 ± 7.1 | 86.5 ± 1.6 |
| TracIn | Vote | 18-23 | 2 | 0.62 ± 0.16 | 87.8 ± 2.1 | 77.7 ± 2.3 | 94.0 ± 0.7 | 80.0 ± 4.0 | 78.1 ± 1.3 | 86.0 ± 1.7 | 61.5 ± 6.4 | 86.8 ± 1.5 |
| TracIn | Vote | all | 2 | 0.61 ± 0.17 | 87.5 ± 2.1 | 77.6 ± 2.2 | 94.2 ± 0.7 | 79.1 ± 4.7 | 77.2 ± 2.4 | 85.8 ± 2.0 | 62.7 ± 5.6 | 86.9 ± 1.2 |
| TracIn | Vote | 06-11 | 2 | 0.58 ± 0.15 | 87.0 ± 2.1 | 78.8 ± 3.0 | 94.1 ± 1.0 | 78.9 ± 4.2 | 76.7 ± 2.3 | 85.3 ± 2.6 | 62.9 ± 6.2 | **87.0 ± 1.3** |
| TracIn | Vote | 12-17 | 2 | 0.58 ± 0.18 | 87.4 ± 1.9 | 76.4 ± 1.6 | 94.0 ± 1.0 | 78.2 ± 3.9 | 77.7 ± 1.8 | **86.4 ± 2.0** | 62.1 ± 5.6 | 86.6 ± 1.5 |
| Cosine | Vote | 18-23 | 2 | 0.57 ± 0.17 | 87.9 ± 1.8 | 76.0 ± 2.0 | 94.1 ± 0.8 | 79.0 ± 5.4 | 78.3 ± 1.7 | 85.7 ± 2.2 | 61.9 ± 7.0 | 86.6 ± 1.6 |
| DataInf | Vote | 12-17 | 2 | 0.56 ± 0.18 | 86.9 ± 2.0 | 76.7 ± 2.3 | 94.0 ± 0.9 | 78.1 ± 4.5 | 77.9 ± 2.5 | 86.2 ± 2.2 | 64.0 ± 5.7 | 86.5 ± 1.5 |
| Cosine | Rank | all | 2 | 0.56 ± 0.17 | 87.5 ± 2.0 | 75.2 ± 2.1 | 94.2 ± 0.9 | 77.8 ± 5.3 | 77.7 ± 1.5 | 86.0 ± 2.1 | 63.7 ± 7.1 | 86.5 ± 1.5 |
| TracIn | Vote | CL | 2 | 0.55 ± 0.16 | 87.9 ± 1.7 | 75.6 ± 1.8 | 94.2 ± 0.9 | 78.1 ± 3.3 | 77.0 ± 2.4 | 86.1 ± 2.2 | 62.5 ± 6.0 | 86.0 ± 1.0 |
| DataInf | Mean | 12-17 | 2 | 0.55 ± 0.16 | 85.6 ± 2.5 | 77.0 ± 2.7 | 94.2 ± 0.9 | 78.9 ± 2.0 | 83.4 ± 1.3 | 81.9 ± 2.8 | **67.9 ± 8.1** | 85.5 ± 2.2 |
| TracIn | Vote | WE | 2 | 0.53 ± 0.17 | 86.5 ± 1.4 | 78.1 ± 1.5 | 94.2 ± 0.8 | 78.2 ± 4.1 | 76.6 ± 1.6 | 85.3 ± 1.6 | 63.1 ± 6.3 | 86.6 ± 1.6 |
| TracIn$_{we}^{10}$ | Vote | WE | 3 | 0.59 ± 0.16 | 86.9 ± 2.1 | 78.8 ± 1.9 | 94.2 ± 0.8 | 78.5 ± 4.3 | 76.1 ± 2.6 | 85.9 ± 2.2 | 63.4 ± 5.5 | 86.7 ± 0.8 |
| DataInf | Vote | 06-11 | 3 | 0.55 ± 0.16 | 87.0 ± 2.3 | 78.0 ± 3.4 | 94.0 ± 0.9 | 79.0 ± 3.3 | 76.5 ± 1.9 | 85.5 ± 2.2 | 62.3 ± 6.5 | 86.9 ± 1.5 |
| DataInf | Rank | 18-23 | 3 | 0.53 ± 0.18 | 87.3 ± 1.9 | 76.5 ± 1.4 | 93.8 ± 1.1 | 79.7 ± 4.1 | 77.0 ± 1.7 | 85.5 ± 2.5 | 62.1 ± 5.6 | 86.4 ± 1.4 |
| DataInf | Rank | all | 3 | 0.52 ± 0.18 | 87.4 ± 2.4 | 76.4 ± 1.9 | 94.4 ± 0.9 | 77.0 ± 4.7 | 76.7 ± 1.6 | 85.4 ± 1.9 | 63.2 ± 5.8 | 85.9 ± 1.2 |
| Cosine | Mean | all | 3 | 0.52 ± 0.16 | 87.8 ± 2.1 | 75.3 ± 1.7 | 94.0 ± 0.6 | 75.2 ± 2.9 | 77.0 ± 2.1 | 85.5 ± 2.2 | 63.6 ± 6.3 | 86.8 ± 1.3 |
| TracIn | Mean | WE | 3 | 0.48 ± 0.17 | 84.3 ± 2.7 | 78.1 ± 3.3 | 93.9 ± 1.0 | 78.4 ± 2.5 | 81.3 ± 2.0 | 81.8 ± 3.6 | 64.1 ± 6.5 | 85.8 ± 1.3 |
| DataInf | Rank | 12-17 | 3 | 0.48 ± 0.19 | 86.2 ± 2.6 | 75.9 ± 1.8 | 94.5 ± 0.9 | 76.1 ± 3.2 | 77.4 ± 1.8 | 85.0 ± 2.2 | 63.2 ± 6.0 | 85.6 ± 2.0 |
| DataInf | Mean | 00-05 | 3 | 0.47 ± 0.17 | 85.1 ± 3.0 | 76.3 ± 3.4 | 93.7 ± 0.8 | 79.0 ± 3.0 | 81.2 ± 2.4 | 82.1 ± 2.2 | 63.0 ± 8.1 | 85.8 ± 1.6 |
| Cosine | Mean | 12-17 | 4 | 0.55 ± 0.16 | 86.7 ± 2.2 | 76.0 ± 2.8 | 94.2 ± 1.0 | 76.1 ± 3.6 | 79.0 ± 2.4 | 84.1 ± 2.2 | 64.7 ± 6.2 | **87.0 ± 1.5** |
| Cosine | Mean | 18-23 | 4 | 0.54 ± 0.16 | **88.0 ± 1.9** | 75.8 ± 2.7 | 94.2 ± 1.0 | 77.4 ± 3.5 | 77.6 ± 1.6 | 85.1 ± 2.4 | 61.6 ± 5.4 | 86.5 ± 1.1 |
| DataInf | Vote | CL | 4 | 0.52 ± 0.17 | 87.8 ± 1.6 | 76.3 ± 2.5 | 94.0 ± 0.8 | 78.5 ± 3.5 | 77.0 ± 1.7 | 85.8 ± 2.1 | 61.7 ± 5.4 | 86.2 ± 1.3 |
| DataInf | Mean | CL | 4 | 0.52 ± 0.16 | 86.7 ± 2.9 | 77.1 ± 2.8 | 94.3 ± 0.4 | 78.7 ± 3.9 | 80.0 ± 2.7 | 84.2 ± 2.5 | 62.6 ± 6.1 | 85.0 ± 1.4 |
| TracIn | Rank | 18-23 | 4 | 0.51 ± 0.16 | 87.3 ± 1.6 | 76.6 ± 2.4 | 94.2 ± 1.3 | 78.0 ± 3.4 | 76.9 ± 1.2 | 85.3 ± 2.3 | 61.0 ± 6.0 | 86.4 ± 1.0 |
| TracIn | Mean | 00-05 | 4 | 0.45 ± 0.15 | 84.4 ± 3.4 | 77.1 ± 4.2 | 93.7 ± 1.0 | 78.3 ± 2.6 | 80.4 ± 4.2 | 82.6 ± 2.6 | 61.9 ± 6.3 | 85.4 ± 2.2 |
| TracIn | Rank | all | 5 | 0.49 ± 0.17 | 87.1 ± 2.0 | 76.4 ± 1.9 | **94.6 ± 1.0** | 75.1 ± 3.2 | 76.6 ± 1.2 | 85.4 ± 2.2 | 62.9 ± 5.5 | 86.0 ± 1.5 |
| DataInf | Mean | 06-11 | 5 | 0.49 ± 0.15 | 83.9 ± 3.2 | 76.8 ± 2.7 | 93.9 ± 0.9 | 77.5 ± 2.9 | 82.4 ± 1.4 | 81.1 ± 2.5 | 66.6 ± 6.8 | 85.7 ± 2.4 |
| TracIn$_{we}^{10}$ | Mean | WE | 5 | 0.46 ± 0.15 | 85.0 ± 2.4 | 78.0 ± 3.8 | 94.1 ± 0.7 | 78.3 ± 2.7 | 79.3 ± 3.5 | 81.6 ± 2.7 | 62.7 ± 4.6 | 84.9 ± 2.0 |
| TracIn$_{we}$ | Mean | WE | 5 | 0.45 ± 0.15 | 84.0 ± 2.5 | 77.2 ± 3.3 | 94.1 ± 0.8 | 78.2 ± 2.4 | 79.5 ± 4.2 | 81.9 ± 3.8 | 63.0 ± 5.0 | 85.5 ± 2.1 |
| Cosine | Vote | all | 6 | 0.52 ± 0.15 | 87.6 ± 2.0 | 75.2 ± 2.4 | 94.2 ± 0.9 | 75.8 ± 4.9 | 76.5 ± 1.5 | 86.2 ± 1.9 | 62.7 ± 7.2 | 86.8 ± 1.1 |
| Cosine | Vote | 12-17 | 6 | 0.47 ± 0.17 | 86.3 ± 2.5 | 75.3 ± 2.1 | 94.2 ± 0.9 | 74.5 ± 4.3 | 77.9 ± 2.0 | 86.0 ± 2.3 | 62.4 ± 5.9 | 86.5 ± 1.5 |
| TracIn | Rank | 12-17 | 6 | 0.42 ± 0.17 | 81.7 ± 11.2 | 76.1 ± 2.0 | 94.2 ± 1.0 | 76.0 ± 3.8 | 77.6 ± 1.8 | 84.7 ± 2.3 | 63.3 ± 5.7 | 84.7 ± 2.3 |
| Random | | | 6 | 0.40 ± 0.15 | 84.3 ± 1.9 | 77.2 ± 2.1 | 93.6 ± 1.1 | 75.8 ± 4.1 | 76.6 ± 1.6 | 80.4 ± 2.0 | 57.8 ± 2.6 | 85.2 ± 2.0 |
| TracIn | Mean | 12-17 | 7 | 0.40 ± 0.13 | 82.2 ± 4.4 | 76.1 ± 3.0 | 94.0 ± 0.7 | 76.9 ± 3.2 | 82.9 ± 2.1 | 81.1 ± 2.7 | 65.1 ± 8.3 | 82.5 ± 4.0 |
| TracIn | Rank | WE | 7 | 0.39 ± 0.16 | 85.7 ± 2.1 | 72.1 ± 2.6 | 94.1 ± 0.8 | 75.8 ± 4.8 | 77.1 ± 3.5 | 83.8 ± 1.5 | 63.4 ± 5.3 | 85.5 ± 1.7 |
| Cosine | Mean | WE | 7 | 0.39 ± 0.15 | 85.1 ± 2.3 | 73.2 ± 2.8 | 93.7 ± 0.9 | 75.0 ± 5.1 | 76.7 ± 4.5 | 82.8 ± 3.4 | 67.5 ± 4.2 | 85.4 ± 2.0 |
| Cosine | Rank | CL | 8 | 0.45 ± 0.16 | 87.5 ± 1.7 | 71.3 ± 1.0 | 94.4 ± 1.0 | 77.5 ± 5.3 | 76.6 ± 1.8 | 85.3 ± 2.2 | 60.1 ± 5.4 | 86.1 ± 0.8 |
| TracIn$_{we}^{10}$ | Rank | WE | 8 | 0.37 ± 0.15 | 85.7 ± 1.7 | 70.9 ± 1.1 | 94.1 ± 0.9 | 75.1 ± 5.1 | 77.7 ± 2.4 | 83.9 ± 1.5 | 63.1 ± 5.3 | 85.3 ± 1.7 |
| DataInf | Rank | 06-11 | 8 | 0.36 ± 0.15 | 86.2 ± 2.2 | 73.6 ± 2.5 | 94.2 ± 1.2 | 73.8 ± 4.5 | 75.9 ± 3.2 | 83.1 ± 1.6 | 63.6 ± 7.3 | 85.6 ± 1.3 |
| Cosine | Rank | 06-11 | 9 | 0.39 ± 0.15 | 86.6 ± 1.7 | 71.9 ± 2.1 | 94.0 ± 1.2 | 74.6 ± 4.8 | 76.7 ± 2.0 | 84.2 ± 2.1 | 64.1 ± 7.2 | 86.2 ± 1.5 |
| TracIn | Rank | 06-11 | 9 | 0.37 ± 0.14 | 86.0 ± 2.0 | 73.9 ± 3.6 | 94.2 ± 1.1 | 73.8 ± 4.1 | 76.2 ± 1.3 | 83.4 ± 1.9 | 63.5 ± 7.9 | 85.9 ± 1.7 |
| Cosine | Mean | 06-11 | 10 | 0.40 ± 0.14 | 85.8 ± 1.4 | 71.6 ± 1.1 | 93.7 ± 1.2 | 73.8 ± 5.4 | 77.5 ± 3.1 | 84.7 ± 1.8 | 65.4 ± 7.3 | 86.3 ± 1.6 |
| TracIn | Mean | 06-11 | 10 | 0.38 ± 0.13 | 82.4 ± 3.6 | 73.5 ± 2.0 | 93.3 ± 1.5 | 77.2 ± 3.6 | 82.7 ± 1.4 | 80.2 ± 3.2 | 64.9 ± 8.9 | 84.5 ± 2.9 |
| DataInf | Mean | 00-05 | 10 | 0.37 ± 0.14 | 85.5 ± 2.3 | 70.6 ± 1.3 | 94.3 ± 0.8 | 74.6 ± 4.7 | 77.5 ± 1.6 | 84.0 ± 2.3 | 63.0 ± 5.8 | 85.4 ± 2.4 |
| TracIn$_{we}$ | Rank | WE | 10 | 0.36 ± 0.15 | 85.8 ± 1.4 | 71.7 ± 1.8 | 93.9 ± 0.9 | 73.9 ± 5.6 | 77.6 ± 2.2 | 84.1 ± 1.6 | 63.6 ± 6.7 | 85.2 ± 1.5 |
| Cosine | Mean | 00-05 | 10 | 0.36 ± 0.14 | 85.2 ± 2.6 | 71.2 ± 2.4 | 93.8 ± 0.9 | 74.6 ± 4.8 | 77.7 ± 2.0 | 84.4 ± 2.0 | 63.5 ± 5.0 | 84.5 ± 2.0 |
| Cosine | Vote | WE | 10 | 0.36 ± 0.14 | 85.4 ± 1.9 | 72.7 ± 3.3 | 93.9 ± 1.0 | 74.0 ± 4.7 | 78.0 ± 2.6 | 83.6 ± 2.1 | 61.7 ± 5.6 | 85.3 ± 2.8 |
| Cosine | Rank | WE | 10 | 0.34 ± 0.15 | 86.0 ± 2.1 | 72.3 ± 2.7 | 93.9 ± 1.1 | 74.4 ± 5.2 | 77.7 ± 2.9 | 84.2 ± 1.6 | 62.9 ± 6.5 | 84.8 ± 1.8 |
| TracIn | Mean | 18-23 | 10 | 0.34 ± 0.12 | 83.1 ± 4.7 | 76.5 ± 1.7 | 93.3 ± 1.5 | 75.2 ± 2.5 | 80.7 ± 2.5 | 82.7 ± 3.0 | 56.5 ± 5.4 | 83.2 ± 2.1 |
| Cosine | Rank | 00-05 | 10 | 0.33 ± 0.14 | 85.7 ± 2.6 | 70.2 ± 1.1 | 94.3 ± 0.8 | 74.3 ± 5.7 | 77.6 ± 2.2 | 83.7 ± 1.9 | 61.4 ± 6.0 | 84.9 ± 2.2 |
| Cosine | Vote | CL | 10 | 0.33 ± 0.15 | 87.0 ± 2.3 | 70.8 ± 1.1 | 94.1 ± 0.7 | 73.8 ± 4.6 | 75.3 ± 1.8 | 85.4 ± 2.3 | 58.8 ± 5.5 | 85.7 ± 1.2 |
| Cosine | Vote | 06-11 | 10 | 0.33 ± 0.15 | 86.1 ± 1.9 | 71.5 ± 2.0 | 94.1 ± 0.6 | 72.5 ± 4.8 | 76.7 ± 1.7 | 84.3 ± 1.7 | 61.4 ± 6.0 | 85.7 ± 1.8 |
| Cosine | Vote | 00-05 | 10 | 0.33 ± 0.15 | 85.6 ± 2.4 | 70.7 ± 1.3 | 94.3 ± 0.6 | 73.9 ± 5.9 | 77.8 ± 2.5 | 80.8 ± 10.1 | 60.5 ± 5.3 | 85.3 ± 1.7 |
| TracIn | Rank | 00-05 | 10 | 0.31 ± 0.14 | 85.8 ± 1.9 | 70.5 ± 1.3 | 94.0 ± 1.2 | 73.6 ± 4.4 | 77.0 ± 2.3 | 83.9 ± 2.4 | 62.3 ± 5.9 | 85.4 ± 2.2 |
| DataInf | Rank | CL | 10 | 0.29 ± 0.15 | 84.9 ± 2.5 | 76.1 ± 2.2 | 94.0 ± 0.8 | 75.0 ± 3.2 | 76.5 ± 1.9 | 83.5 ± 2.0 | 60.8 ± 4.7 | 82.3 ± 1.3 |
| TracIn | Mean | all | 10 | 0.29 ± 0.12 | 79.8 ± 4.2 | 77.2 ± 3.0 | 90.7 ± 3.2 | 75.9 ± 2.9 | 79.1 ± 2.4 | 79.9 ± 3.0 | 60.0 ± 5.7 | 82.6 ± 2.2 |
| TracIn | Mean | CL | 10 | 0.21 ± 0.10 | 80.2 ± 4.2 | 75.9 ± 2.7 | 90.0 ± 2.7 | 74.6 ± 2.4 | 78.2 ± 1.7 | 79.8 ± 3.4 | 58.7 ± 5.1 | 81.6 ± 2.0 |
| TracIn | Rank | CL | 10 | 0.18 ± 0.11 | 79.1 ± 4.0 | 75.4 ± 1.9 | 87.9 ± 3.7 | 76.1 ± 2.6 | 76.8 ± 2.0 | 79.1 ± 3.1 | 59.2 ± 4.4 | 83.2 ± 1.8 |
| Cosine | Mean | CL | 10 | 0.17 ± 0.11 | 83.7 ± 2.8 | 71.6 ± 1.9 | 92.5 ± 1.7 | 75.2 ± 4.5 | 76.3 ± 1.7 | 79.1 ± 1.9 | 60.7 ± 4.5 | 81.8 ± 1.3 |

Rows are ordered according to mean rank from the best method to the worst. Two baselines are Random (dropping 30% uniformly) and Full (removing all noise (20%) and 10% of random clean samples). A good filtering method should outperform the Random, the lower bound, and approach the Full, upper bound. The tables highlight the best non-baseline methods for each GLUE dataset. AUC measure is specified for the NDR curve; a high value indicates skew of noise towards the beginning of the influence range.

Table 5: Llama-3.2 1B best **test set accuracy** on GLUE after 30% filtering with influence scores. Methods are ordered from best to worst with average ranking across 10 runs per setup. The highest accuracy means are highlighted. In contrast to Roberta and Mistral, influence methods demonstrate poor performance of noise detection compared to Random baseline. Llama noise distribution on figure 21 confirms that a lot of mislabeled training samples become influential.

| Method | Agg | Layer | Rank | Win Rate | QNLI | MRPC | SST2 | QQP | COLA | MNLI | RTE | STSB |
|---|---|---|---|---|---|---|---|---|---|---|---|---|
| Full | | | 0 | 0.97 ± 0.12 | 83.8 ± 1.5 | 79.9 ± 2.5 | 93.5 ± 0.9 | 84.4 ± 1.9 | 83.4 ± 0.5 | 82.0 ± 1.7 | 64.9 ± 4.3 | 85.9 ± 1.3 |
| Cosine | Rank | 04-07 | 1 | 0.65 ± 0.15 | 77.5 ± 3.5 | 70.2 ± 1.3 | 91.3 ± 1.7 | 73.3 ± 2.3 | 75.8 ± 2.0 | **77.3** ± 2.7 | 61.4 ± 4.0 | 80.9 ± 3.0 |
| Random | | | 1 | 0.64 ± 0.15 | 76.7 ± 5.1 | 71.6 ± 2.5 | 89.5 ± 2.0 | **80.2** ± 2.1 | **80.5** ± 0.9 | 73.3 ± 2.7 | 56.3 ± 3.9 | **82.2** ± 2.5 |
| DataInf | Rank | 04-07 | 1 | 0.64 ± 0.15 | 76.9 ± 3.4 | 70.7 ± 1.3 | 90.8 ± 1.7 | 73.0 ± 1.7 | 76.0 ± 2.4 | 76.9 ± 2.9 | 61.9 ± 4.1 | 81.5 ± 2.7 |
| DataInf | Mean | 04-07 | 1 | 0.63 ± 0.15 | 76.5 ± 3.6 | 70.9 ± 3.9 | 88.1 ± 6.4 | 77.5 ± 3.3 | 78.1 ± 2.6 | 73.6 ± 4.2 | **63.0** ± 4.1 | 78.9 ± 3.9 |
| TracIn | Rank | 04-07 | 1 | 0.63 ± 0.15 | 76.6 ± 3.4 | 70.6 ± 1.0 | 90.8 ± 2.3 | 72.9 ± 2.6 | 75.7 ± 2.2 | 77.0 ± 2.9 | 61.4 ± 4.7 | 80.9 ± 3.2 |
| DataInf | Rank | 00-03 | 1 | 0.59 ± 0.16 | 77.7 ± 2.5 | 70.4 ± 1.3 | 89.1 ± 4.5 | 73.2 ± 3.2 | 75.7 ± 2.2 | 76.0 ± 2.6 | 61.3 ± 4.7 | 78.9 ± 2.7 |
| DataInf | Mean | 00-03 | 1 | 0.58 ± 0.16 | 76.2 ± 2.4 | 71.2 ± 1.6 | 84.5 ± 10.1 | 76.5 ± 3.9 | 78.5 ± 1.4 | 73.4 ± 3.5 | 59.9 ± 3.6 | 78.1 ± 3.2 |
| TracIn$_{we}$ | Vote | WE | 1 | 0.58 ± 0.17 | 74.6 ± 4.3 | 71.8 ± 2.0 | 89.5 ± 3.1 | 75.2 ± 4.1 | 78.2 ± 0.5 | 73.4 ± 3.4 | 58.0 ± 4.1 | 78.6 ± 2.7 |
| TracIn$_{we}^{10}$ | Vote | WE | 1 | 0.58 ± 0.17 | 73.9 ± 4.9 | 72.0 ± 2.1 | 90.0 ± 3.2 | 75.7 ± 4.1 | 76.3 ± 1.8 | 73.3 ± 3.1 | 58.3 ± 3.9 | 78.9 ± 2.6 |
| Cosine | Rank | all | 1 | 0.57 ± 0.16 | 77.5 ± 3.3 | 70.3 ± 0.9 | 91.3 ± 2.1 | 72.3 ± 1.8 | 75.6 ± 1.1 | 77.0 ± 3.1 | 61.8 ± 4.8 | 71.9 ± 3.3 |
| Cosine | Rank | 08-11 | 1 | 0.57 ± 0.16 | 76.5 ± 4.2 | 70.7 ± 1.0 | 91.2 ± 2.2 | 73.1 ± 2.2 | 75.4 ± 1.3 | 76.7 ± 3.3 | 61.5 ± 4.8 | 71.0 ± 2.5 |
| DataInf | Vote | 04-07 | 1 | 0.56 ± 0.18 | 74.2 ± 4.7 | 70.9 ± 1.0 | 91.1 ± 1.7 | 73.9 ± 4.7 | 75.5 ± 2.2 | 74.8 ± 2.8 | 59.2 ± 3.9 | 79.6 ± 3.1 |
| TracIn | Vote | 00-03 | 1 | 0.56 ± 0.18 | 73.6 ± 5.1 | **72.4** ± 2.3 | 90.1 ± 3.0 | 74.4 ± 4.3 | 77.0 ± 1.8 | 72.9 ± 3.4 | 58.0 ± 4.4 | 77.7 ± 3.9 |
| DataInf | Rank | all | 1 | 0.56 ± 0.17 | 77.2 ± 3.1 | 70.6 ± 1.0 | 90.2 ± 4.1 | 72.3 ± 2.3 | 75.3 ± 1.1 | 76.9 ± 3.0 | 60.5 ± 3.7 | 73.5 ± 1.6 |
| TracIn | Rank | all | 1 | 0.55 ± 0.16 | 77.7 ± 3.4 | 70.8 ± 1.1 | 90.8 ± 2.7 | 71.9 ± 2.0 | 75.2 ± 1.5 | 76.4 ± 3.3 | 59.8 ± 3.7 | 73.2 ± 2.4 |
| DataInf | Vote | 00-03 | 1 | 0.54 ± 0.18 | 73.3 ± 5.6 | 72.1 ± 2.0 | 90.2 ± 2.8 | 74.2 ± 4.3 | 76.8 ± 1.8 | 73.0 ± 3.4 | 58.5 ± 4.0 | 77.4 ± 4.7 |
| TracIn | Vote | WE | 1 | 0.54 ± 0.18 | 75.1 ± 4.4 | 71.4 ± 1.6 | 89.3 ± 2.8 | 74.0 ± 4.3 | 75.9 ± 1.5 | 73.4 ± 3.0 | 58.6 ± 3.6 | 78.1 ± 1.9 |
| DataInf | Rank | 08-11 | 1 | 0.53 ± 0.17 | 76.6 ± 3.8 | 70.8 ± 0.7 | 90.4 ± 3.1 | 72.6 ± 1.8 | 74.9 ± 1.3 | 76.4 ± 3.7 | 60.1 ± 3.7 | 72.3 ± 2.2 |
| Cosine | Mean | 04-07 | 2 | 0.59 ± 0.15 | 77.6 ± 3.0 | 70.0 ± 1.0 | 91.1 ± 1.9 | 73.4 ± 2.7 | 75.1 ± 2.0 | 74.5 ± 3.1 | 61.3 ± 3.8 | 81.6 ± 2.8 |
| Cosine | Rank | 00-03 | 2 | 0.58 ± 0.15 | 78.0 ± 2.4 | 70.0 ± 1.0 | 89.6 ± 4.0 | 72.6 ± 3.3 | 75.9 ± 2.1 | 76.1 ± 2.7 | 61.5 ± 4.5 | 78.9 ± 2.7 |
| Cosine | Mean | 00-03 | 2 | 0.58 ± 0.15 | 78.2 ± 1.4 | 70.3 ± 1.2 | 90.1 ± 2.7 | 72.1 ± 3.9 | 76.0 ± 2.3 | 74.5 ± 3.3 | 62.0 ± 3.9 | 79.0 ± 3.5 |
| TracIn | Rank | 04-07 | 2 | 0.57 ± 0.16 | 77.7 ± 2.4 | 70.0 ± 1.1 | 88.6 ± 5.5 | 73.3 ± 3.1 | 75.9 ± 2.1 | 75.6 ± 2.7 | 61.0 ± 4.7 | 78.3 ± 3.1 |
| TracIn | Vote | 04-07 | 2 | 0.55 ± 0.18 | 74.5 ± 4.6 | 71.0 ± 1.0 | 90.8 ± 2.1 | 73.8 ± 4.5 | 75.4 ± 2.3 | 74.4 ± 3.0 | 59.0 ± 3.9 | 79.2 ± 3.2 |
| TracIn$_{we}^{10}$ | Mean | WE | 2 | 0.52 ± 0.15 | 74.6 ± 3.4 | 70.7 ± 2.9 | 83.3 ± 7.4 | 76.3 ± 2.9 | 79.1 ± 3.0 | 72.6 ± 4.8 | 57.9 ± 4.2 | 77.0 ± 3.7 |
| DataInf | Mean | all | 2 | 0.50 ± 0.16 | 74.6 ± 4.2 | 70.1 ± 2.5 | 81.7 ± 12.0 | 75.2 ± 3.6 | 79.1 ± 2.4 | 72.3 ± 3.0 | 59.5 ± 4.6 | 73.1 ± 6.1 |
| Cosine | Mean | 08-11 | 2 | 0.49 ± 0.16 | 76.3 ± 3.8 | 70.9 ± 1.3 | 91.9 ± 1.5 | 72.8 ± 2.8 | 75.1 ± 1.4 | 71.4 ± 3.5 | 60.0 ± 2.7 | 71.8 ± 2.4 |
| DataInf | Vote | all | 2 | 0.46 ± 0.18 | 72.1 ± 6.0 | 71.0 ± 1.4 | 89.4 ± 4.9 | 72.8 ± 4.0 | 75.2 ± 1.8 | 72.0 ± 3.6 | 58.4 ± 4.3 | 76.6 ± 3.4 |
| Cosine | Vote | 04-07 | 3 | 0.54 ± 0.15 | 77.6 ± 3.7 | 70.1 ± 1.0 | 90.2 ± 2.4 | 70.7 ± 2.7 | 74.9 ± 2.6 | 76.0 ± 2.9 | 60.3 ± 4.0 | 81.6 ± 2.2 |
| TracIn | Mean | 04-07 | 3 | 0.53 ± 0.14 | 75.6 ± 3.3 | 70.0 ± 3.4 | 81.0 ± 10.2 | 76.0 ± 4.2 | 78.4 ± 2.0 | 73.1 ± 4.0 | 61.2 ± 5.2 | 75.8 ± 5.4 |
| TracIn$_{we}$ | Rank | WE | 3 | 0.52 ± 0.16 | 76.2 ± 3.7 | 70.7 ± 1.4 | 88.2 ± 3.7 | 73.6 ± 4.2 | 76.5 ± 2.7 | 73.4 ± 3.6 | 58.8 ± 4.5 | 77.7 ± 2.7 |
| TracIn | Rank | 08-11 | 3 | 0.50 ± 0.17 | 75.8 ± 4.7 | 70.9 ± 0.8 | 90.3 ± 3.3 | 72.1 ± 2.0 | 74.8 ± 1.8 | 76.0 ± 4.2 | 59.1 ± 3.3 | 72.1 ± 1.9 |
| Cosine | Mean | all | 3 | 0.49 ± 0.16 | 77.2 ± 2.6 | 70.5 ± 0.9 | **92.3** ± 1.1 | 72.5 ± 2.2 | 75.3 ± 1.3 | 70.9 ± 2.8 | 59.9 ± 3.4 | 72.7 ± 3.6 |
| DataInf | Mean | 08-11 | 3 | 0.49 ± 0.15 | 74.4 ± 5.4 | 69.9 ± 2.2 | 82.1 ± 12.7 | 74.9 ± 2.8 | 78.6 ± 2.2 | 72.3 ± 3.7 | 58.6 ± 4.5 | 72.7 ± 4.8 |
| DataInf | Vote | 08-11 | 3 | 0.46 ± 0.18 | 72.8 ± 5.6 | 70.9 ± 1.3 | 89.9 ± 3.1 | 72.9 ± 3.5 | 75.0 ± 1.4 | 72.3 ± 4.4 | 58.1 ± 4.6 | 75.4 ± 3.3 |
| TracIn | Vote | all | 3 | 0.42 ± 0.18 | 71.8 ± 5.8 | 71.1 ± 1.3 | 89.7 ± 3.6 | 72.3 ± 3.9 | 75.0 ± 1.9 | 71.9 ± 3.7 | 57.9 ± 4.4 | 75.9 ± 3.4 |
| TracIn | Rank | 12-15 | 3 | 0.36 ± 0.17 | 70.5 ± 5.9 | 71.2 ± 1.6 | 88.0 ± 6.2 | 72.0 ± 3.2 | 74.4 ± 1.4 | 70.5 ± 4.4 | 57.1 ± 4.3 | 74.2 ± 2.6 |
| TracIn | Mean | 00-03 | 4 | 0.52 ± 0.14 | 76.1 ± 2.4 | 70.8 ± 2.2 | 82.2 ± 10.6 | 75.9 ± 3.8 | 78.1 ± 2.7 | 73.6 ± 4.2 | 59.9 ± 4.0 | 74.9 ± 4.4 |
| TracIn$_{we}$ | Mean | WE | 4 | 0.51 ± 0.15 | 73.0 ± 4.9 | 71.2 ± 1.9 | 79.3 ± 12.0 | 74.8 ± 3.9 | 79.5 ± 1.8 | 72.9 ± 4.5 | 57.9 ± 4.4 | 75.3 ± 2.9 |
| Cosine | Mean | WE | 4 | 0.48 ± 0.15 | 76.0 ± 4.4 | 70.7 ± 1.5 | 89.2 ± 2.8 | 72.6 ± 3.1 | 74.9 ± 2.3 | 72.8 ± 5.6 | 60.2 ± 4.6 | 75.1 ± 3.6 |
| TracIn | Vote | 08-11 | 4 | 0.42 ± 0.18 | 72.0 ± 5.6 | 70.8 ± 1.4 | 90.2 ± 3.0 | 71.8 ± 3.6 | 74.9 ± 1.4 | 72.0 ± 4.0 | 57.7 ± 4.6 | 75.3 ± 3.3 |
| DataInf | Rank | CL | 4 | 0.40 ± 0.17 | 75.2 ± 5.1 | 71.1 ± 1.4 | 85.1 ± 6.1 | 71.0 ± 2.4 | 74.7 ± 1.3 | 74.7 ± 3.3 | 59.1 ± 3.8 | 70.5 ± 2.1 |
| DataInf | Vote | 12-15 | 4 | 0.37 ± 0.17 | 71.3 ± 6.0 | 70.9 ± 1.4 | 88.7 ± 5.5 | 71.8 ± 3.0 | 74.5 ± 1.6 | 70.9 ± 4.1 | 57.1 ± 4.4 | 74.3 ± 2.7 |
| Cosine | Vote | WE | 5 | 0.50 ± 0.15 | 77.6 ± 3.6 | 70.5 ± 1.7 | 88.4 ± 2.2 | 72.6 ± 4.3 | 76.2 ± 2.6 | 74.1 ± 3.4 | 57.0 ± 3.0 | 79.0 ± 3.1 |
| Cosine | Vote | 00-03 | 5 | 0.50 ± 0.16 | 78.1 ± 3.3 | 69.8 ± 1.4 | 88.3 ± 3.0 | 70.6 ± 3.9 | 75.4 ± 2.3 | 75.4 ± 2.8 | 60.6 ± 4.6 | 77.8 ± 3.2 |
| TracIn$_{we}^{10}$ | Rank | WE | 5 | 0.49 ± 0.16 | 76.8 ± 3.6 | 70.2 ± 1.3 | 88.7 ± 4.1 | 72.4 ± 5.1 | 75.0 ± 2.2 | 74.1 ± 3.4 | 58.7 ± 3.8 | 79.4 ± 2.0 |
| DataInf | Rank | 12-15 | 5 | 0.43 ± 0.16 | 75.4 ± 3.7 | 71.3 ± 1.4 | 85.9 ± 9.4 | 71.7 ± 2.1 | 75.1 ± 1.0 | 74.3 ± 4.2 | 58.6 ± 2.7 | 70.4 ± 2.1 |
| TracIn | Mean | WE | 6 | 0.45 ± 0.14 | 73.7 ± 4.3 | 70.4 ± 2.0 | 76.8 ± 13.3 | 74.0 ± 3.8 | 79.0 ± 2.8 | 72.3 ± 5.1 | 58.2 ± 4.9 | 74.2 ± 4.7 |
| Cosine | Rank | WE | 6 | 0.43 ± 0.16 | 77.2 ± 2.5 | 69.9 ± 1.0 | 86.9 ± 7.4 | 72.0 ± 3.8 | 75.0 ± 1.8 | 73.5 ± 3.6 | 58.6 ± 4.7 | 75.5 ± 4.1 |
| TracIn | Rank | WE | 6 | 0.43 ± 0.16 | 76.6 ± 2.4 | 69.8 ± 1.3 | 86.9 ± 6.5 | 72.1 ± 3.4 | 74.8 ± 1.9 | 73.7 ± 3.6 | 59.2 ± 4.7 | 76.1 ± 3.1 |
| DataInf | Vote | CL | 6 | 0.33 ± 0.16 | 70.2 ± 6.5 | 71.0 ± 2.1 | 87.6 ± 7.7 | 71.8 ± 3.5 | 73.9 ± 1.8 | 70.6 ± 4.7 | 57.0 ± 4.0 | 74.0 ± 3.8 |
| Cosine | Rank | 12-15 | 7 | 0.42 ± 0.15 | 75.8 ± 4.1 | 70.3 ± 0.9 | 88.1 ± 6.2 | 71.0 ± 2.2 | 75.5 ± 1.6 | 75.0 ± 4.0 | 59.5 ± 3.6 | 68.1 ± 1.4 |
| TracIn | Rank | 12-15 | 7 | 0.36 ± 0.15 | 75.1 ± 4.9 | 71.0 ± 1.4 | 82.9 ± 12.8 | 70.7 ± 2.0 | 74.7 ± 1.1 | 73.6 ± 4.3 | 57.5 ± 2.3 | 70.3 ± 2.2 |
| Cosine | Vote | 12-15 | 7 | 0.35 ± 0.16 | 73.1 ± 3.6 | 70.6 ± 0.9 | 90.6 ± 1.1 | 70.7 ± 3.6 | 74.2 ± 1.2 | 69.7 ± 2.6 | 59.3 ± 3.0 | 68.4 ± 2.5 |
| TracIn | Mean | 08-11 | 8 | 0.39 ± 0.14 | 70.7 ± 7.2 | 70.6 ± 2.7 | 79.7 ± 14.4 | 73.0 ± 2.9 | 77.7 ± 3.1 | 69.9 ± 3.9 | 57.4 ± 5.1 | 71.8 ± 3.9 |
| Cosine | Vote | 08-11 | 9 | 0.42 ± 0.15 | 77.7 ± 3.3 | 70.2 ± 0.6 | 89.8 ± 2.4 | 69.9 ± 2.0 | 74.2 ± 1.3 | 76.1 ± 2.5 | 58.0 ± 4.1 | 70.9 ± 2.8 |
| Cosine | Vote | all | 9 | 0.41 ± 0.14 | **78.8** ± 3.1 | 70.3 ± 0.9 | 88.5 ± 3.0 | 70.4 ± 1.4 | 74.4 ± 1.4 | 75.8 ± 3.0 | 58.8 ± 4.4 | 70.8 ± 2.5 |
| Cosine | Rank | CL | 9 | 0.40 ± 0.13 | 75.4 ± 4.4 | 70.2 ± 0.8 | 87.7 ± 6.3 | 71.6 ± 3.2 | 73.8 ± 1.5 | 76.1 ± 2.7 | 60.7 ± 4.3 | 67.7 ± 1.8 |
| Cosine | Vote | 12-15 | 9 | 0.33 ± 0.14 | 77.2 ± 3.6 | 70.2 ± 0.8 | 86.6 ± 4.6 | 68.9 ± 1.6 | 74.2 ± 1.2 | 74.9 ± 2.8 | 56.5 ± 3.1 | 68.5 ± 2.3 |
| Cosine | Vote | CL | 9 | 0.32 ± 0.14 | 76.2 ± 5.4 | 70.1 ± 1.0 | 86.7 ± 4.8 | 69.5 ± 2.6 | 71.8 ± 1.8 | 75.6 ± 2.4 | 57.5 ± 3.4 | 69.3 ± 2.5 |
| DataInf | Mean | 12-15 | 9 | 0.31 ± 0.14 | 70.6 ± 6.3 | 69.2 ± 3.6 | 77.7 ± 10.0 | 72.9 ± 3.7 | 77.8 ± 2.5 | 69.8 ± 4.5 | 56.4 ± 3.4 | 69.6 ± 5.0 |
| TracIn | Vote | CL | 9 | 0.28 ± 0.15 | 69.2 ± 5.9 | 70.7 ± 1.7 | 87.9 ± 6.2 | 72.0 ± 3.8 | 73.4 ± 1.5 | 70.2 ± 4.5 | 56.6 ± 4.4 | 73.4 ± 3.7 |
| DataInf | Mean | CL | 9 | 0.28 ± 0.13 | 66.7 ± 5.8 | 69.4 ± 4.8 | 81.9 ± 7.1 | 72.8 ± 4.5 | 76.4 ± 2.3 | 68.4 ± 3.4 | 56.1 ± 2.2 | 68.1 ± 6.0 |
| Cosine | Mean | CL | 9 | 0.23 ± 0.12 | 70.3 ± 4.0 | 70.2 ± 1.3 | 79.8 ± 10.2 | 70.8 ± 2.4 | 73.6 ± 1.5 | 69.3 ± 2.4 | 60.5 ± 2.7 | 65.6 ± 1.2 |
| TracIn | Rank | CL | 9 | 0.22 ± 0.12 | 70.7 ± 7.9 | 70.6 ± 1.7 | 82.2 ± 9.6 | 72.6 ± 4.0 | 73.7 ± 2.0 | 71.4 ± 4.6 | 55.4 ± 2.0 | 70.0 ± 2.6 |
| TracIn | Mean | 12-15 | 9 | 0.22 ± 0.11 | 66.5 ± 6.1 | 68.5 ± 5.1 | 77.2 ± 10.9 | 72.0 ± 4.0 | 75.7 ± 2.3 | 66.5 ± 3.8 | 55.3 ± 2.0 | 69.4 ± 2.6 |
| TracIn | Mean | CL | 9 | 0.17 ± 0.09 | 65.2 ± 5.3 | 68.4 ± 4.9 | 83.4 ± 5.9 | 69.9 ± 3.1 | 74.1 ± 1.7 | 65.6 ± 4.2 | 55.2 ± 2.3 | 69.2 ± 3.1 |
| TracIn | Mean | all | 9 | 0.16 ± 0.09 | 65.6 ± 5.4 | 68.6 ± 4.5 | 84.0 ± 6.1 | 70.1 ± 3.0 | 73.4 ± 2.1 | 65.4 ± 4.1 | 55.4 ± 2.6 | 68.9 ± 2.8 |

Table 6: Mistral 7B best **test set accuracy** on GLUE after 30% filtering with influence scores. Methods are ordered from best to worst with average ranking across 10 runs per setup. The highest accuracy means are highlighted.

| Method | Agg | Layer | Rank | Win Rate | QNLI | MRPC | SST2 | QQP | COLA | MNLI | RTE | STSB |
|---|---|---|---|---|---|---|---|---|---|---|---|---|---|
| Full | | | 0 | $0.96 \pm 0.12$ | $87.1 \pm 1.6$ | $77.2 \pm 1.2$ | $93.8 \pm 1.1$ | $84.2 \pm 2.1$ | $83.4 \pm 1.3$ | $82.1 \pm 2.8$ | $69.0 \pm 3.5$ | $78.5 \pm 3.2$ |
| DataInf | Vote | 00-07 | 1 | $0.84 \pm 0.22$ | $82.9 \pm 1.7$ | $\mathbf{74.2} \pm 2.4$ | $89.3 \pm 2.2$ | $\mathbf{80.7} \pm 2.3$ | $80.6 \pm 1.9$ | $79.0 \pm 3.0$ | $66.6 \pm 2.9$ | $77.2 \pm 3.2$ |
| TracIn | Vote | 00-07 | 1 | $0.83 \pm 0.23$ | $83.5 \pm 1.6$ | $74.1 \pm 2.7$ | $89.2 \pm 2.1$ | $80.3 \pm 2.4$ | $80.1 \pm 1.6$ | $78.5 \pm 2.8$ | $66.6 \pm 3.2$ | $76.7 \pm 3.1$ |
| DataInf | Vote | 08-15 | 1 | $0.82 \pm 0.21$ | $83.9 \pm 2.0$ | $72.5 \pm 1.6$ | $90.2 \pm 2.3$ | $78.9 \pm 2.4$ | $77.1 \pm 1.9$ | $79.4 \pm 3.0$ | $66.6 \pm 4.0$ | $78.1 \pm 3.1$ |
| DataInf | Vote | all | 1 | $0.82 \pm 0.20$ | $84.1 \pm 1.9$ | $72.5 \pm 1.5$ | $89.5 \pm 2.1$ | $77.0 \pm 2.3$ | $76.6 \pm 1.9$ | $\mathbf{79.7} \pm 2.5$ | $67.6 \pm 3.7$ | $\mathbf{78.5} \pm 3.1$ |
| TracIn | Vote | all | 1 | $0.81 \pm 0.19$ | $\mathbf{84.4} \pm 2.1$ | $72.1 \pm 1.6$ | $\mathbf{90.7} \pm 1.9$ | $77.3 \pm 2.4$ | $76.4 \pm 1.8$ | $79.4 \pm 2.9$ | $67.7 \pm 3.4$ | $78.4 \pm 3.1$ |
| TracIn | Vote | 08-15 | 1 | $0.80 \pm 0.22$ | $83.8 \pm 1.6$ | $72.3 \pm 1.3$ | $90.1 \pm 2.0$ | $78.4 \pm 2.3$ | $76.5 \pm 1.8$ | $79.4 \pm 3.0$ | $66.8 \pm 2.9$ | $77.9 \pm 2.9$ |
| TracIn$_{we}^{10}$ | Vote | WE | 1 | $0.78 \pm 0.25$ | $82.1 \pm 1.8$ | $74.1 \pm 2.0$ | $88.2 \pm 2.5$ | $79.0 \pm 2.6$ | $80.2 \pm 1.9$ | $77.5 \pm 2.5$ | $65.7 \pm 3.1$ | $76.3 \pm 3.4$ |
| TracIn | Vote | 24-31 | 1 | $0.76 \pm 0.23$ | $83.3 \pm 1.8$ | $72.2 \pm 1.3$ | $90.6 \pm 1.9$ | $75.3 \pm 2.8$ | $75.8 \pm 1.3$ | $\mathbf{79.7} \pm 3.1$ | $66.9 \pm 3.7$ | $77.8 \pm 3.4$ |
| TracIn | Vote | CL | 1 | $0.75 \pm 0.23$ | $83.8 \pm 2.0$ | $72.1 \pm 1.3$ | $90.3 \pm 1.9$ | $75.9 \pm 2.8$ | $75.4 \pm 1.2$ | $79.3 \pm 2.6$ | $66.9 \pm 3.4$ | $78.0 \pm 3.5$ |
| DataInf | Vote | 16-23 | 1 | $0.75 \pm 0.24$ | $83.5 \pm 1.7$ | $72.0 \pm 1.4$ | $89.1 \pm 2.0$ | $75.1 \pm 3.0$ | $75.6 \pm 1.7$ | $79.5 \pm 3.0$ | $\mathbf{67.9} \pm 3.5$ | $78.0 \pm 3.1$ |
| DataInf | Vote | 24-31 | 1 | $0.74 \pm 0.24$ | $83.6 \pm 1.7$ | $72.4 \pm 2.0$ | $88.9 \pm 2.1$ | $74.7 \pm 2.8$ | $75.4 \pm 1.2$ | $79.4 \pm 2.6$ | $67.4 \pm 3.5$ | $77.8 \pm 3.0$ |
| TracIn$_{we}$ | Vote | WE | 2 | $0.76 \pm 0.26$ | $82.2 \pm 1.8$ | $72.7 \pm 2.1$ | $87.7 \pm 2.2$ | $78.8 \pm 2.5$ | $\mathbf{81.0} \pm 1.7$ | $77.3 \pm 3.0$ | $65.6 \pm 2.9$ | $76.7 \pm 2.9$ |
| TracIn | Vote | WE | 2 | $0.76 \pm 0.27$ | $82.2 \pm 1.6$ | $73.0 \pm 2.3$ | $87.4 \pm 1.9$ | $78.5 \pm 2.9$ | $79.3 \pm 2.2$ | $77.6 \pm 3.2$ | $66.0 \pm 3.0$ | $76.4 \pm 3.3$ |
| TracIn | Vote | 16-23 | 2 | $0.73 \pm 0.22$ | $83.6 \pm 1.8$ | $71.3 \pm 1.0$ | $90.5 \pm 2.0$ | $75.8 \pm 2.8$ | $75.6 \pm 1.5$ | $79.6 \pm 2.7$ | $67.1 \pm 3.3$ | $77.9 \pm 2.8$ |
| DataInf | Vote | CL | 2 | $0.71 \pm 0.26$ | $83.4 \pm 1.8$ | $71.8 \pm 1.3$ | $88.1 \pm 2.1$ | $75.1 \pm 3.6$ | $75.6 \pm 1.3$ | $79.1 \pm 2.6$ | $66.9 \pm 3.8$ | $77.1 \pm 2.9$ |
| DataInf | Mean | 08-15 | 3 | $0.63 \pm 0.24$ | $82.5 \pm 4.0$ | $71.6 \pm 3.6$ | $82.1 \pm 11.4$ | $76.2 \pm 4.3$ | $76.6 \pm 2.6$ | $76.1 \pm 2.1$ | $66.2 \pm 3.9$ | $74.0 \pm 4.1$ |
| TracIn | Mean | 00-07 | 3 | $0.59 \pm 0.27$ | $80.4 \pm 1.7$ | $72.0 \pm 1.2$ | $82.0 \pm 5.6$ | $76.4 \pm 3.7$ | $77.7 \pm 2.5$ | $75.7 \pm 2.9$ | $63.8 \pm 4.7$ | $73.8 \pm 3.3$ |
| TracIn$_{we}^{10}$ | Mean | WE | 3 | $0.57 \pm 0.25$ | $79.7 \pm 2.4$ | $72.1 \pm 2.1$ | $82.1 \pm 2.6$ | $77.3 \pm 2.2$ | $78.9 \pm 1.9$ | $75.3 \pm 3.3$ | $62.5 \pm 3.3$ | $72.8 \pm 4.0$ |
| DataInf | Mean | 00-07 | 3 | $0.57 \pm 0.25$ | $80.7 \pm 2.1$ | $71.9 \pm 2.6$ | $83.2 \pm 3.2$ | $76.7 \pm 3.2$ | $76.9 \pm 1.9$ | $76.4 \pm 3.0$ | $61.9 \pm 3.7$ | $71.6 \pm 3.6$ |
| DataInf | Mean | all | 3 | $0.53 \pm 0.29$ | $80.2 \pm 4.5$ | $70.6 \pm 4.9$ | $75.0 \pm 12.9$ | $73.1 \pm 2.8$ | $76.1 \pm 1.8$ | $75.5 \pm 2.2$ | $64.7 \pm 5.2$ | $73.9 \pm 4.1$ |
| Cosine | Rank | 08-15 | 3 | $0.50 \pm 0.25$ | $83.1 \pm 1.4$ | $70.7 \pm 1.5$ | $87.7 \pm 3.5$ | $70.8 \pm 2.5$ | $74.0 \pm 2.0$ | $75.6 \pm 3.7$ | $63.8 \pm 4.1$ | $73.3 \pm 2.7$ |
| DataInf | Rank | 08-15 | 3 | $0.50 \pm 0.26$ | $82.5 \pm 1.5$ | $71.6 \pm 1.8$ | $86.6 \pm 4.0$ | $72.2 \pm 2.2$ | $74.3 \pm 1.6$ | $74.2 \pm 3.5$ | $63.6 \pm 3.9$ | $72.2 \pm 2.8$ |
| DataInf | Rank | 00-07 | 3 | $0.48 \pm 0.27$ | $80.1 \pm 2.5$ | $71.6 \pm 1.1$ | $84.2 \pm 4.3$ | $72.3 \pm 3.1$ | $75.3 \pm 2.3$ | $75.7 \pm 3.0$ | $63.1 \pm 3.6$ | $72.0 \pm 2.4$ |
| DataInf | Rank | all | 3 | $0.44 \pm 0.25$ | $79.2 \pm 1.8$ | $71.8 \pm 0.9$ | $81.9 \pm 7.3$ | $72.2 \pm 1.8$ | $75.5 \pm 1.4$ | $72.8 \pm 3.7$ | $64.2 \pm 2.9$ | $69.6 \pm 2.2$ |
| DataInf | Rank | 16-23 | 3 | $0.43 \pm 0.24$ | $77.9 \pm 4.7$ | $70.2 \pm 3.4$ | $69.3 \pm 9.4$ | $72.5 \pm 2.5$ | $76.4 \pm 1.8$ | $73.5 \pm 3.0$ | $66.1 \pm 4.0$ | $71.1 \pm 3.6$ |
| TracIn$_{we}$ | Mean | WE | 4 | $0.51 \pm 0.25$ | $79.4 \pm 3.1$ | $71.3 \pm 1.5$ | $80.3 \pm 5.5$ | $76.1 \pm 3.1$ | $78.9 \pm 1.6$ | $74.3 \pm 3.4$ | $62.3 \pm 3.6$ | $72.5 \pm 4.2$ |
| TracIn | Mean | WE | 4 | $0.50 \pm 0.25$ | $79.4 \pm 1.9$ | $71.3 \pm 1.4$ | $80.4 \pm 7.1$ | $75.0 \pm 2.8$ | $79.0 \pm 1.8$ | $74.7 \pm 4.0$ | $62.4 \pm 3.2$ | $71.5 \pm 4.4$ |
| TracIn | Mean | 08-15 | 4 | $0.50 \pm 0.22$ | $78.8 \pm 5.5$ | $70.8 \pm 3.7$ | $75.7 \pm 13.7$ | $75.3 \pm 4.3$ | $76.2 \pm 1.9$ | $72.7 \pm 2.2$ | $65.1 \pm 5.7$ | $74.0 \pm 4.9$ |
| TracIn$_{we}$ | Rank | WE | 4 | $0.49 \pm 0.26$ | $78.7 \pm 2.4$ | $72.2 \pm 1.5$ | $82.8 \pm 3.8$ | $72.5 \pm 2.6$ | $77.6 \pm 1.9$ | $73.6 \pm 2.5$ | $62.9 \pm 3.8$ | $73.0 \pm 2.8$ |
| Cosine | Mean | 00-07 | 4 | $0.48 \pm 0.25$ | $79.2 \pm 3.4$ | $71.6 \pm 1.3$ | $83.7 \pm 4.9$ | $74.6 \pm 3.4$ | $75.4 \pm 1.8$ | $74.4 \pm 4.0$ | $63.1 \pm 3.0$ | $72.4 \pm 3.4$ |
| TracIn$_{we}^{10}$ | Rank | WE | 4 | $0.48 \pm 0.27$ | $79.1 \pm 2.4$ | $71.4 \pm 1.5$ | $82.8 \pm 2.9$ | $73.3 \pm 2.1$ | $76.7 \pm 2.3$ | $74.1 \pm 2.5$ | $62.5 \pm 3.6$ | $73.9 \pm 3.2$ |
| Cosine | Rank | 00-07 | 4 | $0.48 \pm 0.27$ | $79.6 \pm 2.6$ | $71.5 \pm 1.5$ | $84.1 \pm 4.3$ | $72.0 \pm 3.0$ | $75.2 \pm 2.2$ | $75.7 \pm 2.9$ | $63.4 \pm 3.3$ | $72.6 \pm 2.7$ |
| TracIn | Rank | 08-15 | 4 | $0.47 \pm 0.27$ | $80.9 \pm 1.4$ | $71.6 \pm 1.3$ | $86.3 \pm 3.8$ | $72.6 \pm 2.5$ | $74.5 \pm 1.7$ | $73.9 \pm 3.5$ | $63.2 \pm 4.2$ | $71.8 \pm 2.3$ |
| TracIn | Rank | WE | 4 | $0.46 \pm 0.26$ | $78.8 \pm 2.4$ | $71.2 \pm 1.5$ | $83.5 \pm 3.5$ | $71.7 \pm 2.0$ | $76.0 \pm 2.3$ | $74.4 \pm 2.8$ | $63.5 \pm 3.7$ | $73.4 \pm 3.5$ |
| TracIn | Rank | 00-07 | 4 | $0.45 \pm 0.27$ | $79.0 \pm 2.7$ | $71.7 \pm 1.2$ | $83.9 \pm 3.7$ | $72.5 \pm 3.3$ | $74.9 \pm 2.2$ | $74.9 \pm 3.3$ | $63.2 \pm 3.5$ | $72.0 \pm 2.7$ |
| Cosine | Rank | WE | 4 | $0.43 \pm 0.26$ | $79.4 \pm 2.8$ | $70.7 \pm 1.4$ | $83.3 \pm 3.3$ | $70.9 \pm 2.6$ | $76.2 \pm 2.4$ | $73.7 \pm 2.0$ | $63.4 \pm 3.3$ | $73.2 \pm 3.6$ |
| TracIn | Rank | all | 4 | $0.41 \pm 0.25$ | $79.1 \pm 2.1$ | $71.7 \pm 1.4$ | $83.0 \pm 5.5$ | $71.7 \pm 1.9$ | $75.4 \pm 1.5$ | $71.9 \pm 3.8$ | $63.6 \pm 3.3$ | $70.3 \pm 2.2$ |
| Cosine | Rank | all | 4 | $0.40 \pm 0.24$ | $81.4 \pm 2.0$ | $70.1 \pm 1.5$ | $83.5 \pm 8.1$ | $69.6 \pm 1.9$ | $73.8 \pm 1.6$ | $76.0 \pm 4.8$ | $64.8 \pm 3.0$ | $67.8 \pm 2.1$ |
| Cosine | Rank | 16-23 | 4 | $0.36 \pm 0.23$ | $80.4 \pm 1.5$ | $70.0 \pm 1.1$ | $74.0 \pm 11.4$ | $69.1 \pm 2.1$ | $73.8 \pm 1.6$ | $75.7 \pm 4.9$ | $66.1 \pm 2.7$ | $67.4 \pm 3.0$ |
| Cosine | Mean | WE | 5 | $0.45 \pm 0.25$ | $79.5 \pm 3.3$ | $71.0 \pm 1.2$ | $82.8 \pm 3.1$ | $72.4 \pm 2.6$ | $75.9 \pm 2.6$ | $74.1 \pm 3.5$ | $63.6 \pm 4.7$ | $72.4 \pm 3.5$ |
| Random | | | 5 | $0.43 \pm 0.23$ | $76.9 \pm 2.3$ | $69.4 \pm 2.3$ | $80.8 \pm 3.2$ | $76.3 \pm 2.5$ | $77.9 \pm 1.7$ | $73.2 \pm 3.7$ | $60.5 \pm 3.3$ | $73.5 \pm 3.2$ |
| Cosine | Vote | 00-07 | 5 | $0.39 \pm 0.24$ | $78.4 \pm 3.4$ | $71.7 \pm 1.0$ | $77.1 \pm 4.1$ | $71.7 \pm 3.9$ | $75.1 \pm 1.7$ | $74.9 \pm 2.1$ | $61.9 \pm 3.8$ | $71.5 \pm 3.1$ |
| Cosine | Mean | 08-15 | 6 | $0.44 \pm 0.22$ | $80.4 \pm 3.4$ | $70.9 \pm 1.3$ | $88.9 \pm 3.4$ | $71.2 \pm 1.8$ | $73.2 \pm 1.9$ | $72.1 \pm 2.4$ | $63.1 \pm 3.6$ | $75.9 \pm 3.0$ |
| Cosine | Vote | WE | 6 | $0.38 \pm 0.25$ | $78.7 \pm 2.2$ | $70.5 \pm 1.4$ | $76.3 \pm 3.1$ | $71.6 \pm 3.6$ | $76.3 \pm 2.5$ | $73.6 \pm 2.7$ | $62.8 \pm 3.7$ | $72.6 \pm 3.3$ |
| DataInf | Rank | 16-23 | 6 | $0.37 \pm 0.24$ | $77.0 \pm 2.3$ | $71.8 \pm 1.3$ | $70.0 \pm 11.2$ | $72.4 \pm 2.0$ | $75.5 \pm 1.2$ | $72.1 \pm 3.6$ | $63.9 \pm 2.0$ | $69.6 \pm 2.2$ |
| Cosine | Vote | 08-15 | 7 | $0.38 \pm 0.23$ | $80.8 \pm 2.6$ | $71.0 \pm 2.1$ | $76.4 \pm 9.7$ | $67.4 \pm 3.3$ | $73.2 \pm 1.8$ | $75.4 \pm 3.4$ | $63.5 \pm 4.3$ | $72.3 \pm 3.0$ |
| DataInf | Rank | 24-31 | 7 | $0.35 \pm 0.20$ | $74.7 \pm 2.0$ | $72.3 \pm 1.0$ | $62.5 \pm 9.6$ | $71.9 \pm 1.9$ | $76.0 \pm 1.3$ | $70.4 \pm 4.2$ | $64.3 \pm 2.6$ | $69.8 \pm 2.0$ |
| TracIn | Rank | 16-23 | 7 | $0.34 \pm 0.22$ | $74.0 \pm 3.1$ | $71.9 \pm 1.0$ | $72.3 \pm 9.1$ | $72.3 \pm 1.7$ | $75.3 \pm 1.4$ | $70.4 \pm 4.1$ | $62.9 \pm 2.3$ | $70.6 \pm 2.5$ |
| DataInf | Rank | CL | 7 | $0.29 \pm 0.19$ | $74.4 \pm 3.0$ | $71.9 \pm 1.2$ | $60.8 \pm 5.8$ | $71.9 \pm 1.9$ | $75.2 \pm 1.6$ | $68.3 \pm 3.1$ | $64.2 \pm 3.2$ | $68.9 \pm 2.2$ |
| Cosine | Mean | all | 7 | $0.28 \pm 0.21$ | $78.3 \pm 3.2$ | $70.3 \pm 1.8$ | $85.9 \pm 4.5$ | $66.9 \pm 2.4$ | $73.4 \pm 1.6$ | $70.1 \pm 2.3$ | $62.3 \pm 2.3$ | $67.7 \pm 1.7$ |
| TracIn | Rank | 24-31 | 8 | $0.32 \pm 0.21$ | $73.1 \pm 3.3$ | $71.7 \pm 1.1$ | $68.4 \pm 7.1$ | $72.0 \pm 1.8$ | $75.5 \pm 1.4$ | $69.4 \pm 3.9$ | $63.3 \pm 3.5$ | $71.2 \pm 2.2$ |
| TracIn | Rank | CL | 9 | $0.32 \pm 0.21$ | $72.3 \pm 3.0$ | $72.0 \pm 1.6$ | $70.6 \pm 6.4$ | $72.3 \pm 1.9$ | $75.5 \pm 1.4$ | $69.5 \pm 3.4$ | $61.6 \pm 2.6$ | $71.2 \pm 2.7$ |
| TracIn | Mean | 16-23 | 10 | $0.34 \pm 0.22$ | $74.4 \pm 5.4$ | $70.5 \pm 3.8$ | $70.2 \pm 5.7$ | $70.7 \pm 3.0$ | $75.9 \pm 1.4$ | $71.6 \pm 2.7$ | $62.5 \pm 4.1$ | $70.7 \pm 4.0$ |
| DataInf | Mean | 24-31 | 10 | $0.31 \pm 0.20$ | $75.9 \pm 4.2$ | $69.1 \pm 4.8$ | $61.6 \pm 9.7$ | $71.4 \pm 2.2$ | $75.9 \pm 1.6$ | $70.8 \pm 2.8$ | $62.6 \pm 4.0$ | $70.4 \pm 2.4$ |
| DataInf | Mean | CL | 11 | $0.29 \pm 0.20$ | $76.1 \pm 3.2$ | $69.1 \pm 4.7$ | $64.7 \pm 9.2$ | $71.8 \pm 1.2$ | $75.8 \pm 1.6$ | $69.7 \pm 3.7$ | $62.2 \pm 3.9$ | $69.0 \pm 2.6$ |
| Cosine | Rank | 24-31 | 11 | $0.26 \pm 0.19$ | $78.5 \pm 1.9$ | $69.9 \pm 1.7$ | $64.2 \pm 10.4$ | $69.1 \pm 1.9$ | $73.0 \pm 1.9$ | $73.1 \pm 4.7$ | $66.0 \pm 3.2$ | $65.7 \pm 3.1$ |
| Cosine | Mean | 16-23 | 11 | $0.25 \pm 0.20$ | $76.0 \pm 4.0$ | $69.9 \pm 1.4$ | $77.3 \pm 9.0$ | $69.4 \pm 1.4$ | $73.5 \pm 1.7$ | $69.9 \pm 3.0$ | $63.8 \pm 2.6$ | $67.0 \pm 2.8$ |
| TracIn | Mean | CL | 12 | $0.27 \pm 0.19$ | $71.0 \pm 2.4$ | $70.3 \pm 4.2$ | $74.1 \pm 3.4$ | $70.8 \pm 2.6$ | $75.3 \pm 1.2$ | $68.6 \pm 3.2$ | $61.1 \pm 3.3$ | $70.3 \pm 3.6$ |
| Cosine | Vote | all | 12 | $0.24 \pm 0.19$ | $79.7 \pm 1.7$ | $70.1 \pm 1.8$ | $65.7 \pm 7.9$ | $66.9 \pm 2.4$ | $72.8 \pm 1.8$ | $74.7 \pm 3.5$ | $62.5 \pm 3.7$ | $65.9 \pm 3.0$ |
| TracIn | Mean | 24-31 | 13 | $0.27 \pm 0.20$ | $73.0 \pm 4.6$ | $70.1 \pm 3.5$ | $70.1 \pm 3.8$ | $70.7 \pm 3.1$ | $75.4 \pm 1.3$ | $70.0 \pm 2.6$ | $61.8 \pm 4.1$ | $70.4 \pm 3.1$ |
| Cosine | Mean | CL | 13 | $0.23 \pm 0.18$ | $78.5 \pm 2.6$ | $69.2 \pm 1.1$ | $66.2 \pm 10.1$ | $68.5 \pm 2.0$ | $72.6 \pm 1.4$ | $72.2 \pm 4.9$ | $65.5 \pm 3.2$ | $66.8 \pm 3.5$ |
| TracIn | Mean | all | 14 | $0.27 \pm 0.19$ | $72.1 \pm 2.8$ | $70.0 \pm 4.3$ | $73.9 \pm 2.1$ | $71.0 \pm 2.8$ | $75.5 \pm 1.1$ | $68.4 \pm 4.0$ | $61.2 \pm 3.8$ | $69.8 \pm 3.4$ |
| Cosine | Vote | 16-23 | 14 | $0.21 \pm 0.17$ | $79.2 \pm 2.0$ | $69.7 \pm 1.3$ | $63.6 \pm 7.1$ | $66.9 \pm 2.0$ | $72.8 \pm 1.7$ | $74.5 \pm 3.7$ | $62.5 \pm 2.9$ | $65.3 \pm 3.4$ |
| Cosine | Vote | CL | 14 | $0.17 \pm 0.14$ | $78.5 \pm 2.4$ | $69.2 \pm 1.2$ | $64.2 \pm 5.8$ | $66.9 \pm 1.8$ | $70.8 \pm 1.4$ | $74.1 \pm 3.5$ | $61.9 \pm 3.5$ | $65.0 \pm 2.6$ |
| Cosine | Mean | 24-31 | 14 | $0.16 \pm 0.15$ | $73.0 \pm 3.5$ | $69.9 \pm 1.4$ | $69.7 \pm 9.0$ | $68.8 \pm 1.4$ | $72.8 \pm 2.0$ | $67.7 \pm 3.0$ | $63.1 \pm 3.4$ | $64.3 \pm 2.8$ |
| Cosine | Vote | 24-31 | 14 | $0.15 \pm 0.13$ | $78.2 \pm 1.8$ | $69.6 \pm 1.6$ | $59.1 \pm 5.6$ | $67.6 \pm 2.1$ | $71.9 \pm 1.6$ | $73.6 \pm 4.1$ | $61.5 \pm 3.5$ | $63.5 \pm 2.6$ |
| Cosine | Mean | CL | 14 | $0.08 \pm 0.08$ | $70.4 \pm 2.0$ | $69.4 \pm 0.9$ | $57.2 \pm 10.0$ | $68.5 \pm 1.8$ | $71.7 \pm 1.5$ | $64.5 \pm 3.8$ | $62.6 \pm 4.2$ | $62.6 \pm 3.0$ |

# G    PROOF OF CANCELLATION EFFECT THEOREM

**Theorem G.1** (**Cancellation Can Improve Influence Estimation.**).  Let $X$ be a training set. Consider:

1. two samples $\bar{x}_1, \bar{x}_2 \in X$, where $\bar{x}_1$ is noisy and $\bar{x}_2$ is clean;

2. two parameter vectors $\theta$ and $\omega$ such that their cancellation scores satisfy $C(\theta) \ll C(\omega)$ with $C(\omega) \to \infty$ for $\{\bar{x}_1, \bar{x}_2\}$;

3. the TracIn influence scores $I_\theta$ (based on $\theta$ alone) and $I_{\theta,\omega}$ (based jointly on $\theta, \omega$);

4. influence score distance between $\bar{x}_1$ and $\bar{x}_2$ w.r.t. weights $\Theta$ and validation samples $X'$: $\Delta_\Theta I(\bar{x}_1, \bar{x}_2 \mid X') = |I(\bar{x}_1, X', \Theta) - I(\bar{x}_2, X', \Theta)|.$

Then there exists a validation point $\bar{x}_3$ such that:
$$\Delta I_{\theta,\omega}(\bar{x}_1, \bar{x}_2 \mid \bar{x}_3) > \Delta I_\theta(\bar{x}_1, \bar{x}_2 \mid \bar{x}_3),$$
i.e. the separation between noisy and clean samples is strictly larger under $I_{\theta,\omega}$, thereby showing that contrary to the claim of (Yeh et al., 2022), the inclusion of weights with high cancellation can *improve* influence estimates.

*Proof.*  Suppose, for contradiction, that no such validation point exists. Then for all $\bar{x}_3$,
$$\Delta I_{\theta,\omega} \leq \Delta I_\theta.$$

Let
$$a_i = \frac{\partial \ell(\bar{x}_i, \theta)}{\partial \theta}, \qquad b_i = \frac{\partial \ell(\bar{x}_i, \omega)}{\partial \omega}.$$

**Cancellation assumptions.** Since $C(\omega) \to \infty$, the gradients at $\omega$ nearly cancel:
$$b_1 b_2 < 0, \qquad |b_1 + b_2| = \varepsilon \to 0^+.$$
For $\theta$, cancellation is finite:
$$a_1 a_2 < 0, \qquad |a_1 + a_2| > 0.$$

**Influence relation.** From the TracIn definition and the cancellation assumptions, without loss of generality, one obtains
$$\Delta I_\theta = |a_3(a_1 - a_2)|, \ \Delta I_{\theta,\omega} = |\begin{bmatrix} a_3 \\ b_3 \end{bmatrix}^\top \begin{bmatrix} a_1 - a_2 \\ b_1 - b_2 \end{bmatrix}| = |a_3(a_1 - a_2)||(1 + \frac{b_1 - b_2}{a_1 - a_2}\frac{b_3}{a_3})|$$

$$C(\omega) \gg C(\theta) \implies \frac{b_1 - b_2}{\epsilon} \gg \frac{a_1 - a_2}{|a_1 + a_2|} \implies \Delta I_{\theta,\omega} \gg \Delta I_\theta \left(1 + \frac{\varepsilon}{|a_1 + a_2|} \cdot \frac{b_3}{a_3}\right).$$

**Contradiction.** By assumption $\Delta I_{\theta,\omega} \leq \Delta I_\theta$, which requires
$$\frac{\varepsilon}{|a_1 + a_2|} \cdot \frac{b_3}{a_3} < 0.$$
Since $\varepsilon > 0$, this inequality forces $a_3 b_3 < 0$.

Thus the assumption can only hold if *all* validation points $\bar{x}_3$ yield anti-aligned gradients under $\theta$ and $\omega$. Such universal anti-alignment is impossible, because $\theta$ and $\omega$ typically share co-directional gradients on at least some validation points. This yields a contradiction.

Therefore, there must exist a validation point $\bar{x}_3$ such that
$$\Delta I_{\theta,\omega} > \Delta I_\theta,$$
completing the proof.  □

**Remark.** This shows that contrary to the claim of (Yeh et al., 2022), inclusion of weights with high cancellation can *improve* discrimination in influence scoring, although the improvement depends on the choice of validation sample. Indeed, there are also cases where $\Delta I_{\theta,\omega} < \Delta I_\theta$, so the proposition in (Yeh et al., 2022) cannot be concluded in full generality.

## H    RANKING AND VOTING ON ROBERTA AND LLAMA

The figures 18, 19 show the change of Roberta-Large and Llama-3.2 1B performance when we filter the training set with rank and vote aggregations across different influence methods. Mistral 7B has the most benefit of these methods (see figure 3). Transparent distributions correspond to statistically insignificant (Wilcoxon p-value 0.1, as we have 10 runs only) or less than 1 percent change.

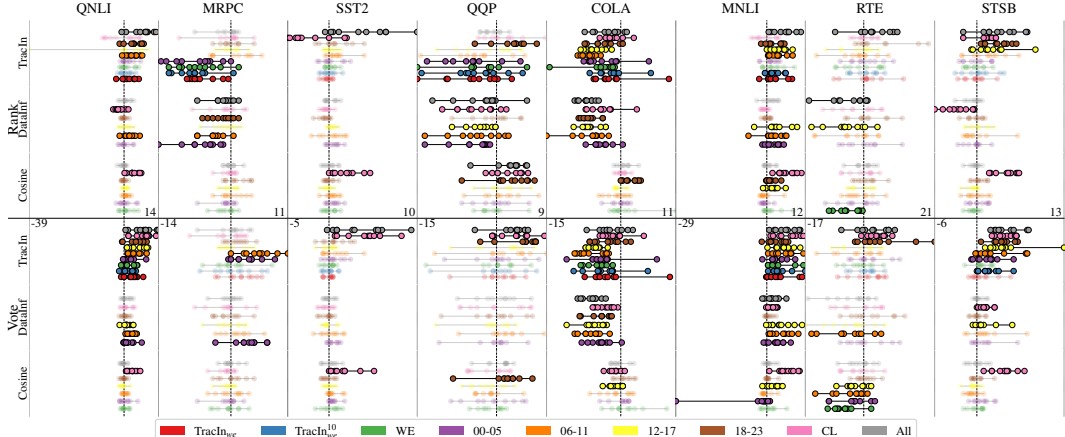

Figure 18: Roberta-Large accuracy change % with ranking and voting compared to mean aggregation. Voting has the most positive effect on the TracIn influence method, greatly boosting the performance of the last layers (CL) scores. COLA is the "toughest" dataset (accuracy drop across layers and methods). The mean aggregation is still better here.

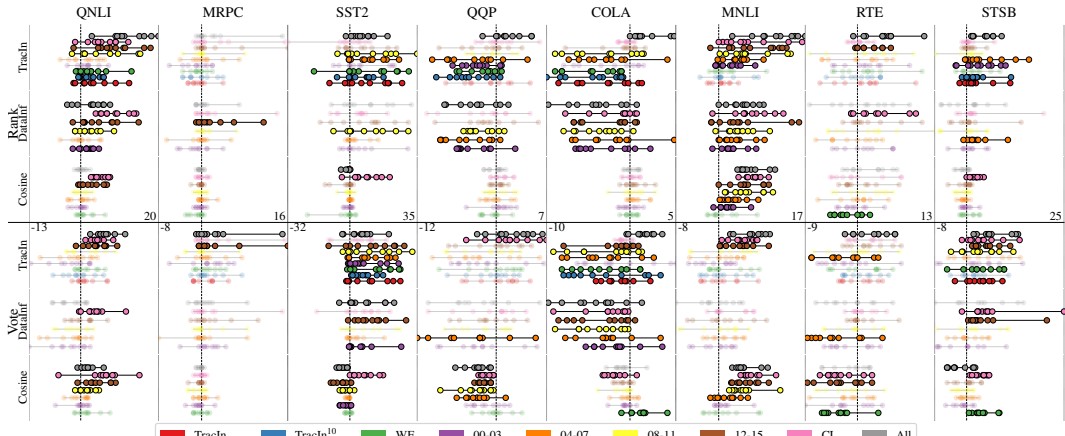

Figure 19: Llama-3.2 1B accuracy change % with ranking and voting compared to mean aggregation. SST2 and MNLI are greatly boosted, while COLA and QQP demonstrate drops.

For NLI tasks, both Ranking and Voting yield consistent improvements, while on CoLA and QQP we observe a performance decline.

# I NOISE INFLUENCE DISTRIBUTION ON MODEL LAYERS

This section presents noise distribution histograms for methods and layers. Every chart contains 10 quantiles, 450 samples each, depicting the relative noise count in the corresponding influence value range. The first and last quantiles represent the least and most influential samples, respectively.

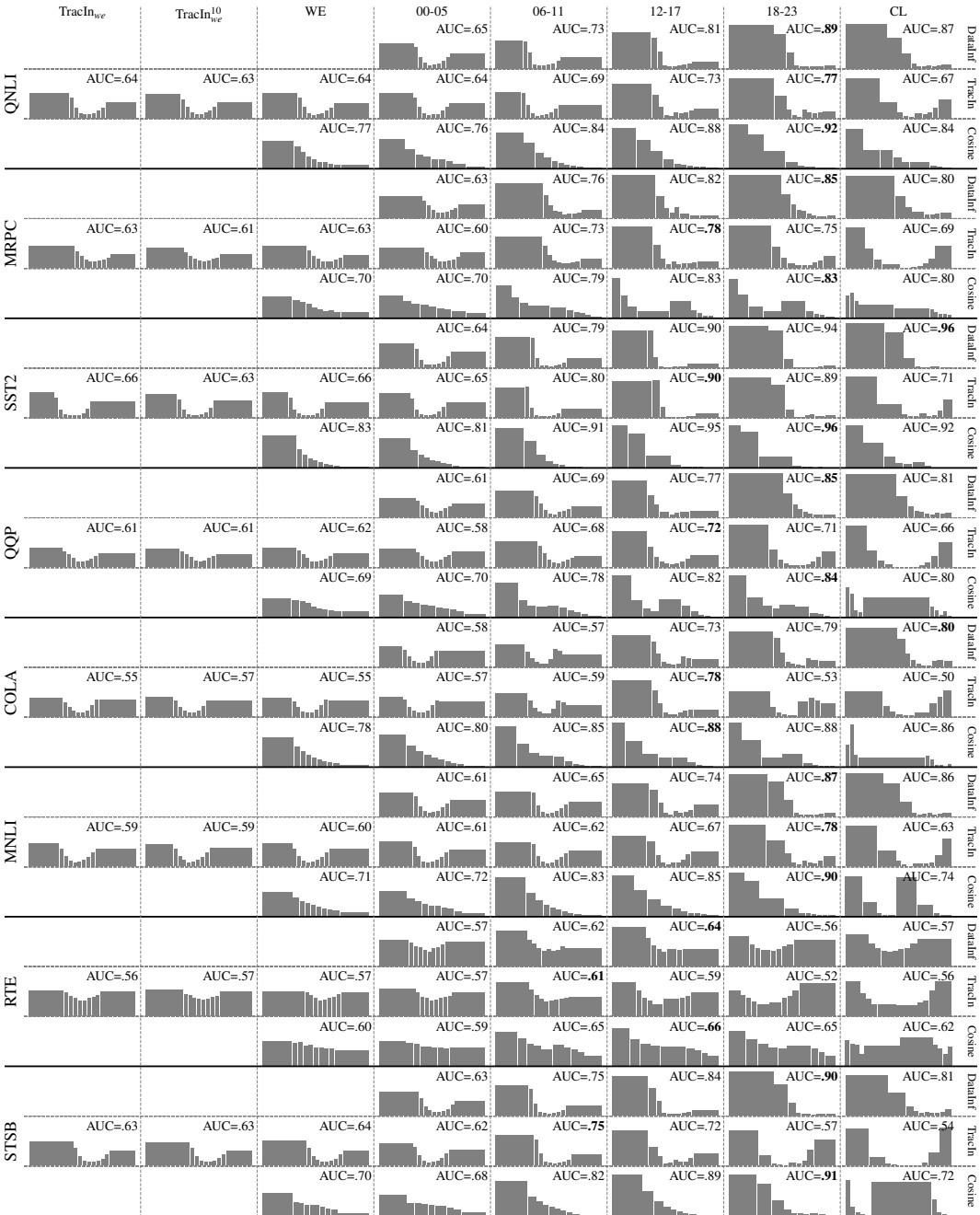

Figure 20: Roberta-Large noise distribution across mean-aggregated influence score range for influence methods, datasets, and selected layers. High bars represent a bigger percentage of noise in the corresponding quantile. The noise is usually in the first quantiles (least influential), as expected. Methods have the least discriminative power on the RTE dataset. TracIn frequently has high noise in the last quantile across layers.

An influence function is not useful for anomaly detection if it is closer to a uniform distribution. For selected methods, most mislabeled samples are concentrated at the beginning. However, some layers

and methods have a high influential noise level. For instance, TracIn demonstrates relatively high spikes on both sides across configurations. TracIn on `WE` and `CL` has a high percentage of influential noise. Generally, a performant attribution should strive to minimize the noise entropy, producing unimodal but not uniform distributions.

The following figures 20, 21, 22 present how mislabeled samples are spread in the influence scores. In the experiments, we discard the first two quantiles.

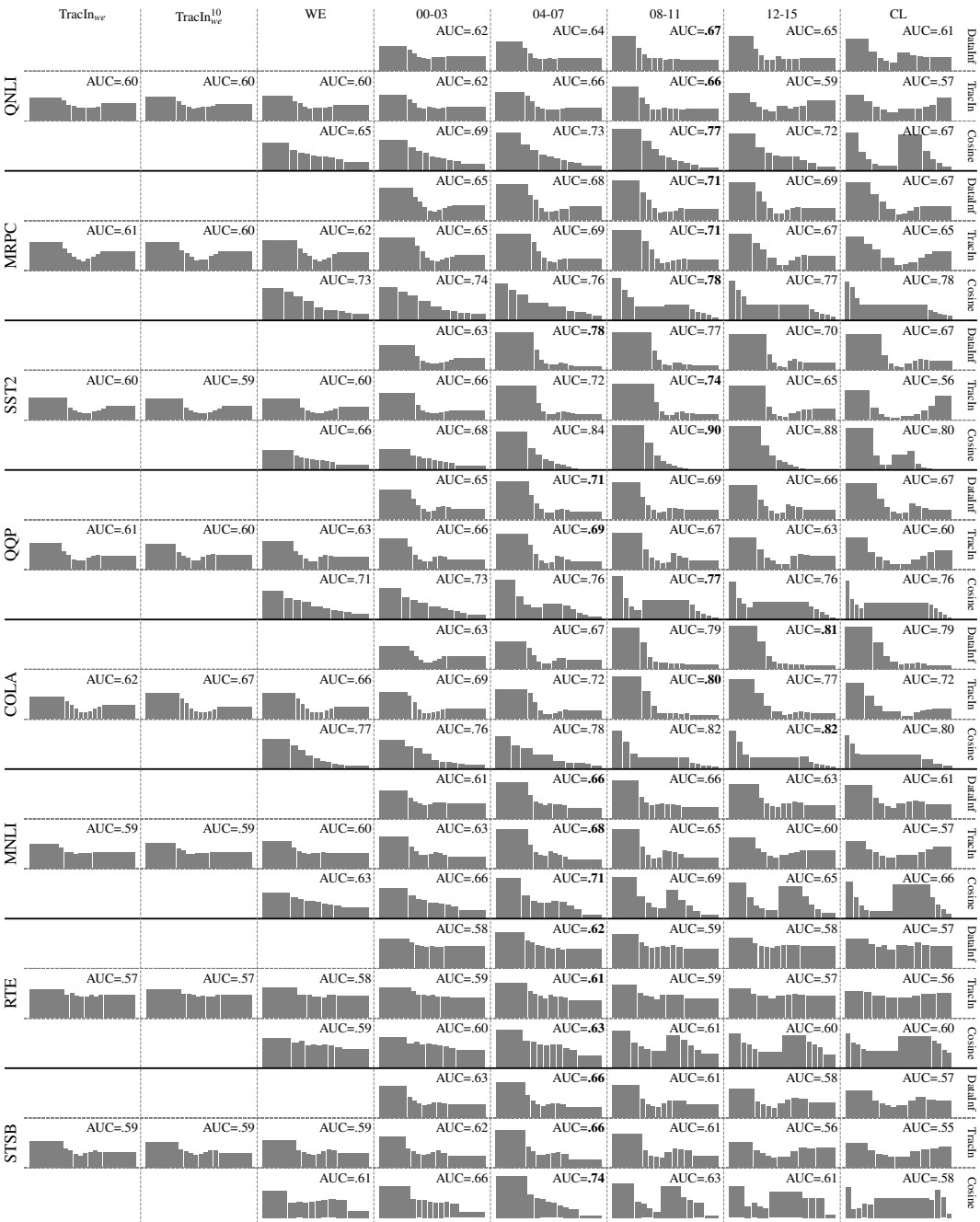

Figure 21: Llama-3.2 1B noise distribution across mean-aggregated influence score range for influence methods, datasets, and selected layers. Similar to Roberta-Large (figure 20), in many cases the noise in concentrated in the first quantile. However, Llama-3.2 1B scores demonstrate less discriminative power, effectively spreading the noise more across the range. In other words, noise has more influence on Llama model.

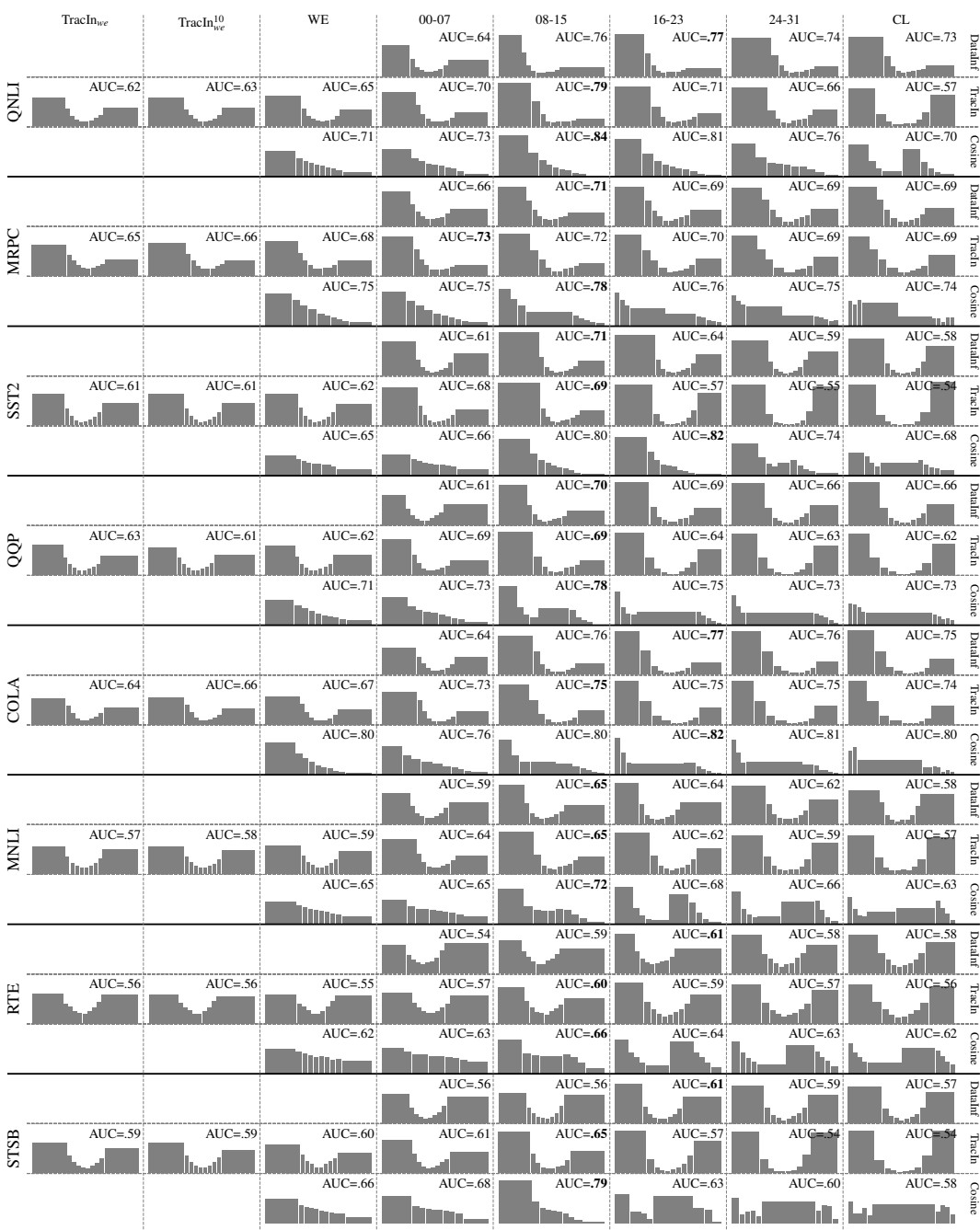

Figure 22: Mistral 7B noise distribution across mean-aggregated influence score range for influence methods, datasets, and selected layers. As with Llama (figure 21), we observe that noise has more influence on deeper Mistral. For instance, even for the DataInf method, we observe a high concentration of mislabeled samples in the last quantiles. Cosine similarity has a noise spike in the middle of influence range instead. The middle layers of the Mistral are less susceptible to the influence of noise across methods.

## J  INFLUENTIAL DATA IDENTIFICATION ON AUTOREGRESSIVE DATASETS

**Datasets**. To further validate the RQ2 (first is not better than last), we conduct the experiment on the autoregressive datasets introduced in Kwon et al. (2024): **Grammar**, **Math**, and **Math (With Reasoning)**. Unlike the settings used in the main text, where detrimental samples were synthetically injected, these datasets contain semantically related instruction categories. In this context, a desirable behavior is that training instructions exert a stronger influence on test samples from the same semantic category. In this way, we assess how well influence scores *recover meaningful relational structure* rather than merely detecting artificial noise. Such scenarios are highly relevant for practical applications, including knowledge probing, debugging data pipelines, and detecting harmful or misaligned training examples.

**New methods**. Additionally, we extend our experiments to incorporate more recent gradient- and activation-based influence estimation techniques. We report the following results for **Outlier Gradient** Chhabra et al. (2025), Kronfluence (**EKFAC**) Grosse et al. (2023), and **RepSim** Li et al. (2025). We detect gradient outliers by scoring training samples with OneClassSVM using an RBF kernel on a per-semantic-category basis. EKFAC scoring combines forward and backward signals to estimate influence. RepSim is based purely on hidden representations measured after each attention layer either on *last* sample token or averaged across all tokens of the sample (*mean*).

**Models**. The results are provided for **Qwen2.5-1.5B** (lr=3e-4) and **Mistral-7B-v0.3** (lr=1e-3).

**Metrics**. We use the evaluation metrics, **AUC** and **Recall**, from the original work Kwon et al. (2024). Higher values indicate bettser alignment between the sample ordering by the influence technique and the actual semantic category of the test samples.

**Conclusions**. Figures 23, 24 present the AUC and Recall across layers for Qwen, and 25, 26 - for Mistral. We observe that RQ2 conclusion still holds for new datasets.

**First**, early layers consistently exhibit weaker performance in AUC and Recall for both models across methods. Similarly, the last attention layers show a drop in AUC-Recall in the majority of cases. Unfortunately, the best-performing layers vary with model-dataset-method, but generally, we observe the performance maximum for 20-24 layers.

**Second**, for both Qwen and Mistral, the **LoRA B** module (attached to **value projections**) delivers the strongest performance across most configurations. Our **practical recommendation** from consistent observations across all evaluated models and datasets is to rely on influence scores computed on these modules. These weights yield the highest NDR, AUC, Recall, and downstream task performance.

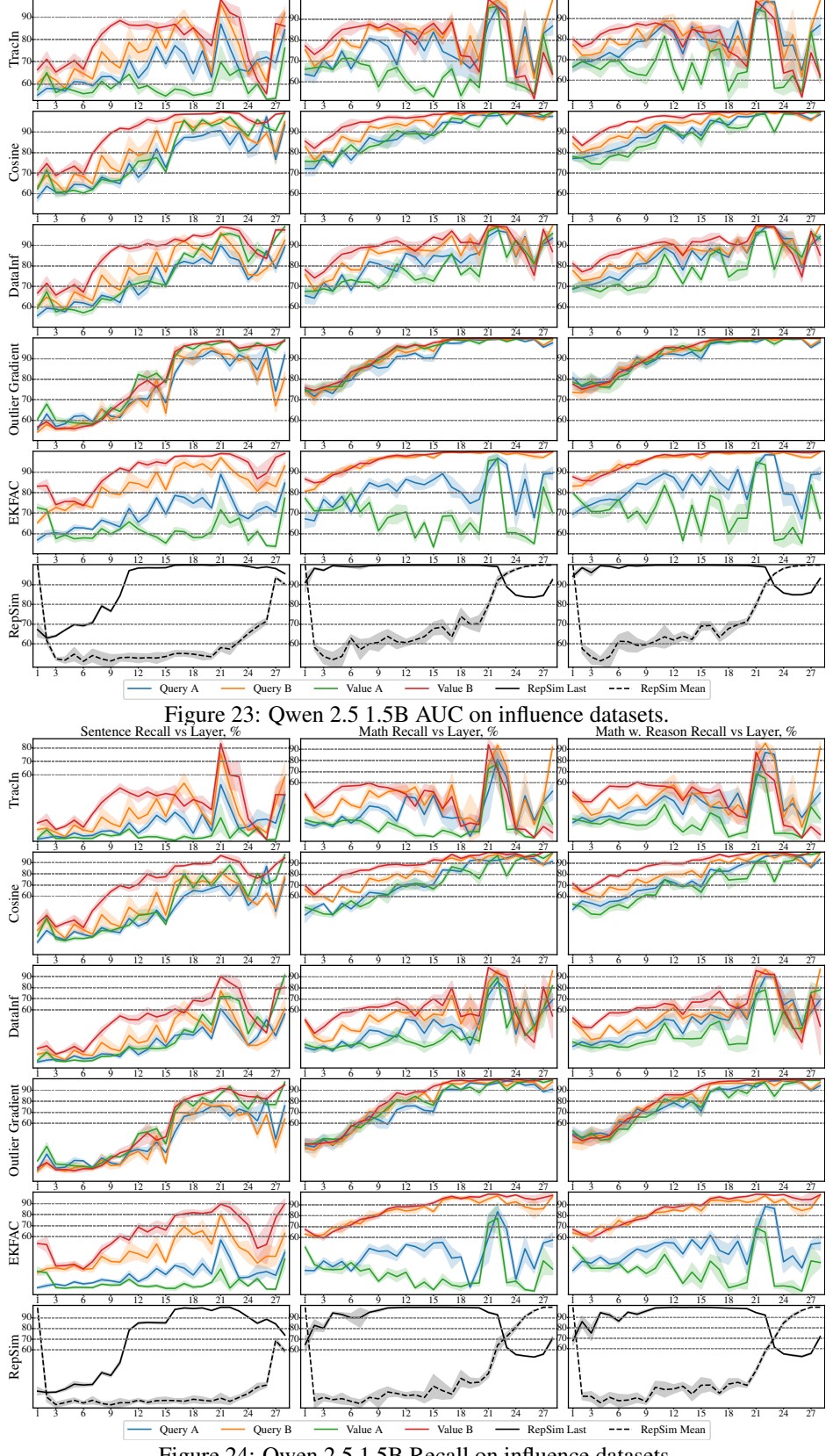

Figure 23: Qwen 2.5 1.5B AUC on influence datasets.

Figure 24: Qwen 2.5 1.5B Recall on influence datasets.

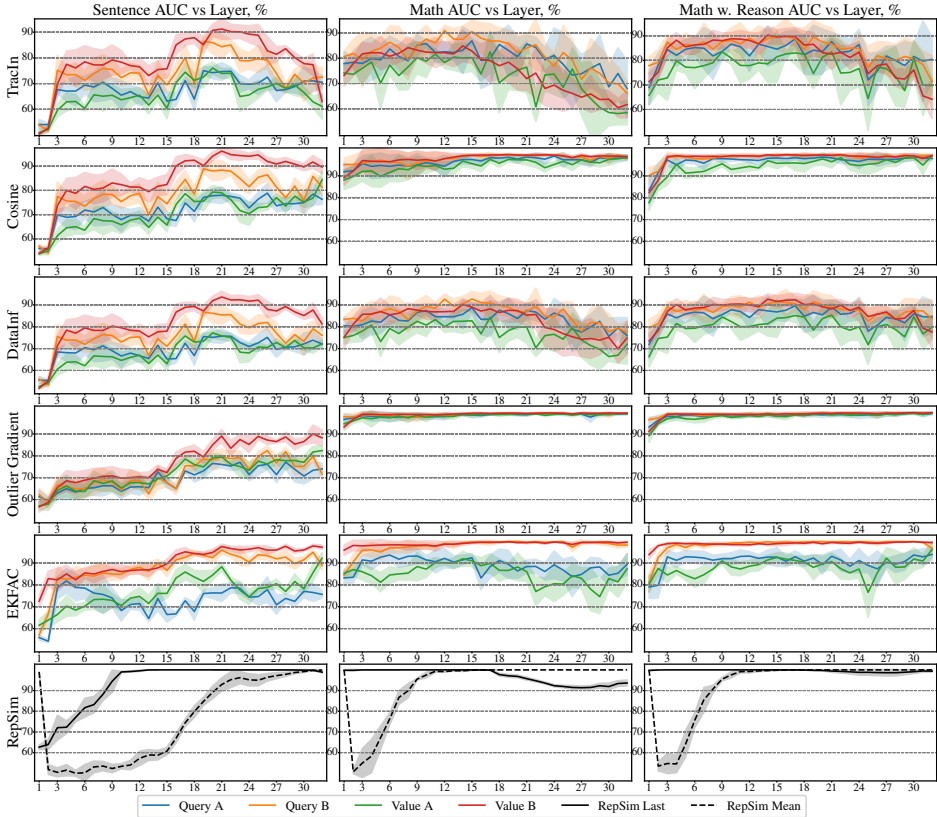

Figure 25: Mitral 7B AUC on influence datasets.

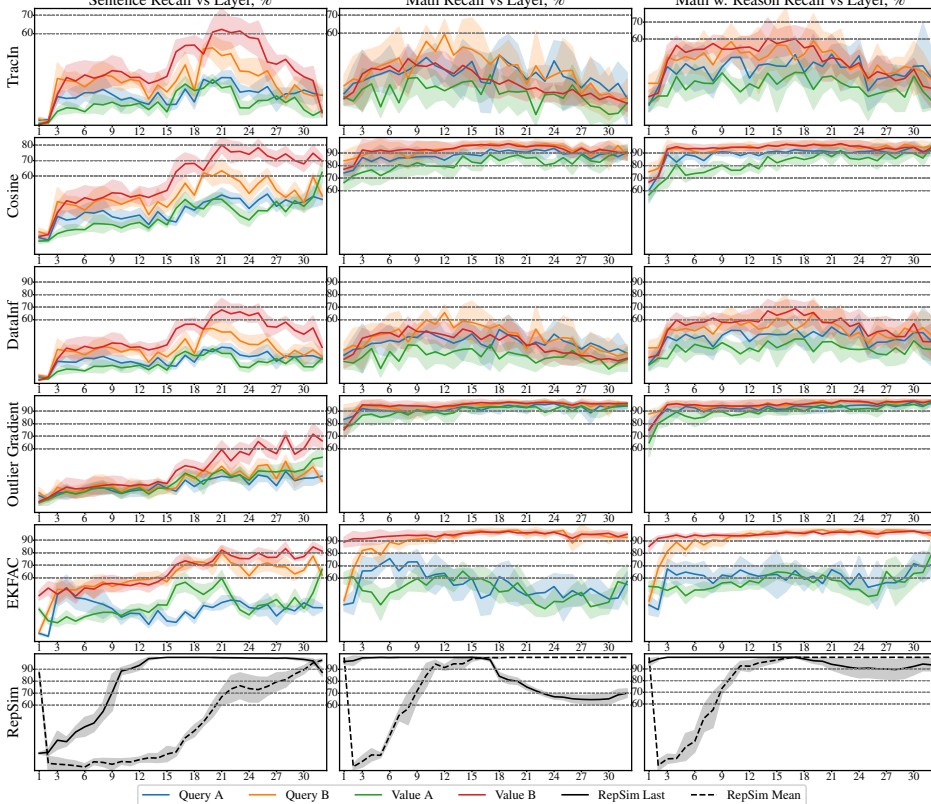

Figure 26: Mistral 7B Recall on influence datasets.

## K    SAMPLE INFLUENCE VARIATION ACROSS MODEL LAYERS

We provide the following visualization of the sample influence ranking trends as additional interpretation of *why middle layers have better performance*. Figures 27 and 28 present these ranking, where highest rank means highest influence score, for Roberta-Large and Qwen-2.5 1.5B, respectively, and three "kinds" of samples:

- **Top Ranked Noise Sample** - the noisy sample with the highest influence across layers;

- **Avg Noise** - the averaged rank of all noisy samples per layer;

- **Avg Benign** - the averaged rank of "clean" samples per layer.

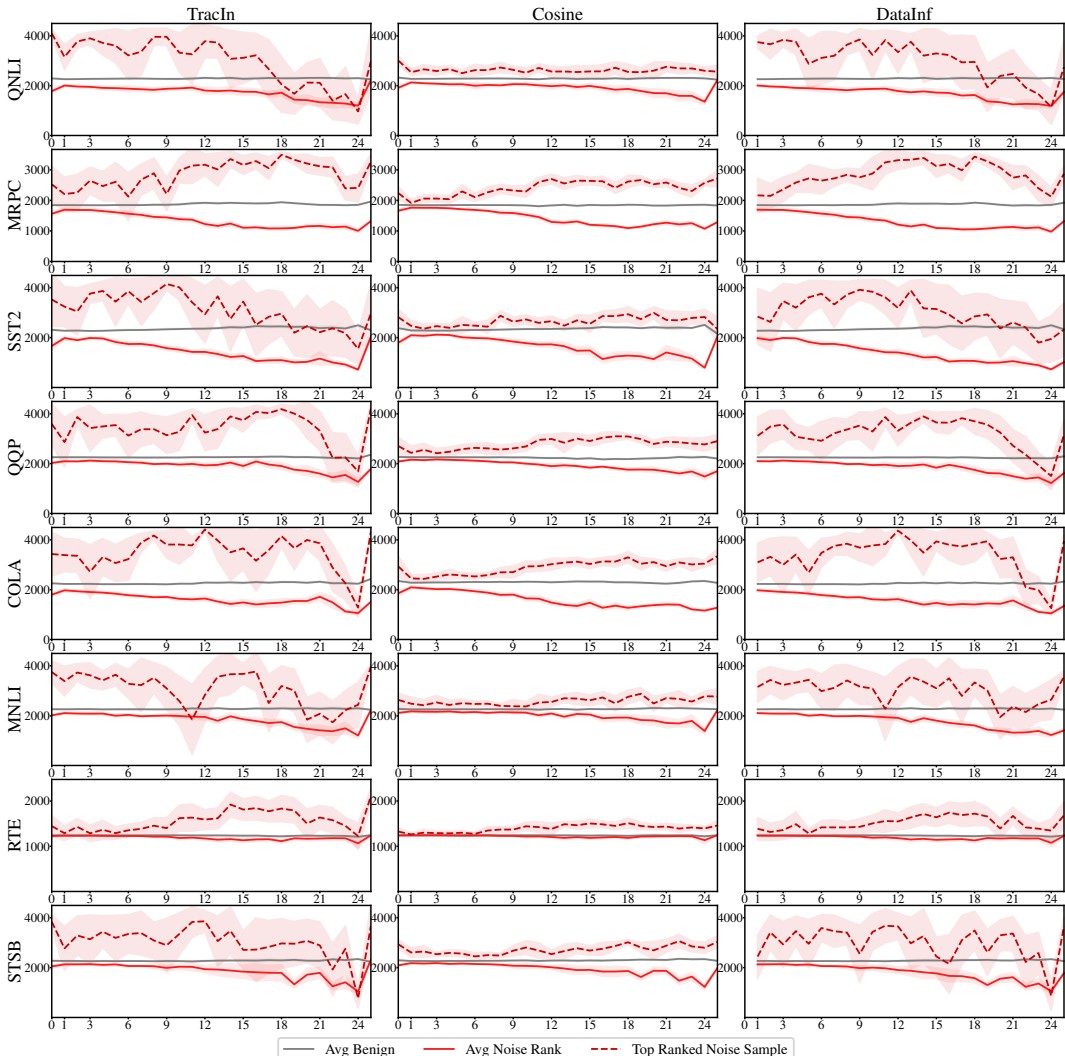

Figure 27: Roberta-Large sample influence rank variation with layer. Mean and 95% confidence intervals are presented for 10 random seeds.

**Conclusion**. We observe that the average noise influence rank drops with layer, i.e. noise becomes less influential. At the same time, the benign samples maintain the influence rank almost at same level. For Roberta-Large, the biggest difference between average ranks is observed at the last influence layers. This corresponds to the downstream performance maximum (Figure 6), where the best accuracy is also observed on later attention layers. Similarly, for Qwen-2.5 1.5B, the noise average rank has minimum on layers 15-20, where the model has also the best downstream performance (Table 3).

At the same time, for Qwen, in many cases, the top ranked noisy sample maintain high values of influence when the average noise rank has minimum, signifying the presence of the noise outliers that may affect the performance.

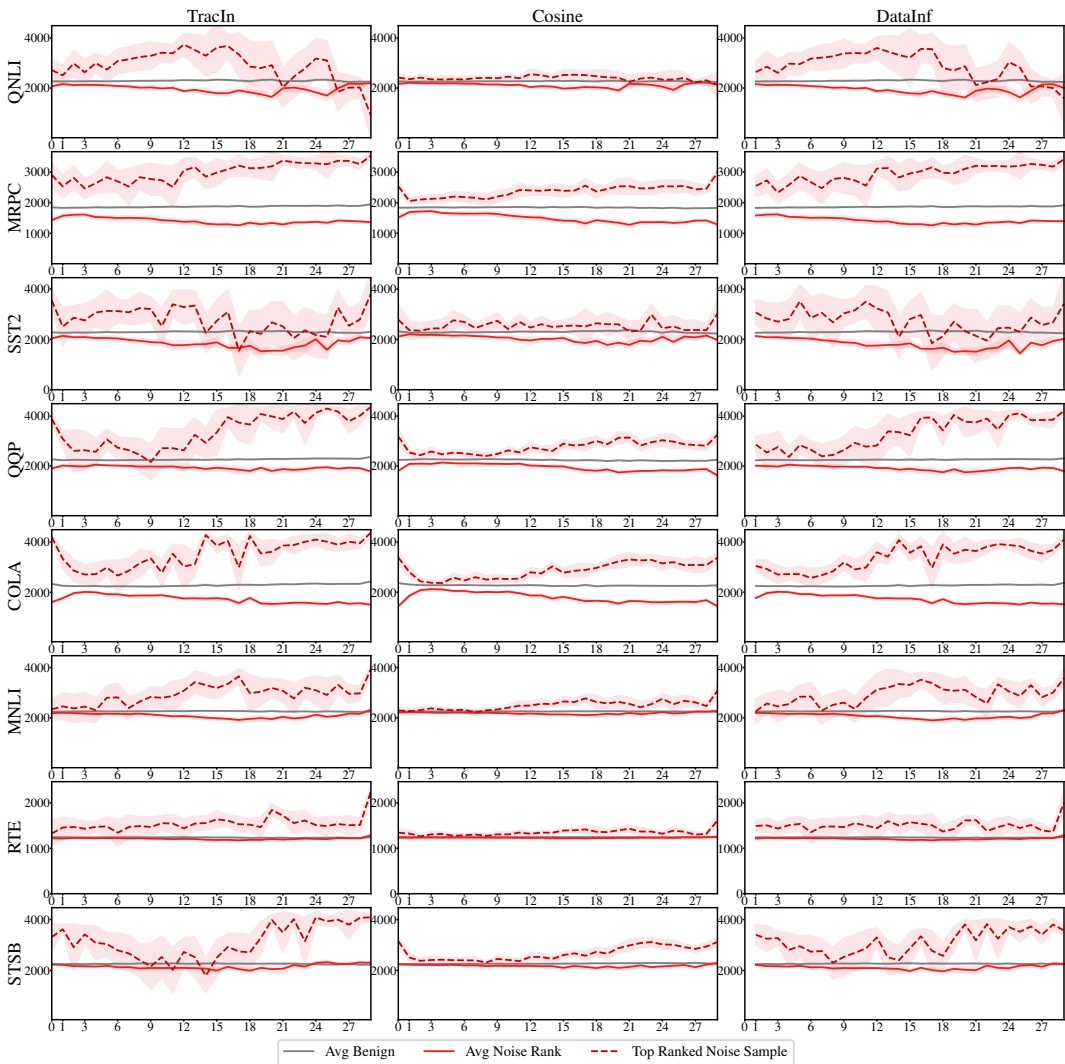

Figure 28: Qwen-2.5 1.5B sample influence rank variation with layer. Mean and 95% confidence intervals are presented for 10 random seeds.

## L   THE EFFECT OF VARYING THE NUMBER OF VOTES IN POSITIONAL VOTING

Formula 10 contains the hyperparameter $k$, the number of votes that each test sample assigns to training samples. The reasonable question is *how the noise detection rate changes with $k$* and *what $k$ would correspond to the best performance*? The following figure 29 presents the NDR for $k$ in range $[10, 100]$ with step 10, for different layers and per NN module of Qwen-2.5 1.5B. The NDR value is averaged across datasets and random seeds.

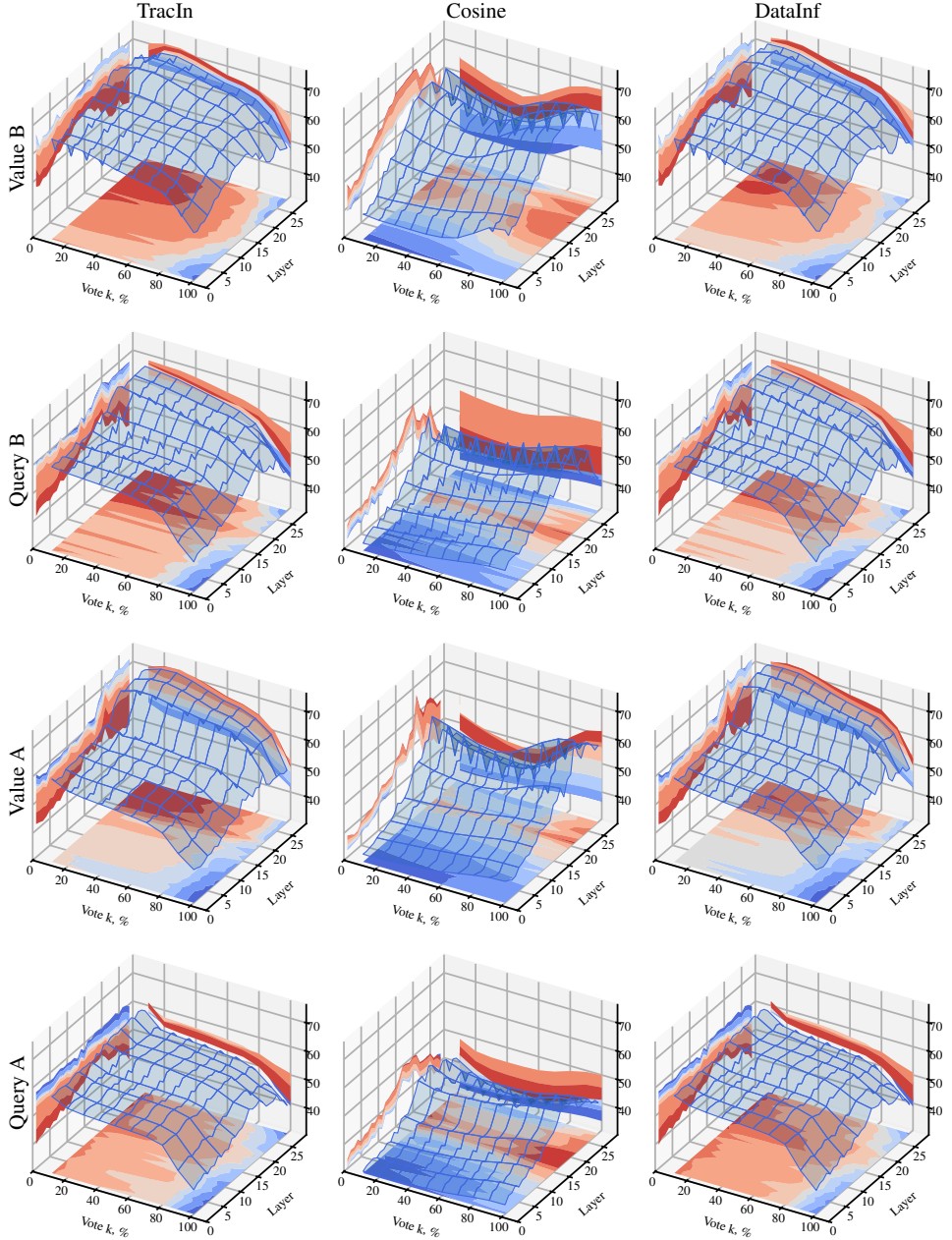

Figure 29: Qwen-2.5 1.5B NDR variation for different Vote $k$ and layer number.

**Conclusion**. NDR values show a clear dependence on the choice of $k$. The range $k \in [10, 50]$ yields the strongest performance for most settings. For DataInf and TracIn, NDR decreases monotonically beyond this range, suggesting reduced discriminative power at larger subset sizes. Cosine shows the opposite pattern, reaching its minimum at $k \approx 50$ and improving for larger $k$. We use $k = 30$ (filtering threshold) in all main experiments. Determining how the optimal $k$ scales with the level of injected noise remains an open question.

# M    LLAMA-3.2 1B INFLUENCE SCORE PERFORMANCE DISCUSSION

In Section 5.2, we compare the training data filtering by influence attribution with random baseline. We found, that particularly for LLaMA-3.2 1B, the influence functions fail to outperform random filtering. In this section, we verify if the best by NDR module (Figure 15), **LoRA B** attached to **value projection**, outperforms the baseline.

The artifacts observed for LLaMA-3.2-1B likely arise from its reduced depth (**16** attention layers) relative to larger models (Roberta-Large - **24**, Qwen-2.5 1B - **28**, Mistral 7B - **32**). The variation in best-performing layers across architectures indicates that reliable influence attribution requires not only well-chosen probes but also effective layer localization. This layer dependence parallels findings in other domains where the success of model interventions hinges on identifying the most relevant layers.

The following table 7 presents the accuracy on GLUE datasets after influence filtering by Value B score of **8th** and **9th** attention layers (best according to NDR). Results include **Mean**, **Rank**, and **Vote** aggregations.

Table 7: Llama-3.2 1B downstream performance of 8th and 9th attention layer influence scores measured for LoRA modules B that are attached to value projections.

| Method | Agg | Layer | Rank | QNLI | MRPC | SST2 | QQP | COLA | MNLI | RTE | STSB |
|---|---|---|---|---|---|---|---|---|---|---|---|
| Full | | | $1.20 \pm 0.80$ | $1.0 \pm 0.0$ | $1.0 \pm 0.0$ | $1.6 \pm 1.9$ | $1.0 \pm 0.0$ | $1.0 \pm 0.0$ | $1.0 \pm 0.0$ | $2.1 \pm 1.0$ | $1.0 \pm 0.0$ |
| Cosine | Vote | v-b-9 | $6.80 \pm 4.10$ | $6.0 \pm 4.5$ | $8.9 \pm 4.1$ | $12.8 \pm 2.2$ | $5.1 \pm 2.3$ | $\mathbf{4.6} \pm 1.5$ | $7.3 \pm 3.5$ | $6.6 \pm 3.7$ | $3.5 \pm 1.3$ |
| Cosine | Vote | v-b-8 | $6.90 \pm 4.40$ | $\mathbf{5.2} \pm 2.0$ | $11.2 \pm 5.1$ | $12.0 \pm 3.4$ | $5.5 \pm 2.2$ | $6.4 \pm 2.9$ | $5.2 \pm 3.3$ | $7.2 \pm 5.0$ | $\mathbf{2.8} \pm 1.0$ |
| DataInf | Vote | v-b-8 | $9.20 \pm 4.50$ | $13.1 \pm 4.9$ | $7.1 \pm 5.1$ | $9.8 \pm 2.5$ | $8.0 \pm 4.3$ | $6.4 \pm 3.7$ | $11.6 \pm 3.4$ | $10.6 \pm 4.8$ | $7.0 \pm 2.6$ |
| Random | | | $9.40 \pm 6.00$ | $8.3 \pm 4.8$ | $10.6 \pm 7.1$ | $13.3 \pm 5.1$ | $\mathbf{3.8} \pm 2.1$ | $6.8 \pm 3.2$ | $14.5 \pm 2.2$ | $14.4 \pm 6.5$ | $3.6 \pm 1.4$ |
| DataInf | Vote | v-b-9 | $9.80 \pm 5.20$ | $16.0 \pm 4.0$ | $5.5 \pm 5.0$ | $8.8 \pm 3.3$ | $7.6 \pm 4.8$ | $5.2 \pm 3.0$ | $15.0 \pm 3.3$ | $11.0 \pm 3.1$ | $9.2 \pm 3.3$ |
| TracIn | Vote | v-b-8 | $9.90 \pm 4.80$ | $14.6 \pm 3.0$ | $7.6 \pm 6.6$ | $8.6 \pm 4.4$ | $9.4 \pm 4.0$ | $7.2 \pm 2.5$ | $11.8 \pm 4.3$ | $11.2 \pm 5.5$ | $8.9 \pm 3.3$ |
| Cosine | Rank | v-b-8 | $10.40 \pm 5.10$ | $6.6 \pm 3.7$ | $15.8 \pm 3.3$ | $8.1 \pm 5.4$ | $12.2 \pm 3.5$ | $14.4 \pm 2.5$ | $\mathbf{4.6} \pm 2.8$ | $9.2 \pm 5.3$ | $12.2 \pm 1.7$ |
| Cosine | Rank | v-b-9 | $10.60 \pm 5.00$ | $8.8 \pm 4.4$ | $12.7 \pm 5.4$ | $6.8 \pm 4.1$ | $12.0 \pm 3.4$ | $12.4 \pm 3.2$ | $7.6 \pm 4.1$ | $8.4 \pm 4.7$ | $16.4 \pm 3.9$ |
| TracIn | Vote | v-b-9 | $10.60 \pm 5.60$ | $17.4 \pm 3.2$ | $\mathbf{5.2} \pm 4.4$ | $9.4 \pm 4.2$ | $9.2 \pm 4.8$ | $7.0 \pm 3.0$ | $15.2 \pm 4.5$ | $12.0 \pm 4.9$ | $9.0 \pm 4.0$ |
| Cosine | Mean | v-b-8 | $10.70 \pm 5.20$ | $10.5 \pm 5.6$ | $15.0 \pm 4.6$ | $5.3 \pm 3.3$ | $13.4 \pm 3.3$ | $15.2 \pm 3.0$ | $9.5 \pm 4.2$ | $7.4 \pm 5.6$ | $9.2 \pm 3.1$ |
| DataInf | Mean | v-b-8 | $11.10 \pm 5.60$ | $10.4 \pm 4.3$ | $12.6 \pm 5.3$ | $13.6 \pm 6.1$ | $9.9 \pm 6.6$ | $8.8 \pm 5.2$ | $12.6 \pm 5.1$ | $9.3 \pm 6.7$ | $11.8 \pm 5.5$ |
| Cosine | Mean | v-b-9 | $11.40 \pm 5.40$ | $10.2 \pm 2.4$ | $13.4 \pm 4.2$ | $\mathbf{5.0} \pm 3.7$ | $13.6 \pm 3.2$ | $16.5 \pm 2.9$ | $12.5 \pm 5.5$ | $\mathbf{6.2} \pm 5.4$ | $13.6 \pm 4.7$ |
| DataInf | Rank | v-b-8 | $11.70 \pm 4.60$ | $8.6 \pm 4.4$ | $11.4 \pm 4.5$ | $10.6 \pm 4.3$ | $16.4 \pm 1.6$ | $16.2 \pm 2.5$ | $7.4 \pm 3.0$ | $13.0 \pm 3.8$ | $10.1 \pm 3.4$ |
| TracIn | Rank | v-b-8 | $12.70 \pm 4.80$ | $10.7 \pm 6.0$ | $12.2 \pm 3.1$ | $12.0 \pm 3.8$ | $17.0 \pm 2.3$ | $16.4 \pm 3.1$ | $7.0 \pm 4.0$ | $14.0 \pm 4.3$ | $12.6 \pm 3.1$ |
| TracIn | Mean | v-b-8 | $12.80 \pm 5.70$ | $13.2 \pm 4.8$ | $13.6 \pm 5.5$ | $16.6 \pm 3.4$ | $12.0 \pm 7.0$ | $8.0 \pm 5.2$ | $14.3 \pm 5.5$ | $10.4 \pm 6.0$ | $14.6 \pm 5.1$ |
| DataInf | Rank | v-b-9 | $12.80 \pm 4.50$ | $10.8 \pm 4.9$ | $11.7 \pm 4.1$ | $12.4 \pm 4.9$ | $15.1 \pm 1.7$ | $15.8 \pm 3.0$ | $8.4 \pm 4.5$ | $12.6 \pm 5.0$ | $15.2 \pm 3.0$ |
| DataInf | Mean | v-b-9 | $13.10 \pm 5.60$ | $11.6 \pm 6.1$ | $12.4 \pm 3.9$ | $16.0 \pm 7.3$ | $9.0 \pm 4.9$ | $10.6 \pm 4.9$ | $16.2 \pm 5.2$ | $13.2 \pm 4.7$ | $16.0 \pm 3.6$ |
| TracIn | Rank | v-b-9 | $14.00 \pm 4.90$ | $11.5 \pm 5.8$ | $11.1 \pm 3.3$ | $12.2 \pm 5.9$ | $17.2 \pm 3.3$ | $17.9 \pm 2.0$ | $10.5 \pm 4.3$ | $15.2 \pm 4.7$ | $16.5 \pm 2.7$ |
| TracIn | Mean | v-b-9 | $14.80 \pm 5.30$ | $15.6 \pm 5.7$ | $11.2 \pm 5.1$ | $15.2 \pm 7.1$ | $12.6 \pm 5.9$ | $13.0 \pm 4.3$ | $17.8 \pm 4.2$ | $16.0 \pm 3.1$ | $17.0 \pm 3.4$ |

We observe that Cosine scores (gradient similarities) outperform Random with Vote aggregation ($k = 20$) on selected modules. The differences are statistically significant. Wilcoxon test p-value is 0.0072 for configuration (Cosine, Vote, v-b-9) and Random; p-value is 0.0085 for (Cosine, Vote, v-b-8) and Random. At the same time, (DataInf, Vote, v-b-8) and Random have the p-value 0.47, signifying that the null hypothesis of their similarity cannot be rejected. At the same time, Mean aggregation cannot outperform the baseline even on the best NDR layers.

**Conclusion**. With the new Vote aggregation strategy, influence scoring outperforms the baselines on locations that perform best according to NDR. Our practical recommendation for LLaMA-3.2-1B, and consistently across the evaluated models, is to rely on influence scores computed on the LoRA B value projection modules. These weights yield strong NDR performance also on autoregressive datasets (Appendix J).

