# OpenReview forum: "First is Not Really Better Than Last: Evaluating Layer Choice and Aggregation Strategies in Language Model Data Influence Estimation"
_ICLR.cc/2026/Conference — ICLR 2026 Poster_

### Official Review · Reviewer_z9e9 · 2025-10-23

**Soundness:** 3
**Presentation:** 3
**Contribution:** 3
**Rating:** 6
**Confidence:** 4

**Summary:**

In this work, the authors study one particular problem in influence function approximation, i.e., which layer to focus on. They invalidated the existing claim that the first layer is the best and proposed some method to select the layer (based on two newly proposed metrics NDR and AUC). Multiple empirical studies are conducted to answer 4 research questions.

**Strengths:**

On the positive side, I enjoy reading the draft for the following reasons.

First, the problem of identifying influential samples is an interesting and important one and in fact existing approaches are less than satisfactory.

Second, some of the results are interesting and relevant, such as those for answering RQ1 and RQ2 (the empirical part).

**Weaknesses:**

On the less positive side, the draft can be improved from the following aspects.

First, the authors seem to be unaware of some recent work questioning the usefulness of the influence function for LLMs (“Do Influence Functions Work on Large Language Models?”). If we were to trust the empirical results (which is not unlike some of your observations) and reasoning in that paper, there is in fact little point in patching this existing influence function estimation method. Kindly discuss.

Second, some of the empirical results and conjectures are rushed and may not be reliable from what I read, given the many variables for such studies, such as the data distribution and the optimization algorithm and so on.

The following are some detailed comments.

Section 2: RELATED WORKS

Comment: Some important recent work is missing such as “Do Influence Functions Work on Large Language Models?” (EMNLP 2025).

Page 4: “Additionally, our proposed Rank method ignores incorrectly predicted validation samples …”

Comment: Kindly explain why.

Page 5: “[RQ4]. How reliably can … ”

Comment: Without first introducing what NDR is, stating this RQ here seems not useful for the readers.

Page 7: Figure 2

Comment: How reliable are these results given the many details involved in this kind of experiment (such as the data distribution, the performance of the optimization algorithm)?

Page 7: “For Llama, influence functions perform worst among all settings: none of the
configurations surpass the uniform removal baseline …”

Comment: This perhaps questions the usefulness of influence functions for LLMs fundamentally.

Page 8: “DataInf and Cosine frequently outperform TracIn, particularly on early–middle attention layers of deeper models, suggesting that these layers encode representations that are most informative for noise filtering.”

Comment: Given the limited scale of the experiments and the complexity of the problem, it is not clear how strong this empirical evidence is.

Page 9: “We validate the hypothesis that NDR and AUC can serve as reliable proxy metrics
for influence estimation by …”

Comment: Kindly explain why intuitively these are good estimations.

**Questions:**

How do you justify the relevance of your work in the presence of the recent observation made in "Do influence functions work for LLMs"?

---

> ### Author Response · Authors · 2025-11-21
> **Response to reviewer z9e9 [1/4]**
>
> We would like to thank the reviewer for the detailed feedback and for helping strengthen our contributions and work. Here we provide the response to the comments raised.
>
> > First, the authors seem to be unaware of some recent work questioning the usefulness of the influence function for LLMs (“Do Influence Functions Work on Large Language Models?”). If we were to trust the empirical results (which is not unlike some of your observations) and reasoning in that paper, there is in fact little point in patching this existing influence function estimation method. Kindly discuss.
>
> We carefully considered the work of Li et al. [1] and will acknowledge and discuss it in the paper revision. Li et al. demonstrate that Hessian-Free, DataInf, and LiSSA perform poorly across tasks such as harmful data identification, class attribution, and backdoor poison detection. Authors apply influence functions to all model layers with LoRA adapters.
>
> In our work, we explore an adjacent but different research goal, of how different model layers contribute to influence analysis, and how these scores should be aggregated across layers for optimal estimation performance, through:
>
> * **RQ2** examines how influence varies with layers (**localization**).
> * **RQ3** studies how to combine independent valuations more effectively (**aggregation**).
>
> More generally, for LLMs, these goals and RQs are closely parallel to those in adjacent but orthogonal work in LLM layer-dependence, such as for model knowledge editing (KE). In KE, locate-then-edit methods explicitly identify and modify the most causally relevant layers [2–5]. Our findings (while derived from the perspective of influence functions) are consistent with this literature, which also reports stronger causal-tracing signals in middle layers. More importantly, our work contrasts with existing conclusions about layer-importance in LLM influence analysis (Yeh et al. [6]), finding novel theoretical and empirical evidence for why middle/later layers are better suited for this analysis than the first few layers.
>
> In this manner, our work actually **augments and complements** the insights of [1], rather than detracting from them. Our insights on localization and aggregation help better understand and interpret influence functions in LLMs, which have been applied across various tasks with varying success  [7-11]. In sum, our work is situated similarly to [1] and other works focusing on better understanding the usage of influence functions in LLMs. These insights and ideas can help develop better influence functions in the future that are even more robust for usage in LLMs.
>
> > Second, some of the empirical results and conjectures are rushed and may not be reliable from what I read, given the many variables for such studies, such as the data distribution and the optimization algorithm and so on.
>
> We understand the concern of the reviewer, but would like to assure them that we have controlled for these variables (optimization algorithm, data distribution, etc.), and would like to emphasize that all our results obtained are scientifically and statistically reliable, in multiple ways, as delineated below:
>
> * To ensure robustness, we ran each experimental configuration (model, task, influence method, layer, module, aggregation method) with **10 random seeds**.
> * For instance, the estimated number of fine-tuning after filtering runs is ~**17280** (4 model, 8 datasets, 3 main infl methods + 2 hf variations, 6 groups of layers, 3 aggregation methods, 10 seeds).
> * We enforce **deterministic behavior** in PyTorch for reproducible training trajectories (Appendix A).
> * The experiment is split into **5 stages**, a new stage (influence calculation, fine-tuning after filtering) starts with the same checkpoint/data for every tried configuration, which creates **fair conditions** for learning trajectory comparison.
> * During the rebuttal period, we undertook a recollection of data for LLaMA 3.2. 1B with new checkpoint initializations and were able to  **reconfirm the same trends** (NDR across layers https://anonymous.4open.science/r/nn-infl-data-6BCA/img/llama-layer-ndr.png
> is similar to Appendix E Figure 14 page 18).
> * Observed differences were statistically validated using **pairwise Wilcoxon tests** and the more conservative **Friedman test** (Llama **p-value 3e-70**) with post-hoc Nemenyi analysis.
> * Training procedure is consistent across configurations: **AdamW optimizer** with a cosine learning rate schedule and warm-up.
> * **Learning rates** were chosen based on common values in prior work (e.g., DataInf [12]) or suggested by the model authors.

---

> > ### Comment · Reviewer_z9e9 · 2025-11-26
> >
> > Thanks for the response. It however seems that you misunderstand the point of [1] - their final point is that the very idea of influence function is wrong (i.e., model parameter change does not quantify model behavior change).

---

> ### Author Response · Authors · 2025-11-21
> **Response to reviewer z9e9 [2/4]**
>
> To address variability in data distribution, we conducted experiments on **8 GLUE datasets** and extended the study to **3 autoregressive datasets** from Kwon et al. [12] with new influence attribution methods.
>
> * Qwen AUC: https://anonymous.4open.science/r/nn-infl-data-6BCA/infl-ds/qwen/autoreg-ds-auc-full.pdf
> * Qwen Recall: https://anonymous.4open.science/r/nn-infl-data-6BCA/infl-ds/qwen/autoreg-ds-recall-full.pdf
> * Mistral AUC: https://anonymous.4open.science/r/nn-infl-data-6BCA/infl-ds/mistral/autoreg-ds-auc-full.pdf
> * Mistral Recal: https://anonymous.4open.science/r/nn-infl-data-6BCA/infl-ds/mistral/autoreg-ds-recall-full.pdf
>
> These new results are consistent with the observations reported in the manuscript: first layers do not outperform last layers in influence estimation.
>
> * Early layers consistently show suboptimal performance in distinguishing beneficial samples.
> * Final layers are also frequently a suboptimal choice for most methods and tasks.
> * The LoRA module B, attached to the v_proj, again demonstrates the strongest and most stable performance in terms of AUC, and Recall.
>
> > Comment: Some important recent work is missing such as “Do Influence Functions Work on Large Language Models?” (EMNLP 2025).
>
> We thank the reviewer for pointing this out, and we apologize for the oversight in referencing it. Following the reviewer’s suggestion, we will definitely include this paper in the Related Work section, as it directly motivates our explorative study and highlights the need to investigate which parts of the model and which influence methods provide meaningful signals. We will also discuss its findings in this section.
>
> > Page 4: “Additionally, our proposed Rank method ignores incorrectly predicted validation samples …”
> Comment: Kindly explain why.
>
> 1. The decision to ignore incorrectly predicted validation samples in Rank method was motivated by preliminary experiments exploring various aggregation strategies and partitions of train/validation sets based on labels or predictions. Our observations showed that both Vote and Rank methods achieve higher NDR scores when incorrectly predicted validation samples are excluded, with Rank performing well when combined with cosine gradient similarity. For clarity and conciseness, in the manuscript, we only reported the two best-performing aggregation strategies so as not to add more density to our findings.
>
> 2. However, based on the reviewer’s excellent point, we provide additional experiments with other aggregating strategies from preliminary experiments (tables are too big to be placed inline, therefore, we share the links):
>
>     * NDR ranks on Roberta: https://anonymous.4open.science/r/nn-infl-data-6BCA/ndr-aggs/roberta-ndr.pdf
>     * NDR ranks on QWEN: https://anonymous.4open.science/r/nn-infl-data-6BCA/ndr-aggs/qwen-ndr.pdf
>
>     In these  NDR ranking tables we tested different aggregation approaches across models.
>
>     * Vote2-c and Rank-c variants were consistently most effective, we report  them in the manuscript.
>     * Suffix “-c” denotes aggregation using only those validation samples that the model correctly predicted in training sample valuations.
>     * We considered the strategies that are more robust to influence score outliers: median, majority, minimal scoring, etc.
>     * Strategies are varied by train-validation set partitioning based on prediction score, gold label, logit distance.
>
> > Comment: Without first introducing what NDR is, stating this RQ here seems not useful for the readers.
>
> We thank the reviewer for this suggestion. We agree that mentioning NDR at this point may be confusing and in the revised manuscript, we can easily rephrase the RQ4 explanation to introduce NDR first by providing a clearer and self-contained description first.
>
> > Comment: How reliable are these results given the many details involved in this kind of experiment (such as the data distribution, the performance of the optimization algorithm)?
>
> To enhance/reiterate previously provided arguments regarding reliability of the results in the context of mentioned figure, we provide the following explanation:
>
> * Each bar in Figure 2 represents the results of **10 random runs of a single setup** and serves as an illustration of the distribution of observed performance.
> * We randomly subsample each GLUE task dataset per seed: 4,500 training and 500 influence samples. Therefore, the variations shown in Figure 2 also reflect **the effects of data distribution** across tasks.
> * We conducted the aforementioned **statistical tests** to confirm significance.
>
> In the revised manuscript, we will update the figure title to explicitly indicate the statistical significance validation.

---

> ### Author Response · Authors · 2025-11-21
> **Response to reviewer z9e9 [3/4]**
>
> > Page 7: “For Llama, influence functions perform worst among all settings: none of the configurations surpass the uniform removal baseline …”
> Comment: This perhaps questions the usefulness of influence functions for LLMs fundamentally.
>
> Thank you for the insightful question. While the direct applicability of influence functions in LLMs is beyond the scope of our paper and RQs, we make several fundamental observations on the overall influence trends across models, and more specifically, LLaMA-3.2 1B, below:
>
> **First**, we would like to emphasize that many of the presented measurements demonstrate highly consistent patterns across models, including LLaMA-3.2 1B. Specifically, we observe the following trends:
>
> * Downstream task performance trends (Appendix B, Figure 6) demonstrate spikes on **middle layers**.
> * Early and middle layers groups ranked higher than others (Appendix C, Table 3) across all models.
> * LoRA module v_proj B influences have higher NDR (Appendix E, Figure 14) across models.
> * Ability of Rank and Vote to improve the influence on many layers and tasks (Appendix H, Figure 18).
> * Similarities in how layers correlate in terms of influence score assignment to training samples (Appendix D, Figure 12).
>
> **Second**, with regards to LLaMA-3.2 1B, we hypothesize that the observed artifacts for the LLaMA-3.2 1B model may stem from its architectural differences, including its substantially smaller number of attention layers (**16**) compared to Qwen (**28**) and Mistral (**32**). The fact that the best-performing layers differ across models suggests that, in practice, achieving reliable influence attribution may require not only well-chosen probing samples but also an effective **weight-localization strategy** that identifies the layers most suitable for attribution. In fact, as mentioned before, this aspect of layer dependence in LLMs is observable in other adjacent but orthogonal work such as knowledge editing [2-5], which finds middle layers are better suited for editing knowledge. This finding of our work also contrasts with prior work by (Yeh et al. [5]), which recommends the first few layers for influence analysis.
>
>
> > Page 8: “DataInf and Cosine frequently outperform TracIn, particularly on early–middle attention layers of deeper models, suggesting that these layers encode representations that are most informative for noise filtering.”
> Comment: Given the limited scale of the experiments and the complexity of the problem, it is not clear how strong this empirical evidence is.
>
> We would like to emphasize that this statement above is supported by the ranking results provided in Appendix C (page 17) and we obtain strong empirical evidence:
>
> * Table 3 presents ranks of all configurations per model across 8 GLUE datasets and 10 random seeds, resulting in 80 executions of the experimental pipeline.
> * Across these cases, the first or second attention layer groups **consistently demonstrate the highest performance**, supporting our conclusion that early – middle layers encode representations most informative for noise filtering.
>
> > Page 9: “We validate the hypothesis that NDR and AUC can serve as reliable proxy metrics for influence estimation by …”
> Comment: Kindly explain why intuitively these are good estimations.
>
> Intuitively, Noise Detection Rate (NDR) is a meaningful proxy for influence estimation because it measures the fraction of noisy samples captured among the first k% of least influential samples. A higher NDR indicates that the method more effectively ranks detrimental samples as low influence, meaning that removing these low-influence samples would successfully filter out noise. Similarly, AUC evaluates the ability of influence scores to separate noisy from benign samples across all thresholds, providing a complementary measure of ranking quality.

---

> ### Author Response · Authors · 2025-11-21
> **Response to reviewer z9e9 [4/4]**
>
> > Question. How do you justify the relevance of your work in the presence of the recent observation made in "Do influence functions work for LLMs"?
>
> To summarize, while the study in [1] is very insightful and revealing (which we will also cite/discuss in the paper revision) there are some key differences between our work and theirs. In [1], authors only evaluate influence methods across all model layers, and do **not** examine (a) how influence estimation performance varies with layer choice or (b) how aggregation strategies may improve results, which forms the basis of our work. In contrast to [1], our work investigates **which parts of the network yield better influence estimation** and explores **how independent valuations can be improved with the aggregation of separate objectives**. This perspective **complements prior observations** and provides actionable guidance for influence estimation in large models, directly impacting future work on building better influence methods for LLMs, similar to [1]. Moreover, our work theoretically and empirically refutes existing knowledge in the field, i.e., the seminal work of (Yeh et al. [6]) which instead recommends first few layers for LLM influence analysis over middle/last layers.
>
> ---
>
> > [1] Li et al. “Do Influence Functions Work on Large Language Models?” EMNLP 2025
>
> > [2] Meng et al. "Locating and Editing Factual Associations in GPT." NeurIPS 2022
>
> > [3] Gupta, Akshat and Anumanchipalli, Gopala. “Rebuilding ROME: Resolving Model Collapse during Sequential Model Editing”. EMNLP 2024.
>
> > [4] Meng et al. “Mass Editing Memory in a Transformer” ACL 2024
>
> > [5] Gupta et al. “A Unified Framework for Model Editing” EMNLP 2024
>
> > [6] Yeh et al. “First is Better Than Last for Language Data Influence” NeurIPS 2022
>
> > [7] Coalson et al. "IF-GUIDE: Influence Function-Guided Detoxification of LLMs" NeurIPS 2025
>
> > [8] Askari et al. "LayerIF: Estimating Layer Quality for Large Language Models using Influence Functions" NeurIPS 2025
>
> > [9] Xia et al. "LESS: Selecting Influential Data for Targeted Instruction Tuning" ICML 2024
>
> > [10] Zhang, H. et al. "Correcting Large Language Model Behavior via Influence Function" AAAI Technical Track on Humans and AI, 2025
>
> > [11] Wang et al. "Generalization v.s. Memorization: Tracing Language Models' Capabilities Back to Pretraining Data", ICLR 2025
>
> > [12] Kwon et al. “DataInf: Efficiently Estimating Data Influence in LoRA-tuned LLMs and Diffusion Models” ICLR 2024

---

> ### Author Response · Authors · 2025-11-26
>
> Dear Reviewer z9e9,
>
> Thank you for engaging with us, we really appreciate it. We understand your point regarding [1] and will make a detailed note of the paper and its findings in the related work section of our paper revision. However, despite any potential issues with applicability, influence estimation has been applied across several LLM tasks with varying degrees of success in recent published work [2-6]. Given the extent of their current usage in the community, our work (and associated RQs) seeks to examine and interpret these influence methods, and more specifically how influence varies with layers (localization), as well as how we can combine independent valuations more effectively (aggregation).
>
> As such, we believe that the research question on assessing the _applicability_ of influence estimation in LLMs is one that is beyond the scope of our work. More specifically, we are building upon past work by Yeh et al (2022) [8] and contrasting with their findings on layer choice and score aggregation across layers (i.e. correcting some of the conclusions made in their work). As outlined above, the results we obtain are conducted across a large and extensive set of experiments across models, datasets, with variables controlled (such as the optimization algorithm, data distribution, etc.), and all the empirical evidence we obtain is statistically significant.
>
> Through our work, and by understanding these layer choice and aggregation of scores across the model better, perhaps better influence methods for LLMs can be developed. For instance, as a potential future direction, it would be interesting to assess how the findings in [1] could potentially improve if certain layers are used for influence analysis as opposed to aggregating across all layers as is the default choice in [1]. This layer-specific analysis for influence across model parameters is also supported by findings in prior work (e.g. please see [3] and [7]).
>
> We would like to thank the reviewer once again for their efforts in reviewing our work and for the insightful discussion. As promised, we will definitely include [1] and their corresponding findings in the related work section of our paper. On that note, if the reviewer has thoughts on how they would like us to do this to best guide readers, we can incorporate these suggestions  in the paper revision we upload prior to December 3rd.
>
> Kind Regards,
>
> Authors.
>
> ___
> ___
>
> **References**:
> 1. Li et al. “Do Influence Functions Work on Large Language Models?” EMNLP 2025
> 2. Coalson et al. "IF-GUIDE: Influence Function-Guided Detoxification of LLMs" NeurIPS 2025
> 3. Askari et al. "LayerIF: Estimating Layer Quality for Large Language Models using Influence Functions" NeurIPS 2025
> 4. Xia et al. "LESS: Selecting Influential Data for Targeted Instruction Tuning" ICML 2024
> 5. Zhang, H. et al. "Correcting Large Language Model Behavior via Influence Function" AAAI Technical Track on Humans and AI, 2025
> 6. Wang et al. "Generalization v.s. Memorization: Tracing Language Models' Capabilities Back to Pretraining Data", ICLR 2025
> 7. Meng et al. "Locating and Editing Factual Associations in GPT." NeurIPS 2022
> 8. Yeh et al. “First is Better Than Last for Language Data Influence” NeurIPS 2022

---

### Official Review · Reviewer_B3C8 · 2025-10-25

**Soundness:** 2
**Presentation:** 2
**Contribution:** 2
**Rating:** 4
**Confidence:** 3

**Summary:**

This paper challenges the assumption that embedding layers are best for computing influence functions in LLMs, showing that middle attention layers often perform better and that the "cancellation effect" metric is unreliable. The authors propose novel aggregation methods (Rank and Vote) that outperform standard averaging, and introduce proxy metrics (Noise Detection Rate and AUC) to evaluate influence functions without costly retraining, demonstrating through experiments on GLUE benchmarks across multiple LLMs that their approaches significantly improve detection of detrimental training samples.

**Strengths:**

1. The paper provides both theoretical (Theorem 5.1) and empirical evidence challenging prior assumptions about optimal layers for influence estimation.

2. The paper introduces well-motivated aggregation strategies (Rank and Vote) that outperform standard averaging, revealing layer-specific behaviors and improving influence estimation performance.

**Weaknesses:**

**1. Limited Evaluation Setting**: The experiments rely solely on synthetically injected label noise (20% uniform flipping) on GLUE benchmarks, which may not reflect real-world data quality issues.

**2. Inconsistent Results Across Models**: The findings show notable inconsistencies, particularly for LLaMA-3.2 1B where influence functions fail to outperform random filtering, and the best-performing layers vary across models, suggesting the conclusions may not generalize.

**Questions:**

1. How do the proposed methods perform on real-world noise beyond synthetic uniform label flipping?

2. Why do influence functions fail on LLaMA-3.2 1B, and how to select layers given different LLM architectures?

---

> ### Author Response · Authors · 2025-11-21
> **Response to reviewer B3C8 [1/2]**
>
> We thank the reviewer for the insightful feedback and for suggesting an important direction for extending our evaluation. In the following discussion we address the reviewer’ points.
>
> > 1. Limited Evaluation Setting: The experiments rely solely on synthetically injected label noise (20% uniform flipping) on GLUE benchmarks, which may not reflect real-world data quality issues.
>
> Thank you for raising this concern. To further validate our findings, we conducted additional experiments on the autoregressive datasets introduced in prior influence work by Kwon et al. [1]: **Grammar**, **Math**, and **Math (With Reasoning)**. Unlike the settings used in the main manuscript (which we opted for to have a neat separation between clean and detrimental samples), where detrimental samples were synthetically injected, these datasets contain semantically related instruction categories. In this context, a desirable behavior is that training instructions exert stronger influence on test samples from the same semantic category. Thus, this setting allows us to assess how well influence scores recover meaningful relational structure rather than merely detecting artificial noise. Such scenarios are highly relevant for practical applications including knowledge probing, debugging data pipelines, and detecting harmful or misaligned training examples.
>
> The following links present our additional results with more recent influence methods Outlier Gradients [2] and Kronfluence (EKFAC) [3]:
>
> * Qwen AUC: https://anonymous.4open.science/r/nn-infl-data-6BCA/infl-ds/qwen/autoreg-ds-auc.pdf
> * Qwen Recall: https://anonymous.4open.science/r/nn-infl-data-6BCA/infl-ds/qwen/autoreg-ds-recall.pdf
> * Mistral AUC: https://anonymous.4open.science/r/nn-infl-data-6BCA/infl-ds/mistral/autoreg-ds-auc.pdf
> * Mistral Recall: https://anonymous.4open.science/r/nn-infl-data-6BCA/infl-ds/mistral/autoreg-ds-recall.pdf
>
> The new experiments **support our conclusions from the GLUE benchmark** (i.e. RQ2, Section 5.2):
>
> * Early layers consistently show suboptimal performance in distinguishing beneficial samples.
> * Final layers are also frequently a suboptimal choice for most methods and tasks.
> * The LoRA module B, attached to the v_proj, again demonstrates the strongest and most stable performance in terms of AUC, and Recall.
>
> As requested by the reviewer, we will incorporate these experimental results into the revised version of the manuscript.
>
> > 2. Inconsistent Results Across Models: The findings show notable inconsistencies, particularly for LLaMA-3.2 1B where influence functions fail to outperform random filtering, and the best-performing layers vary across models, suggesting the conclusions may not generalize.
>
> Thank you for raising the excellent question. While we are extremely grateful for the reviewer’s hard work and efforts in reviewing our work, we would like to respectfully and sincerely disagree with the conclusions regarding generalization above by providing additional clarifications, insights, and justifications derived from our experiments. We are looking forward to engaging with the reviewer further and to further strengthening our work with the reviewer’s help.
>
> **First**, we would like to emphasize that many of the presented measurements demonstrate highly consistent patterns across models, including LLaMA-3.2 1B. Specifically, we observe the following trends:
>
> * Downstream task performance trends (Appendix B, Figure 6) demonstrate spikes in the middle layers.
> * Early and middle layer groups ranked higher than others (Appendix C, Table 3) across all models.
> * LoRA module v_proj B influences have higher NDR (Appendix E, Figure 14) across models.
> * Ability of Rank and Vote to improve the influence on many layers and tasks (Appendix H, Figure 18).
> * Similarities in how layers correlate in terms of influence score assignment to training samples (Appendix D, Figure 12).
>
> **Second**, with regard to LLaMA-3.2 1B, we hypothesize that the observed artifacts for the LLaMA-3.2 1B model may stem from its architectural differences, including its substantially smaller number of attention layers (**16**) compared to Qwen (**28**) and Mistral (**32**). The fact that the best-performing layers differ across models suggests that, in practice, achieving reliable influence attribution may require not only well-chosen probing samples but also an effective **weight-localization strategy** that identifies the layers most suitable for attribution. In fact, this aspect of layer dependence in LLMs is observable in other adjacent but orthogonal work, such as knowledge editing. For instance, many knowledge editing methods actually parallel the **localize-then-edit paradigm [4-7]**, where the success of an intervention depends critically on correctly identifying the model layers where the targeted knowledge is stored and only editing those layers for maximum editing performance [4].

---

> ### Author Response · Authors · 2025-11-21
> **Response to reviewer B3C8 [2/2]**
>
> > Question 1. How do the proposed methods perform on real-world noise beyond synthetic uniform label flipping?
>
> We provided a detailed response addressing the first weakness point above. The following links present the new experimental results on additional datasets, as well as the extended evaluation using the newer influence-estimation methods Outlier Gradients [2] and Kronfluence (EKFAC) [3].
>
> * Qwen AUC: https://anonymous.4open.science/r/nn-infl-data-6BCA/infl-ds/qwen/autoreg-ds-auc.pdf
> * Qwen Recall: https://anonymous.4open.science/r/nn-infl-data-6BCA/infl-ds/qwen/autoreg-ds-recall.pdf
> * Mistral AUC: https://anonymous.4open.science/r/nn-infl-data-6BCA/infl-ds/mistral/autoreg-ds-auc.pdf
> * Mistral Recall: https://anonymous.4open.science/r/nn-infl-data-6BCA/infl-ds/mistral/autoreg-ds-recall.pdf
>
> > Question 2. Why do influence functions fail on LLaMA-3.2 1B, and how to select layers given different LLM architectures?
>
> We have already noted architectural differences as a likely contributing factor in our response to the weaknesses outlined above.
>
> LLaMA-3.2-1B is the shallowest autoregressive model in our study (even shallower than RoBERTa-Large’s 24-layer encoder), which likely contributes to its weaker separation between detrimental and beneficial samples when using gradient-based influence scores.
>
> Our practical recommendation for LLaMA-3.2-1B, and consistently across all evaluated models, is to rely on influence scores computed on the LoRA **v_proj B** modules. These weights yield strong NDR performance, and for LLaMA in particular, the LoRA **v_proj B** at layers 8–9 shows potential to outperform the baselines.
>
> ---
>
> > [1] Kwon et al. “DataInf: Efficiently Estimating Data Influence in LoRA-tuned LLMs and Diffusion Models” ICLR 2024
>
> > [2] Chhabra et al. "Outlier Gradient Analysis: Efficiently Identifying Detrimental Training Samples for Deep Learning Models." ICML 2025.
>
> > [3] Grosse et al. “Studying Large Language Model Generalization with Influence Functions” ArXiv, 2023, https://arxiv.org/abs/2308.03296
>
> > [4]. Meng et al. "Locating and Editing Factual Associations in GPT." NeurIPS 2022
>
> > [5] Gupta, Akshat and Anumanchipalli, Gopala. “Rebuilding ROME: Resolving Model Collapse during Sequential Model Editing”. EMNLP 2024.
>
> > [6] Meng et al. “Mass Editing Memory in a Transformer” ACL 2024
>
> > [7] Gupta et al. “A Unified Framework for Model Editing” EMNLP 2024

---

> > ### Comment · Reviewer_B3C8 · 2025-11-27
> >
> > Thank you for your comments. They addressed all of my concerns, so I've updated the score to 6.

---

> > > ### Author Response · Authors · 2025-11-27
> > >
> > > Dear Reviewer B3C8,
> > >
> > > We are grateful for your engagement and are happy to hear that your concerns were addressed. We appreciate your help in strengthening our work and contributions.
> > >
> > > Regards,
> > >
> > > Authors.

---

### Official Review · Reviewer_UsFs · 2025-11-01

**Soundness:** 3
**Presentation:** 4
**Contribution:** 3
**Rating:** 6
**Confidence:** 4

**Summary:**

This paper shows that middle attention layers, not the first embedding layers, are most effective for estimating data influence in LLMs. It proposes new aggregation methods and a Noise Detection Rate metric, achieving better and more reliable influence estimation across multiple models and datasets.

**Strengths:**

1. The paper is clearly written and well organized, making it pleasant and easy to follow.

2. The work challenges the established conclusion [1] that the embedding layer is the most informative for LLM data influence estimation. The authors provide both theoretical and empirical analyses showing that the "gradient cancellation effect" can be unreliable in practice, and offer a counterexample (Theorem 5.1). This contributes a fresh perspective and theoretical insight to the field of LLM interpretability.

3. The motivation is clearly presented and supported by experiments. The proposed method is conceptually simple, easy to implement, and practically useful.

4. The experimental setup is comprehensive, covering multiple datasets and LLM architectures. It also evaluates several influence estimation methods, which strengthens the empirical analysis.

5. The authors have provided open-source code to facilitate reproducibility, which is highly appreciated.

```
[1] Yeh, Chih-Kuan, et al. "First is better than last for language data influence." Advances in Neural Information Processing Systems 35 (2022): 32285-32298.
```

**Weaknesses:**

1. The current baselines are mostly classical. Comparing with more recent influence estimation approaches would make the study more comprehensive and convincing.

2. Some recent studies [2] restrict gradient computation to specific layers for efficiency reasons. Computing influence across all layers in large LLMs could be computationally expensive. A discussion or quantitative analysis of the computational cost of the proposed approach would strengthen the paper.

3. The current analysis primarily reports overall performance metrics. Including visualizations that illustrate how influence rankings of individual samples differ across layers would provide valuable interpretability.

4. The intuition or mechanism explaining why middle layers perform better remains somewhat underexplored. A deeper discussion would be helpful.

5. The paper could also benefit from a more detailed analysis of how architectural differences between LLMs affect the observed influence patterns and performance differences.

```
[2] Chhabra, Anshuman, et al. "Outlier Gradient Analysis: Efficiently Identifying Detrimental Training Samples for Deep Learning Models." International Conference on Machine Learning (2025).
```

**Questions:**

How sensitive is the positional voting approach to k?

---

> ### Author Response · Authors · 2025-11-21
> **Response to reviewer UsFs [1/3]**
>
> We thank the reviewer for the thoughtful feedback and for highlighting valuable directions to improve the clarity and completeness of our work. Below, we address the raised concerns and identified weaknesses to aid in strengthening our work further.
>
> > 1. The current baselines are mostly classical. Comparing with more recent influence estimation approaches would make the study more comprehensive and convincing.
>
> Thank you for the excellent suggestion to improve upon our work. Our choice of baselines was motivated by the desire to analyze influence attribution in a setting comparable to Yeh et al. [1], where TracIn served as the primary point of reference. We included additional classical influence methods to examine the broader generalizability of the conclusion that “the first is not necessarily better than the last.”
>
> To address the reviewer’s concern with classical influence methods, we have extended our experiments to incorporate more recent gradient-based influence estimation techniques. Specifically, we now report results using **Outlier Gradient** [2] (as suggested by the reviewer) and Kronfluence (**EKFAC**) [3], evaluated on the autoregressive datasets considered in the original DataInf paper by Kwon et al [4]. Below, we provide links to the additional experimental results that we plan to incorporate into the revised manuscript in accordance with the reviewer’s suggestion:
>
> * Qwen AUC: https://anonymous.4open.science/r/nn-infl-data-6BCA/infl-ds/qwen/autoreg-ds-auc.pdf
> * Qwen Recall: https://anonymous.4open.science/r/nn-infl-data-6BCA/infl-ds/qwen/autoreg-ds-recall.pdf
> * Mistral AUC: https://anonymous.4open.science/r/nn-infl-data-6BCA/infl-ds/mistral/autoreg-ds-auc.pdf
> * Mistral Recall: https://anonymous.4open.science/r/nn-infl-data-6BCA/infl-ds/mistral/autoreg-ds-recall.pdf
>
> These **figures demonstrate that our findings remain consistent** even when more recent and modern influence methods are included.
>
> * Early layers consistently exhibit weaker performance in AUC and Recall.
> * For both Qwen and Mistral, the LoRA B module (attached to value projections) delivers the strongest performance across most configurations.
> * Classical influence methods tend to show performance degradation in later layers, whereas Outlier Gradient and EKFAC stabilize and reach a plateau.
>
> > 2. Some recent studies [2] restrict gradient computation to specific layers for efficiency reasons. Computing influence across all layers in large LLMs could be computationally expensive. A discussion or quantitative analysis of the computational cost of the proposed approach would strengthen the paper.
>
> Thank you for the excellent point. In general, the dominant cost of influence estimation arises from gradient collection and Hessian inversion; once gradients are obtained, the DataInf inverse-Hessian approximation is computed directly from them using Formula (5) in Section 3.1. To clarify the computational overhead of our approach, we include additional quantitative analysis below. The table reports the mean computation times (in seconds) for our implementation on the Llama-3.2-1B model, illustrating the cost associated with per-layer gradient extraction and influence computation.
>
> Influence computation times (link due to char limits): https://anonymous.4open.science/r/nn-infl-data-6BCA/infl-times/infl-times.md
>
> From this table, we observe the following computational trends:
>
> * **TracIn_we** and **TracIn_we_top_k** from [1] incur the highest preparation time, as they require pairwise train–test interactions and CPU-based searches for common tokens. In contrast, most other methods benefit from GPU-accelerated operations.
> * **Outlier Gradient** exhibits relatively high computation times due to its reliance on a CPU-based OneClassSVM.
> * **DataInf** achieves runtimes comparable to TracIn, as it primarily involves GPU-accelerated dot products between validation–train and train–train gradient pairs.
> * **Cosine** similarity incurs slightly higher cost than TracIn and DataInf, likely due to the additional normalization step.
> * We do not apply **DataInf** or **Outlier Gradient** to the WE layers. DataInf’s memory requirements make its application infeasible for such large layers, while the current implementation of Outlier Gradient requires CPU execution, leading to prohibitive runtimes.

---

> ### Author Response · Authors · 2025-11-21
> **Response to reviewer UsFs [2/3]**
>
> (Continuation of 2.) Furthermore, we would like to emphasize that we had also provided the discussion on time complexity and optimization of gradient computations in **Appendix A (page 14, lines 746-755)**:
>
> **Batching.** Equations 3-5 depend on dot-product $\nabla l^T_{\bar{x}'} \nabla l_{\bar{x}}$. To maintain the gradients for $n$ training samples, $k$ validation samples, and $m$ parameters, TracIn and Cosine requires $O(nkm)$, DataInf -- $O(nkm + n^2m)$ of memory. In one iteration, we pick $n_1$ training and $k_1$ validation samples s.t. $O(n_1k_1m)$ gradients fit the available GPU memory, and compute $O(n_1k_1)$ influence scores between sample pairs. Staying in $O(n_1k_1m)$ limit requires the iteration through all pair of $\lceil n/n_1 \rceil$ training and $\lceil k/k_1 \rceil$ validation batches.
>
> > 3. The current analysis primarily reports overall performance metrics. Including visualizations that illustrate how influence rankings of individual samples differ across layers would provide valuable interpretability.
>
> That is a good point. We agree that this can help improve the paper and provide further interpretability of results. In response to the reviewer’s suggestion, we have conducted additional experiments and provide visualizations that illustrate how influence rankings vary across layers for two different models. We will include these results in the revised manuscript as part of the appendix as well.
>
> * Roberta rank variations: https://anonymous.4open.science/r/nn-infl-data-6BCA/infl-ranks/roberta/infl-rank-variations-median.pdf
> * Qwen rank variations: https://anonymous.4open.science/r/nn-infl-data-6BCA/infl-ranks/qwen/infl-rank-variations-median.pdf
>
> The figures illustrate how the mean influence ranks of noisy and benign samples vary across layers. In this analysis, higher ranks correspond to higher influence values. The trends **“Avg Benign”** and **“Avg Noise”** represent the average rank within each respective subset, while **“Top Ranked Noise Sample”** reflects the most influential noisy sample (on average) at each layer.
>
> These results show that, for most models, the average influence of noisy samples decreases in the middle layers compared to the initial and final layers. However, individual noisy samples with high influence can still appear at these layers, indicating that detrimental examples may exert localized but non-negligible effects even when the overall noise signal is reduced.
>
> > 4. The intuition or mechanism explaining why middle layers perform better remains somewhat underexplored. A deeper discussion would be helpful.
>
> We hypothesize that the observed robustness of middle layers to detrimental samples can be connected to other adjacent but orthogonal work in layer-dependence of LLMs. For instance, findings from the study of model knowledge representation and editing, in particular, locate-then-edit approaches, such as ROME [4], R‑ROME [5], MEMIT [6], and EMMET [7], explicitly identify the layers in which knowledge about specific facts is stored before applying targeted modifications on those layers. Using causal tracing, ROME demonstrates that, in GPT2-XL, factual knowledge is primarily encoded in **middle layers** (e.g., around layer 17). By analogy, these layers may naturally act as a more stable “core representation” within the network, making their activations less sensitive to noisy or misleading training samples, which could explain the superior performance observed in influence-based attribution analyses in our paper as well.

---

> ### Author Response · Authors · 2025-11-21
> **Response to reviewer UsFs [3/3]**
>
> > 5. The paper could also benefit from a more detailed analysis of how architectural differences between LLMs affect the observed influence patterns and performance differences.
>
> From the results, we observe that Llama‑3.2‑1B architecture exhibits structural limitations that reduce the quality of gradient-based influence scores relative to larger or deeper models. The model has only **16 layers** and a hidden (residual) dimensionality of **2,048**, compared to Qwen‑2.5‑1.5B (28 layers, 1,536 hidden) and Mistral‑7B (32 layers, 4,096 hidden). We think that the shallow depth and narrower residual stream constrain the model’s ability to resolve which training examples influence the output. Furthermore, its modest MLP intermediate width (8,192) and limited attention head configuration (8 key/value heads) restrict the non-linear transformations available to encode complex or subtle feature interactions. In the context of influence functions, these architectural limitations can manifest as attenuated sensitivity to individual training samples, resulting in lower contrast between truly informative versus noisy or cross-class examples. Consequently, gradient-based attribution scores derived from Llama‑3.2‑1B may fail to reliably highlight detrimental or misaligned training samples, producing less discriminative influence patterns compared to deeper and wider architectures such as Qwen or Mistral. Nonetheless, the results reveal qualitatively similar NDR distributions and performance spikes as observed in the other models.
>
> In contrast, the larger Mistral and Qwen architectures quantitatively demonstrate that their middle attention layers capture higher rates of noisy samples than Llama in the influence estimation, which we can potentially relate to the models’ architectural depth as well.
>
> > Question. How sensitive is the positional voting approach to k?
>
> Thank you for the great suggestion for an additional experiment. To address the reviewer’s question, we report NDR values for a range of k from 10 to 100 in increments of 10. The figure below shows these results for the **Value-B** LoRA module (the best-performing module by NDR) using **TracIn (Hessian-free)** as the influence estimator.
>
> We provide a set of figures illustrating the variation of NDR :
>
> * NDR vs vote-k and layer on Roberta: https://anonymous.4open.science/r/nn-infl-data-6BCA/vote-k/roberta/layers-ndr-vote_k.pdf
> * NDR vs vote-k and layer on Qwen: https://anonymous.4open.science/r/nn-infl-data-6BCA/vote-k/qwen/layers-ndr-vote_k.pdf
> * Ordered table (from best to worst) Roberta NDR: https://anonymous.4open.science/r/nn-infl-data-6BCA/vote-k/roberta/table.pdf
> * Ordered table (from best to worst) Roberta NDR: https://anonymous.4open.science/r/nn-infl-data-6BCA/vote-k/qwen/table.pdf
>
> The main conclusions from this analysis are as follows:
>
> * Values of k [10,50] yield the strongest NDR performance in most settings.
> * As k increases beyond this range, NDR consistently decreases for **DataInf** and **TracIn**, indicating reduced discriminative quality at larger subset sizes.
> * **Cosine** exhibits a different trend, with its lowest NDR around k=50 and improved performance for larger k values (approximately 60-100).
>
> ---
>
> > [1] Yeh et al. “First is Better Than Last for Language Data Influence.” NeurIPS 2022
>
> > [2] Chhabra et al. "Outlier Gradient Analysis: Efficiently Identifying Detrimental Training Samples for Deep Learning Models." ICML 2025
>
> > [3] Grosse et al. “Studying Large Language Model Generalization with Influence Functions” ArXiv, 2023, https://arxiv.org/abs/2308.03296
>
> > [4]. Meng et al. "Locating and Editing Factual Associations in GPT." NeurIPS 2022
>
> > [5] Gupta, Akshat and Anumanchipalli, Gopala. “Rebuilding ROME: Resolving Model Collapse during Sequential Model Editing”. EMNLP 2024.
>
> > [6] Meng et al. “Mass Editing Memory in a Transformer.” ACL 2024
>
> > [7] Gupta  et al. “A Unified Framework for Model Editing.” EMNLP 2024

---

### Official Review · Reviewer_xbbS · 2025-11-02

**Soundness:** 2
**Presentation:** 3
**Contribution:** 3
**Rating:** 6
**Confidence:** 3

**Summary:**

The paper studies which LLM layers are informative for sample-level influence and proposes simple cross-layer aggregation (Rank/Vote) to combine per-layer scores. It further uses no-retrain proxies (e.g., NDR/AUC) to pre-screen configurations across models and tasks.

**Strengths:**

- Addresses a practical question (which layers to use and how to aggregate) with simple, general aggregation operators (Rank/Vote).
- Provides broad empirical evaluation across models/tasks and includes no-retrain proxies (NDR/AUC) that are useful in practice.

**Weaknesses:**

- **Related-work positioning should be strengthened.** Add a concise paragraph clarifying scope vs. **knowledge editing** (e.g., ROME, MEND, MEMIT):  Explicitly discuss the differences and connections in terms of **“where” (layers/locations)** and **“how” (locality vs. cross-layer aggregation)**. **No new experiments are required**—a brief positioning and citations will suffice.

- **Novelty is relatively simple.** (Aggregation is straightforward; theory is light.)

**Questions:**

Same with weakness

---

> ### Author Response · Authors · 2025-11-21
> **Response to reviewer xbbS [1/2]**
>
> We would like to thank the reviewer for the provided feedback and suggested points of improvement. We appreciate their support in helping strengthen our work and contributions. Below, we answer the raised questions:
>
> > 1. “Related-work positioning should be strengthened. Add a concise paragraph clarifying scope vs. knowledge editing.”
>
> Thank you for the excellent suggestion. As per request, we will add the following discussion on Knowledge Editing (KE) methods in the revised version of the paper:
>
> Knowledge editing encompasses a family of *locate-then-edit* methods that first identify where factual associations are stored in a model and then apply targeted weight modifications. **ROME** [1] and **R‑ROME** [2] use causal tracing, thereby estimating the effect of activation patching on model outputs, to locate the MLP layer whose activations most strongly encode a fact, followed by a rank-one overwrite at that layer. **MEMIT** [3] generalizes this approach by recognizing that knowledge is distributed, computing per-layer causal-effect scores to select top-k contributing layers and distribute edits accordingly. **EMMET** [4] further refines aggregation strategies to reduce redundancy among correlated layers. In contrast, our work investigates which layers are most affected by beneficial or detrimental training samples, rather than factual content. Unlike causal tracing in knowledge editing, we focus on gradient-based influence attributions and their distribution across layers for fine-grained influence analysis. Consistent with findings in KE studies [1-4], we observe that **middle layers exhibit influence signals** that more effectively distinguish between informative and noisy samples.
>
> > 2. Novelty is relatively simple. (Aggregation is straightforward; theory is light.)
>
> Thank you for raising this concern. While we are grateful for the reviewer’s hard work and effort in reviewing our work, we respectfully and sincerely disagree with this assessment of novelty. Our work addresses research questions that have not been explored in prior studies and contrasts with conclusions drawn in prior work in a novel manner (novel theoretical as well as empirical evidence). As the reviewer notes, our research investigates **where** influence attribution methods manifest most effectively within a model and **how** aggregated scoring can be improved when considering multiple independent valuations.
>
> 1. While aggregation may appear straightforward in principle, the space of possible strategies for combining independent valuations is large. Our goal was to explore a subset of promising strategies. In the manuscript, due to space constraints, we report only the most performant aggregation methods based on NDR proxy measures.
>
>     In preliminary experiments, we had actually explored a wider set of variations which we present in the following tables (tables are too big to be placed inline, therefore, we share the link):
>
>     * NDR ranks on Roberta: https://anonymous.4open.science/r/nn-infl-data-6BCA/ndr-aggs/roberta-ndr.pdf
>     * NDR ranks on QWEN: https://anonymous.4open.science/r/nn-infl-data-6BCA/ndr-aggs/qwen-ndr.pdf
>
>     In the  NDR ranking tables for Roberta and Qwen, we tested different aggregation approaches across models.
>
>     * Vote2-c and Rank-c variants were consistently most effective, which is why we describe them in our manuscript.
>     * Suffix “-c” denotes aggregation using only those validation samples that the model correctly predicted in training sample valuations.
>     * We considered the strategies that are more robust to influence score outliers: median, majority, minimal scoring, etc.
>     * Strategies are varied by train-validation set partitioning based on prediction score, gold label, logit distance.
>
>     In the revised version of the paper, we will include a reference to our public code repository with the implementation of attempted aggregation methods.

---

> ### Author Response · Authors · 2025-11-21
> **Response to reviewer xbbS [2/2]**
>
> 2. From a theoretical perspective, we formally present a counterexample to the arguments of Yeh et al. [5] (Appendix G), which has not been done in prior work. Our goal was to formally present the conditions under which a higher cancellation effect actually leads to better separation of detrimental and beneficial samples in the influence score range. This theoretical analysis and empirical evidence of low cancellation effect correction motivated us to consider better performance predictors, such as NDR and AUC.
>
> ---
>
> > [1] Meng et al. “Locating and Editing Factual Associations in GPT.” NeurIPS 2022
>
> > [2] Gupta, Akshat and Anumanchipalli, Gopala. “Rebuilding ROME: Resolving Model Collapse during Sequential Model Editing.” EMNLP 2024
>
> > [3] Meng et al. “Mass Editing Memory in a Transformer.” ACL 2024
>
> > [4] Gupta et al. “A Unified Framework for Model Editing” EMNLP 2024
>
> > [5] Yeh et al. “First is Better Than Last for Language Data Influence” NeurIPS 2022

---

> > ### Comment · Reviewer_xbbS · 2025-11-27
> >
> > Thank you for the detailed and thoughtful rebuttal, as well as the additional experiments and clarifications. The new discussion on knowledge editing and layer localization, together with the broader exploration of the aggregation design space, helps clarify the positioning and novelty of the work. The counterexample to Yeh et al. and the extended aggregation analyses also address my main concerns about the apparent simplicity of the aggregation component.
> >
> > Overall, I feel that my earlier comments have been adequately addressed. I am happy to maintain my marginally positive assessment (score 6) and leave the final decision to the AC.

---

> > > ### Author Response · Authors · 2025-11-27
> > >
> > > Dear Reviewer xbbS,
> > >
> > > Thank you for all your time and effort spent on the review, and for helping us strengthen our work further. We are glad to hear that our rebuttal was able to address your concerns satisfactorily.
> > >
> > > Regards,
> > >
> > > Authors.

---

### Meta-Review · Area_Chair_1MT5 · 2026-01-11

**Summary:**

The reviewers generally found the paper's motivation—identifying optimal layers and aggregation strategies for influence functions—to be interesting and the proposed methods (Rank/Vote, NDR) to be practically useful. However, they raised several concerns that informed the initial decision process:

Reviewers UsFs and B3C8 noted that the experiments relied heavily on classical influence methods (e.g., TracIn) and synthetic label noise on GLUE, requesting more modern baselines (e.g., Outlier Gradient, Kronfluence) and more realistic autoregressive datasets. Reviewer xbbS requested a clearer distinction between this work and Knowledge Editing (KE) literature. Reviewer z9e9 pointed out a missing discussion on recent work questioning the fundamental utility of influence functions in LLMs (Li et al., EMNLP 2025). Reviewers requested more insight into why middle layers perform better, visualizations of influence rankings (UsFs), and explanations for the poor performance on the Llama-3.2 1B model (B3C8). Questions were raised regarding computational costs (UsFs), the sensitivity of the voting parameter $k$ (UsFs), and the robustness of the empirical results given optimization variables (z9e9).

**Reviewer Concerns:**

The authors provided a comprehensive rebuttal that addressed the majority of the reviewers' concerns.

The authors conducted significant additional experiments, incorporating Outlier Gradient and EKFAC methods on autoregressive datasets (Grammar, Math), showing that their conclusions regarding layer choice hold in these settings. This directly addressed the primary concerns of UsFs and B3C8.

The authors added a discussion positioning their work alongside KE methods (localize-then-edit), which satisfied Reviewer xbbS.Mechanisms and Llama Failure: The authors offered a hypothesis linking "middle layer" efficacy to "core representations" found in KE literature. They convincingly argued that Llama-3.2 1B's performance issues stem from its architectural shallowness compared to deeper models like Qwen and Mistral.Technical Details: The authors provided the requested computational cost analysis, visualizations of rank variations, sensitivity analysis for $k$, and clarifications on statistical robustness (10 seeds, deterministic training), addressing the specific technical queries of UsFs and z9e9.

Reviewer z9e9 maintained a philosophical disagreement regarding the relevance of patching influence functions if recent literature (Li et al.) suggests they do not accurately quantify model behavior change. While the authors argued that improving localization/aggregation is a valid contribution given the method's widespread usage, the reviewer remained skeptical of the premise, though this did not result in a negative score.

**Reviewer Scores:**

Reviewer B3C8 participated in the discussion and explicitly stated that the authors addressed all their concern, and raised their score from 4 to 6, though the review is brief.

Reviewer xbbS acknowledged the rebuttal, specifically the KE comparison and theoretical clarifications, and confirmed their score would remain a 6.

Reviewer z9e9 engaged in a debate regarding the Li et al. paper. While they did not fully agree with the authors' stance on the fundamental utility of influence functions, they kept their score at 6. It is unlikely the score would have changed further, as the disagreement was conceptual rather than technical.

Reviewer UsFs did not post a final response to the rebuttal. However, the authors provided every specific item requested by this reviewer: new baselines (Outlier Gradient/EKFAC), computational cost tables, rank visualizations, and sensitivity analysis.

---

### Decision · Program_Chairs · 2026-01-26

Accept (Poster)